# Benchmarking data-driven inversion methods for the estimation of local CO$_2$ emissions from XCO$_2$ and NO$_2$ synthetic satellite images

Diego Santaren[1], Janne Hakkarainen[2], Gerrit Kuhlmann[3], Erik Koene[3], Frédéric Chevallier[1], Iolanda Ialongo[2], Hannakaisa Lindqvist[2], Janne Nurmela[2], Johanna Tamminen[2], Laia Amorós[2], Dominik Brunner[3] and Grégoire Broquet[1]

[1]Laboratoire des Sciences du Climat et de l'Environnement, LSCE/IPSL, CEA-CNRS-UVSQ, Université Paris-Saclay, Gif-sur-Yvette, France
[2]Finnish Meteorological Institute, Helsinki, Finland
[3]Swiss Federal Laboratories for Materials Science and Technology (EMPA), Dübendorf, Switzerland

*Correspondence to*: diego.santaren@lsce.ipsl.fr

**Abstract.**

The largest anthropogenic emissions of carbon dioxide (CO$_2$) come from local sources such as cities and power plants. The upcoming Copernicus CO$_2$ Monitoring Mission (CO2M) will provide satellite images of the CO$_2$ and NO$_2$ plumes associated with these sources at a resolution of 2 km × 2 km and with a swath of 250 km. These images could be exploited with atmospheric plume inversion methods to estimate local CO$_2$ emissions at the time of the satellite overpass and the corresponding uncertainties. To support the development of the operational processing of satellite column-averaged CO$_2$ dry air mole fraction (XCO$_2$) and tropospheric column NO$_2$ imagery, this study evaluates "data-driven inversion methods", i.e., computationally light inversion methods that directly process information from satellite images, local winds and meteorological data, without resorting to computationally expensive dynamical atmospheric transport models. We have designed an objective benchmarking exercise to analyse and compare the performance of five different data-driven inversion methods: two implementations with different complexity for the cross-sectional flux approach (CSF and LCSF) and one implementation for the Integrated Mass Enhancement (IME), the Divergence (Div) and the Gaussian Plume model inversion (GP) approaches. This exercise is based on pseudo-data experiments with simulations of synthetic "true" emissions, meteorological and concentration fields, and CO2M observations in a domain of 750 km × 650 km centred on Eastern Germany over 1-year. The performance of the methods is quantified in terms of accuracy in the single-image (from individual images) or annual average (from the full series of images) emission estimates and in terms of number of instant estimates for the city of Berlin and 15 power plants in this domain. Several ensembles of estimations are conducted, using different scenarios for the available synthetic datasets. These ensembles are used to analyse the sensitivity of the performance to the loss of data due to cloud cover, to the uncertainty in the wind or to the added value of simultaneous NO$_2$ images. The GP and the LCSF methods generate the most accurate estimates from individual images. The deviations between the emission estimates and the true emissions from these two methods have similar Interquartile Ranges (IQR):

between ~20 % and ~60 % depending on the scenario. When taking the cloud cover into account, these methods produce
respectively 274 and 318 instant estimates from the ~500 daily images that cover significant portions of the plumes from the
sources. Filtering the results based on the associated uncertainty estimates can improve the statistics of the IME and CSF
methods, but at the cost of a large decrease in the number of estimates. Due to a reliable estimation of uncertainty and thus a
suitable selection of estimates, the CSF method achieves similar if not better statistics of accuracy for instant estimates
compared to the GP and LCSF methods after filtering. In general, the performances for retrieving single-image estimates are
improved when, in addition to $XCO_2$ data, collocated $NO_2$ data are used to characterise the structure of plumes. With respect
to the estimates of annual emissions, the root mean square errors (RMSE) are for the most realistic benchmarking scenario
20 % (GP), 27 % (CSF), 31 % (LCSF), 55 % (IME) and 79 % (Div). This study suggests that the Gaussian plume and/or the
cross-sectional approaches are currently the most efficient tools to provide estimates of $CO_2$ emissions from satellite images
and their relatively light computational cost will enable analysis of the massive amount of data provided by future missions
of satellite $XCO_2$ imagery.

## 1 Introduction

The satellite imagery of column-averaged $CO_2$ dry air mole fractions ($XCO_2$) has been identified as an essential
component of a future atmospheric observing system to monitor anthropogenic $CO_2$ emissions, and in particular to detect
and monitor hotspot atmospheric plumes and thus emissions, in order to verify emission reductions or assess national
budgets (Ciais et al., 2015; Pinty et al., 2017). The Copernicus $CO_2$ Monitoring (CO2M mission  was designed to meet these
objectives with a constellation of two to three Low Earth Orbit (LEO) satellites flying in a sun-synchronous low-earth orbit
crossing the Equator around 11:30 local time. Each satellite will carry an imaging spectrometer providing images of $XCO_2$
and of $NO_2$ tropospheric column densities (referred to as $NO_2$ hereinafter) along a 250 km wide swath with a resolution of 2
km $\times$ 2 km (Sierk et al., 2019). Current satellite missions, like Sentinel-5 Precursor (Sentinel-5P) and the third Orbiting
Carbon Observatory (OCO-3, when targeting specific sources in its Snapshot Area Map -SAM- mode), already deliver $NO_2$
column-density and $XCO_2$ images, albeit, for the former, at a resolution coarser than CO2M, and for the latter, over areas
and at a frequency much smaller than with CO2M. Upcoming missions, such as Global Observing SATellite for Greenhouse
gases and Water cycle (GOSAT-GW, Kasahara et al., 2020), MicroCarb (in its "city-mode", Pascal et al., 2017) and Twin
ANthropogenic Greenhouse gas Observers (TANGO, Landgraf et al., 2020), are expected to increase the amount of $CO_2$ and
$NO_2$ images of the plumes from emission hotspots.
Operational services are being developed such as the Copernicus capacity for anthropogenic CO2 emissions monitoring
and verification support (CO2MVS, Pinty et al., 2017; Janssens-Maenhout et al., 2020), to process these $XCO_2$ and $NO_2$
images for the monitoring of emissions in a systematic and global way at spatial and time scales that are relevant for
policymakers and to support emission mitigation actions. Plume inversion systems are used to derive estimates of the $CO_2$
emissions from local sources using satellite images of the corresponding atmospheric plumes. One of the key elements of

operational services will thus be standard plume inversion methods providing precise and reliable data in an automated and fast manner. Various plume inversion approaches and implementations are now regularly used to process the existing spaceborne atmospheric plumes images (Varon et al., 2018; Zheng et al. 2020; Kuhlmann et al., 2021; Nassar et al., 2021; Jacob et al., 2022; Hakkarainen et al., 2023a). Therefore, there is a need to benchmark in a quantitative way the plume inversion methods for the estimation of local emissions of $CO_2$, and more generally of greenhouse gases and pollutants.

Monitoring anthropogenic $CO_2$ emissions of point sources or cities from satellite $XCO_2$ images is challenging as corresponding column-average enhancements are often small compared with the local fluctuations of the "background" $CO_2$ field due to biogenic $CO_2$ fluxes and to neighbour anthropogenic sources, and with the typical level of errors in the $XCO_2$ retrievals (Buchwitz et al., 2013). Despite this challenge, the potential of $CO_2$ imagers to estimate anthropogenic emissions has been demonstrated with observing system simulation experiments (OSSEs) using synthetic data, for power plants (Bovensmann et al., 2010), cities (Pillai et al., 2016; Broquet et al., 2018; Wang et al., 2020) and in a more general way, at local to national scales (Santaren et al., 2021). Furthermore, several studies have shown that the joint analysis of co-located $NO_2$ satellite observations strongly enhances the skill to detect the $XCO_2$ enhancement plumes from sources in $XCO_2$ images, and consequently to estimates the corresponding $CO_2$ emissions (Reuter et al., 2019; Kuhlmann et al., 2021). $NO_2$ observations are indeed characterised by a better signal-to-noise ratio and a generally small and low-amplitude background field, due to the relatively short lifetime of nitrogen oxides ($NO_x$).

$CO_2$ emissions of large point sources and cities can be estimated from satellite images by plume inversion systems integrating the observations with dynamical transport model simulations of atmospheric $CO_2$ concentrations (e.g., Broquet et al., 2018; Ye et al., 2020; Santaren et al., 2021). In principle, the use of such dynamical models could support the analysis of the 3D dynamical patterns of the observed plume and thus the accuracy of the inversion. They could also support the derivation of the spatial distribution of the emissions within cities, and of the temporal variation of the emissions corresponding to a plume in the hours preceding each satellite overpass. However they can be strongly impacted by modelling errors which become critical at local scale, when trying to model plumes from emission hotspots over a few tens to a few hundreds of kilometres (Brunner et al., 2023). Furthermore, their computational burden hampers their use for a global and routine coverage of the sources in an operational context. *Data-driven plume inversion methods* appear to be currently more suitable for such wide-scale applications (Ehret et al., 2022). These are computationally light inversion methods that directly process information from satellite images and local winds and meteorological data (typically from operational weather analyses), without resorting to dynamical atmospheric transport models.

The main data-driven approaches for estimating local emissions based on satellite images of plumes that have been tested and analysed in a significant number of studies are:

1) the Integrated Mass Enhancement (IME) approach, which relates the total mass of plumes to the corresponding emissions; it has been used for retrieving $CH_4$ emissions from airborne observations (Frankenberg et al., 2016) or from fine-scale satellite data (Varon et al., 2018)

2) the Gaussian plume approach which extracts emissions from the fit of plume shapes by Gaussian functions and was applied for instance to estimate power plant $CO_2$ emissions from OCO-2 satellite data (Nassar et al. 2017; 2021)

3) the cross-sectional flux approach which infers emissions from the fluxes passing through cross-sections of the plumes and whose potential to estimate $CO_2$ emissions of power plants with $CO_2$ and $NO_2$ satellite imagery data was assessed, for instance, by Kuhlmann et al. (2021)

4) the divergence (Div) approach, which derives emissions from the application of the divergence operator to fields of fluxes and which was originally designed to estimate nitrogen oxide ($NO_x$) emissions from $NO_2$ data provided by the TROPOMI satellite imagery (e.g. Beirle et al., 2019; 2021, 2023) and was more recently adapted to the quantification of $CO_2$ emissions (Hakkarainen et al., 2022). Contrarily to the other methods of this study, the Div method is generally used to generate annual estimates from average fields extracted from multiple images.

Against this background, the aim of this study is to benchmark these four data driven plume inversion approaches for the monitoring of $CO_2$ emission hotspots with CO2M images. We present a benchmarking framework to objectively evaluate and compare the performance of different implementations of the four data-driven approaches (Sect. 2.1) to estimate $CO_2$ local emissions from such satellite data. For this purpose, we use one year of synthetic satellite observations closely mimicking those expected from the upcoming CO2M mission (Sect. 2.2) that were generated in the European Space Agency (ESA) funded SMARTCARB project from high-resolution atmospheric transport simulations (e.g. Brunner et al., 2019; Kuhlmann et al., 2020). The emissions of the city of Berlin and 15 large power plants are estimated from these synthetic satellite data and the ability of the different inversion methods is assessed by comparing their estimates to the corresponding *true* values used by the atmospheric transport model. Performances of the different inversion approaches are evaluated for 1) single-image estimates that are retrieved from daily images (Sect. 3) and, 2) annual estimates that are computed from the inversion of one year of data (Sect. 4). Furthermore, performances are analysed for different scenarios regarding the data used by the inversions, where the impacts of considering the cloud cover in the data, the uncertainties in the wind and the use of collocated $NO_2$ data are assessed. Finally, results are discussed by analysing 1) the potential of ensemble approaches that would gather different inversion methods and, 2) the trade-off between overall accuracy and number of estimates when the cases are filtered based on the uncertainties in the estimates computed by the plume inversion methods (Sect. 5).

## 2 Data and methods

### 2.1 Data-driven inversion methods

Five different emission quantification methods are evaluated in this study: (1) the integrated mass enhancement method (IME), (2) the cross-sectional flux (CSF) method, (3) the light cross-sectional flux (LCSF) method, (4) the Gaussian plume (GP) method and (5) the divergence (Div) method. More precisely, what is studied here are specific configurations of certain methods as is the case for the CSF and LCSF "methods" which are derived from the same general approach. But, hereinafter we will refer to these configurations as methods to avoid weighing down the text. The general approaches have been widely

used and described in previous papers such as Varon et al. (2018) and Beirle et al. (2019, 2021). The specific implementations of the CSF and Div methods tested here have been used extensively by the authors in previous studies (Kuhlmann et al., 2019, 2020, 2021 and Hakkarainen et al., 2022). They have been slightly upgraded in the course of this benchmarking exercise to improve their stability, accuracy, and capability of running in a fully automated way. Details of the methods are presented in an accompanying study by Kuhlmann et al. (2023). Further details about the theory of the Div method and its application are given in Koene et al. (2023) and Hakkarainen et al. (2022, 2023b). All algorithms and tools used in this work have been integrated into a Python library for *data-driven emission quantification* (ddeq), which has been made publicly available and is described in Kuhlmann et al. (2024). We provide below a short description of these methods with an emphasis on their relative advantages and limitations and on the way they estimate uncertainty. The main features of the methods are summarised in Table 1 and illustrated in Figure 1 and Figure A1. Table 1 also lists the computation times of the methods calculated for the same inversion example using the same hardware. As the methods have all been implemented in the same Python package, the timings are directly comparable.

All methods except the Div method can provide estimates derived from individual satellite images. The Div approach as implemented here is based on the averaging of information contained within multiple images and hence typically delivers annual estimates. We will hereinafter refer to the IME, CSF, LCSF and GP methods as single-image methods. These methods share a common algorithmic sequence that starts with identifying clusters of enhancements above a background in satellite images. Subsequently, these clusters are assigned to plumes from specific known sources, and finally, the emissions of the corresponding sources are estimated. The plume detection combines the first two stages and can be used to discern plumes from unreported sources; however the ability of the different approaches to detect unknown point sources has not been studied here, as the primary focus is to analyse their potential to detect and process plumes of known sources from CO2M-like satellite images (see Sect. 2.2). It is worth mentioning that the divergence, cross-sectional flux and machine-learning approaches are particularly well-suited for automatic detection of plumes from unknown sources (Zheng et al., 2020; Beirle et al., 2021; Schuit et al., 2023). Moreover, as previously mentioned, a benefit of the CO2M mission is the availability of co-registered $XCO_2$ and $NO_2$ columns, which can further benefit the plume detection and emission quantification steps.

Obtaining the column enhancements over the background can be achieved with different thresholding techniques as detailed below. When it comes to $NO_2$, the global background field is insignificant but in the case of $CO_2$, its amplitude is important and can vary significantly in space and time due to biogenic and other anthropogenic fluxes surrounding the sources of interest and due to gradients in the background. Another common feature is the need for defining an effective wind speed, which describes the average mass transport of $CO_2$ within the plumes. This a major challenge as wind speed varies with altitude whereas satellite images contain integrated column measurements with no vertical resolution. Additionally, the horizontal resolutions of wind products are generally different from those of satellite images. To address these limitations, the methods determine effective winds in a more or less sophisticated manner.

Finally, all methods have implemented some quality control on their estimates. These checks are more or less restrictive
depending on the methods and may filter out, for example, cases with overlapping plumes originating from neighbouring
sources. Further details are provided in Kuhlmann et al. (2023). It is worth emphasizing  the fact that our implementation of
the GP method discards values that are below 1/4 or beyond 4 times the "true" values averaged one hour before the satellite
overpass (10:00 to 11:00 UTC); this filtering stabilises the otherwise underdetermined inversion. Unlike the other methods,
the GP method thus uses a priori information about the source strength, which artificially improves its performance.
**2.1.1 Cross-sectional flux (CSF) inversion method**
The cross-sectional flux inversion method has been used in many studies such as for example the determination of $CH_4$
emissions of point sources from high-resolved satellite data for which its superiority over other methods has been
demonstrated within the framework of the study of Varon et al. (2018). In brief, this method calculates the fluxes through
single or multiple cross-sections of the plumes as the product of effective winds and integrals of column mass enhancements
along plume transects (line densities). Under the assumption of steady-state conditions, these fluxes are equivalent to the
emissions. The CSF method used in this study has been used by Kuhlmann et al. (2020, 2021) for the estimation of $CO_2$
emissions from $CO_2$ and $NO_2$ images. These studies have demonstrated that the inclusion of $NO_2$ observations significantly
increases the number and precision of the estimates.
The plume detection module of the CSF approach determines in a first stage the $CO_2$ or $NO_2$ pixels that are significantly
enhanced above the background with a statistical z-test (Kuhlmann et al., 2021). To perform this, a Gaussian kernel to
average local observations values is applied and the background field is at this stage computed by applying a median filter.
The parameters defining the z-test were carefully assessed in order to get enough valid pixels to describe a plume while
avoiding false detections (Kuhlmann et al. 2019). The detected pixels are then grouped by a labelling algorithm and assigned
to a source. Finally, a curve representing the centerlines of the plume is fitted to the detected pixels.
For the quantification of $CO_2$ emissions, the CSF method groups the detected plume pixels into sub-polygons along the
curved plume, whose width equals ~5 km (2-3 pixels of CO2M data). All detected pixels within a sub-polygon are used to
construct a single estimate of the line density. Following Reuter et al. (2019), the CSF method assumes that the plume
transect follows a Gaussian behaviour, after removing the background signal with a normalised convolution. To obtain the
line densities, the integration of the fitted Gaussian functions does not require any additional computation as the line
integrals are simply equal to the amplitude parameters of the fitted Gaussian functions. Then, in order to be converted into
fluxes, line densities are multiplied by effective winds which are the horizontal winds at the corresponding source locations
and times of the satellite overpasses, vertically weighted by the GNFR-A/SNAP-1 emission profile (Brunner et al., 2019).
Finally, the $CO_2$ emission of a given source retrieved from a given satellite image is computed by averaging the $CO_2$
estimated fluxes of all the sub-polygons describing the plume downstream of the source. The uncertainty in the emission
estimate is then computed by propagation of the uncertainties in the line densities computation and in the wind; the
uncertainties in the line densities are extracted from the standard deviation of the sub-polygon estimates and capture mostly
satellite data noise through uncertainty in the Gaussian fitting.
When $NO_2$ data are used in conjunction with $CO_2$, detections of plumes are first performed for $NO_2$, while the $CO_2$ and
$NO_2$ enhancements are fitted simultaneously by Gaussian functions that share the same mean (or central location) and the
same standard deviation. Thus, the fit of $CO_2$ enhancements takes advantage of the better signal-to-noise ratio of $NO_2$ data
by better constraining the parameters of the Gaussian functions, which provides more accurate estimates of $CO_2$ line
densities and hence $CO_2$ emissions.
**2.1.2 Light cross-sectional flux (LCSF) inversion method**
The light cross-sectional flux method shares the same theoretical foundations as the CSF method, but its implementation
is largely different. It is derived from the method originally developed by Zheng et al. (2020) to estimate the $CO_2$ emissions
of cities and industrial areas in China that produce atmospheric plumes clearly detectable in transects of OCO-2 data which
are characterised by a resolution of few $km^2$ and by a  swath about 10 km wide, which is almost 25 times narrower than the
~250 km wide swath of the CO2M instruments. This method has been applied to the routine and automatic estimation of
isolated clusters of $CO_2$ emissions worldwide (Chevallier et al., 2020) and to study the temporal variability of the emissions
based on several years of OCO-2 and OCO-3 data (Chevallier et al., 2022). The method has undergone significant
modifications for this comparative study, where the location of the emission sources is known, in order to fully harness the
potential of high-resolution satellite imagery.
For a given source and satellite overpass, the LCSF method performs a simple detection of the plume by extracting from
the satellite image an area which is 100 km wide in across-wind (perpendicular) direction and which extends downwind the
source over a distance equal to the distance travelled by the wind in one hour. The method then selects the pixels of the
extracted area where $XCO_2$ or $NO_2$ enhancements – simply defined as the difference between data values and the average
data of the area – are greater than the spatial variability, i.e. the standard deviation of the data contained within the area.
The quantification of the source emission is then performed on each selected enhancement by extracting again a 100 km
wide across-wind area centred at the enhancements and extending 10 km (~5 CO2M pixels) downwind from the
enhancements. The sums of a linear term accounting for large scale variations in the background fields and a Gaussian
function describing the plume cross-section perpendicular to the wind direction  are then fitted to the data contained within
these areas. The plume detection and fitting of the enhancements can be carried out in the same way when $NO_2$ data are
available. And, standard deviations and means of the Gaussian functions fitted with $NO_2$ data are then used for fitting $CO_2$
enhancements; $CO_2$ data constrain in this case only the amplitudes of the $CO_2$ Gaussian functions. This allows transferring
information derived from $NO_2$ data when estimating $CO_2$ emissions from $CO_2$ data.
$CO_2$ line densities are, as for the CSF method, derived from the Gaussian functions fitted with $CO_2$ data and converted
into emission estimates by the multiplication of an effective wind. For the LCSF method, this effective wind is extracted at
the location of the enhancements and at an altitude above ground of 100 m, as preliminary tests have shown that extracting

winds at the altitude of 100 m yields, for the LCSF approach, better inversion results compared to other altitudes or alternative methods of computing the effective winds. This result may be reflecting a trade-off between the need to account for emission injection heights higher than 100 m when considering isolated power plants, and lower than 100 m when considering the mix of sources within cities, whose emissions are not dominated by large power plants (Brunner et al., 2024). The automatic process of sources limits the ability to derive a case by case selection of the height for the wind extraction, but a finer option for future analysis might be to discriminate this selection as a function of the type of target (considering at least isolated power plants vs. urban areas).

Finally, under steady-state atmospheric conditions, the cross-sectional $CO_2$ flux derived at each selected enhancement is equivalent to the upwind source emissions. Therefore, as several enhancements belonging to a same atmospheric signature of a source are generally processed, the algorithm produces multiple individual estimates of the source emission; the estimate computed by the method for a given source and from a given image is then computed as the median value of these individual estimates; the use of the median helping to reduce the impact of outliers. Moreover, uncertainties in the individual estimates provided by the LCSF method are computed by propagation of the errors derived by the fitting algorithm when generating the line densities; uncertainties in the final estimates are finally the median of these uncertainties.

### 2.1.3 Gaussian plume (GP) inversion method

The Gaussian plume inversion approach assumes that observed plumes can be described with Gaussian plume models. This approach has been widely used such as for example in the determination of $CH_4$ point source emissions (Varon et al., 2018), the use of OCO-2 data to quantify $CO_2$ emissions from power plants (Nassar et al., 2017), or in a framework to estimate at the global scale $CO_2$ emissions from large cities and point sources (Wang et al., 2020). Compared to previous Gaussian plume inversions, the GP inversion method used in this work allows the Gaussian plume model (like the CSF method) to handle curved plumes (see Sect 3.2.1 in Hakkarainen et al., 2023b).

The detection of plumes, i.e. of the $CO_2$ or $NO_2$ enhancements from the background, is carried out using the same algorithm as for the CSF method. Then, the inversion uses a Levenberg-Marquardt least-squares optimization to find the optimal parameters of the Gaussian functions fitting the enhancements and, of the Bézier curves describing the centre lines of the plumes (Hakkarainen et al., 2023b). If $NO_2$ data and $CO_2$ data are simultaneously available, then the Gaussian plume model is first fitted to the $NO_2$ observations and the optimised parameters regarding the plume shape are subsequently used as first guesses for the fitting to $CO_2$ observations. These derived parameters are constrained to remain close to the optimised parameters obtained from the fitting of $NO_2$ data. Finally, the uncertainties in the Gaussian plume estimates are obtained by propagation of the uncertainties in the fitted parameters for the wind speed and for the source strength.

To ensure the convergence of the minimization algorithm, first-guessed values of the fitted parameters need to be carefully prescribed: parameters of the centre-line curves, for example, are initialised from the curves retrieved by the plume detection algorithm, and the initial wind speed is calculated as in the CSF method (see Sect. 2.1.1). Most importantly, the prior values of emission parameters are set to the *true* summertime source emission strength. Thus, unlike any of the other

methods studied in this work, the GP method integrates an important constraint on the emissions which implies that the estimated values, hence the method's performance, are not entirely determined by the information contained within the synthetic satellite observations alone. This limitation should be taken into account when applying this method to invert from real satellite data emissions of sources whose amplitudes are barely known.

### 2.1.4 Integrated mass enhancement (IME) method

The IME method integrates the total mass enhancements of $CO_2$ or $NO_2$ above the background that can be associated with detectable plumes. Then, following Frankenberg et al. (2016), the relationship between IMEs and emissions (Q) can be approximated by a linear relationship defined by the residence times ($\tau$) of the species within the plumes (Eq. 1):

$$Q = \frac{1}{\tau} IME \ (1)$$

$$\tau = \frac{U_{eff}}{L} \ (2)$$

The residence time can in turn be expressed as a characteristic plume length $L$ divided an effective wind speed $U_{eff}$ (Eq. 2) . For example, Varon et al. (2018), who applied the IME method with $CH_4$ observations, derived $U_{eff}$ from 10 m wind speeds using large eddy simulations (LES). Here, the plume detection algorithm which identifies either $CO_2$ or $NO_2$ enhancements from the background is the same as the one used by the CSF and GP methods, but the detected area of the plume over which the integration is performed is dilated using a circular kernel in order to increase the number of integrated pixels (Hakkarainen et al., 2023b). Missing values are filled using a normalised convolution and estimates are rejected when less than 75 % of valid pixels are available for the detected plume.  The characteristic length $L$ is computed from the centre-line of the plume as the arc length to the most distant detected pixel minus 10 km, but at least 10 km. Moreover, the effective wind speed $U_{eff}$ is extracted by using the same vertically weighted average as the CSF method. If $NO_2$ observations are used in conjunction with $CO_2$ observations, the integration area is established by the application of the plume detection algorithm with $NO_2$ data. Then, to estimate $CO_2$ emissions, the IME is calculated over this area with $CO_2$ observations. Finally, the uncertainty in the IME estimates is computed by propagation of uncertainty from the single sounding precision of satellite data and an estimate of the uncertainty in the wind speed.

### 2.1.5 Divergence method

The divergence method, initially introduced by Beirle et al. (2019, 2021), was used to estimate $NO_x$ emissions based on TROPOMI $NO_2$ observations. For this study, the method has been modified in order to estimate $CO_2$ emissions, as outlined in Hakkarainen et al. (2022) where a detailed theoretical analysis of this approach can be found in the supplementary material. The divergence method is based on the continuity equation at steady state (Jacob, 1999), where the divergence of a vector field $F$ (flux) is defined as the difference between emissions $E$ and sinks $S$ (Eq. 3):

$$\nabla \cdot F = E - S \quad (3)$$

$$F = (F_x, F_y) = (\Delta I \cdot U_{eff}, \Delta I \cdot V_{eff}) \quad (4)$$


Since $CO_2$ lifetime is extremely long, the sink term can be neglected. However, before applying the divergence operator to
$XCO_2$ images, the atmospheric background needs to be removed in order to extract purely the $XCO_2$ enhancements. For this
purpose, a median filter is applied to the data and the resulting field is subtracted from the original data. Moreover, in order
to improve the accuracy of the estimates when $CO_2$ noise levels are high, data first undergo a denoising process using a 5×5
pixel mean filter. The flux field $F$ is then defined at each pixel by the Eq. 4where $\Delta I$ is the vertical column density
enhancement above background, and $U_{eff}$ and $V_{eff}$ are the eastward and northward winds, respectively, interpolated at the
location of the pixel and at the time of the satellite observations, and vertically averaged using the GNFR-A/SNAP-1
emission profile (Brunner et al., 2019).
Divergence maps are computed from the mass flux field using a finite difference approximation. The divergence map is
then averaged over a long period to enhance the emission signal, while reducing the impact of noise and the spatio-temporal
variations of the $CO_2$ background. Here, divergence maps are averaged over one year. In theory, the divergence method can
also be used to estimate emissions from single-overpass images such as the cross-sectional flux method (as the two methods
are in theory similar, see Koene et al. 2024). However, we choose in this study to focus on the standard application of this
method (e.g., Beirle et al. 2019, 2021, 2023; Hakkarainen et al., 2022, Sun et al., 2022), which provides temporally averaged
estimates. Appendix A provides a brief overview of the performance when estimating emissions from individual images with
different versions of the divergence approach.
For a specific source, the annual estimate of the emissions is then computed from the enhancement in the averaged
divergence field by using a peak fitting approach which fits the divergence map by a function including a Gaussian and a
linear term centred at the source (Beirle et al, 2021). Emissions, and more generally the parameters, of the peak function are
determined by an adaptive Markov chain Monte Carlo (MCMC) that also provides the uncertainties in the estimates from the
standard deviations of the sampled posterior distributions of the parameters.
**2.2. Synthetic satellite observations of $CO_2$ and $NO_2$**
In this study, synthetic satellite observations of $CO_2$ and $NO_2$ were generated from atmospheric simulations in order to
evaluate and compare the ability of the methods described in Sect. 2.1 for retrieving $CO_2$ or $NO_2$ emissions from point
sources or urban areas using satellite imagery akin to that provided by the upcoming CO2M mission. These simulated
satellite data are readable by the ddeq Python library and were produced as part of the SMARTCARB project and have been
extensively described and used in previous works (e.g. Brunner et al., 2019; Kuhlmann et al., 2019; 2020; 2021). They are
openly accessible from https://doi.org/10.5281/zenodo.4048227  (Kuhlmann et al., 2020b).
Atmospheric concentrations of $CO_2$ and $NO_2$ were simulated by the COSMO-GHG atmospheric transport model (Jähn et
al., 2020) with a vertical resolution of 60 levels up to an altitude of 24 km and with a horizontal resolution of about 1 km × 1
km for a domain centred over the city of Berlin. The domain extends about 750 km in the east-west and 650 km in the south-

north direction. Simulations provided hourly outputs for nearly the entire year 2015. In order to generate realistic simulations, initial and lateral boundary conditions for meteorological variables and tracers were extracted from products of the European Centre for Medium-Range Weather Forecasts (ECMWF) and MeteoSwiss (Kuhlmann et al., 2019). Furthermore, $CO_2$ emissions included both the anthropogenic and biospheric components which were interpolated onto the COSMO grid at a temporal resolution of one hour: anthropogenic emissions were largely derived from the TNO/MACC-3 inventory (Kuenen et al., 2014) and biospheric fluxes were simulated with the Vegetation Photosynthesis and Respiration Model (VPRM, Mahadevan et al., 2008). $NO_x$ emissions were also derived from the TNO-MACC-3 inventory and atmospheric simulations used a simplified $NO_x$ chemistry with a fixed $NO_x$ decay time of 4 hours. $NO_x$ concentrations were converted to $NO_2$ concentrations using an empirical equation for the evolution of $NO_2 : NO_x$ ratios downwind of emission sources (Düring et al., 2011).

To generate synthetic satellite observations similar to CO2M observations, the $XCO_2$ and $NO_2$ column densities derived from the COSMO-GHG simulations were sampled at the resolution of 2 km × 2 km along 250 km wide satellite tracks (Kuhlmann et al., 2019); these tracks were computed using an orbit simulator and correspond to a hypothetical constellation of six CO2M satellites. In addition to $XCO_2$ and $NO_2$ column-average data, a cloud mask was generated from the total cloud fraction computed by the COSMO-GHG model. For $CO_2$ data, all pixels with cloud fraction larger than 1 % were removed as $CO_2$ retrievals are strongly impacted by clouds (Taylor et al., 2016). For $NO_2$ data, less sensitive to clouds, a threshold of 30 % on the cloud fraction was used to select valid pixels (e.g. Boersma et al., 2011). Figure 2 illustrates a COSMO-GHG simulation of $XCO_2$ over the SMARTCARB domain, on which are represented synthetic $XCO_2$ data corresponding to a CO2M satellite overpass.

For the purposes of this benchmarking study, we use the configuration of the SMARTCARB dataset where the CO2M constellation consists of three satellites. By choosing this, we follow the recommendation of Kuhlmann et al. (2021) that a constellation of at least three CO2M satellites is necessary for a proper estimation of the annual emissions from weak sources and in regions such as central Europe where cloud cover dramatically reduces the number of estimates. When ignoring clouds, this constellation of three satellites leads to observing each local source within the SMARTCARB domain once every other day; if we consider that a satellite image is usable if there are at least 50 data pixels next and downwind to the source, then we can use about 3000 images to determine the emissions of the 16 local sources considered in this study. But, if we consider the cloud cover, only 500 images remain usable.

The characteristics of the uncertainties in the synthetic CO2M observations were computed using three different uncertainty scenarios (low, medium, high). Simulated $XCO_2$ column densities were thus assigned random errors by employing various levels of instrumental noise in the error parameterization formula. This formula, used for generating the errors, takes into account the Solar Zenith Angle (SZA) and surface albedos (Buchwitz et al., 2013). The $NO_2$ column densities were assumed to be characterised by random uncertainties of different constant values depending on the chosen uncertainty scenario. These values are defined for clear sky conditions and increase in the presence of clouds; nearly doubling for a cloud fraction of 30 %. No systematic errors were prescribed for either $XCO_2$ or $NO_2$ column averaged data.

In this study, the characteristics of the random uncertainties prescribed to the synthetic data are chosen according to the
requirements of the CO2M mission (Meijer et al., 2019). For $XCO_2$ retrievals, random errors are generated using the error
parameterization formula with a single sounding precision of 0.7 ppm for vegetation albedos and a SZA of 50°. For $NO_2$
retrievals, a single sounding precision in cloud-free conditions of $2 \times 10^{15}$ molecules $cm^{-2}$ is prescribed.

### 2.3. Benchmarking scenarios

The relative performance of the different inversion methods to estimate $CO_2$ emissions are evaluated for the 15 strongest
point sources of the SMARTCARB domain and for the city of Berlin (Fig. 2 and Table 1 in Kuhlmann et al., 2021). These
16 sources cover a large emission range that extends from 3.7 $MtCO_2.yr^{-1}$ for the power plant located in Chvaletice (CZ) to
40.3 $MtCO_2.yr^{-1}$ for the power plant located in Jänschwalde (DE); these values being the annual mean emissions at the time
of the satellite overpass (10:30 UTC) used in the COSMO-GHG simulations. It is worth mentioning that the distribution of
the source emissions is skewed towards the lowest value as the median emission rate in the collection is around 9.6
$MtCO_2.yr^{-1}$ and 75 % of the sources emit less than 14 $MtCO_2.yr^{-1}$.
In order to thoroughly evaluate the relative performance of the different methods and the sensitivity of these
performances to different factors, the benchmarking study is carried out according to several scenarios that share the same
features for the simulated data and for the source collection that have been described above. The most optimistic or ideal
scenario corresponds to the application of inversions to $CO_2$ and $NO_2$ images without the removal of pixels associated to
cloud-cover (ignoring the clouds modelled with the COSMO-GHG model; we label such inversions "cloud-free" hereafter)
and with a perfect knowledge of the wind field (i.e. using directly the winds from the COSMO-GHG model, denoted
SMARTCARB winds). It is the ideal case because 1) the joint analysis of $NO_2$ and $CO_2$ images strengthen the estimates
compared to the analysis of $CO_2$ images only; 2) ignoring the potential loss of data due to cloud cover in the $CO_2$ and $NO_2$
images yield full images, whose analysis is more robust than that of partial images, and thus provides a higher number and
precision of estimates. . The results derived from this benchmarking scenario should be seen as an upper limit of what the
inversion methods could achieve in terms of accuracy and number of estimates. The most realistic scenarios take cloud cover
into account and use winds extracted from the ERA5 wind product (Hersbach et al., 2020) that is independent from the
inverted data and whose resolution (~0.25°) is much coarser than that of the SMARTCARB winds (~0.01°). The results
derived from this benchmarking scenario should be seen as a lower limit for the method's performance.
The differences between the ERA5 and SMARTCARB wind products are significant at the 16 sources considered in this
study: the annual mean biases between these two wind products in 2015 range from 0.1 $ms^{-1}$ to 1.5 $ms^{-1}$ depending on the
source with an average value across the sources of 0.6 $ms^{-1}$ while RMSEs range from 1.1 $ms^{-1}$ to 2.1 $ms^{-1}$ depending on the
source with an average value across the sources of 1.5 $ms^{-1}$ (Fig. A2). The biases per source are systematically positive since
SMARTCARB tends to provide larger winds than ERA5. With such differences, comparing scenarios with the same
characteristics but using different wind products allows us to gain insight into the method's sensitivity to wind uncertainties.
Additional benchmarking scenarios were designed to test the sensitivity of the methods with respect to other factors,
including the consideration of cloud cover in satellite data and the use of $NO_2$ for plume detection and characterization. All
benchmarking scenarios are listed in Table 2.

## 2.4. Benchmarking metrics

For a given benchmarking scenario, the performances of the different inversion methods can be evaluated through the
number of single-image estimates that can be retrieved regarding the number of available satellite images: ~500 or ~3000
considering or ignoring the cloud cover in the data. Performances can be assessed as well through the quality of the
estimates; the accuracies of the methods are then assessed by comparing the estimates retrieved from single satellite
overpasses to the corresponding *true* values that were used to generate the synthetic satellite data. More precisely, inversion
results are analysed in terms of distributions of the differences between the estimated and the true emissions of all the
sources considered in this study. We will refer to these differences in the following as *deviations*. More precisely, our
analysis will mostly focus on examining the distributions of the *relative* deviations, i.e. the differences between estimated
and true emissions divided by the true emissions, in order to fairly compare results across sources with significantly different
magnitudes (Sect. 2.3). Furthermore, to properly describe distributions that may be very different from Gaussian
distributions, box plots are used, in which the median values, the interquartile ranges (IQRs), the 10[th] and the 90[th] percentiles
of the distributions are represented.
The ability of the different inversion methods to estimate source emissions can also be analysed from the study of the
annual or monthly averages of the single-image estimates. Benchmarking results are then evaluated for each source in terms
of relative deviations of the annual/monthly estimates from the annual/monthly true emissions and, in terms of Root Mean
Square Errors (RMSE) in order to provide a global indicator for the accuracy of the annual/monthly estimates across all
sources.
In this study, the annual/monthly averages of the single-image estimates for a given source are computed using three
different methods which are 1) the arithmetic means of all the single-image estimates of the source emission that have been
generated from inverting one year/month of data, 2) the means of these estimates weighted by the inverse of their computed
variances (Sect. 2.1) and 3) the medians of these estimates. The annual/monthly inverse variance weighted means
incorporate the information provided by the methods on the quality of the estimates when averaging, whereas the
annual/monthly medians are statistical indicators that are more robust to outliers than the means. Moreover, since the Div
method is applied by temporally averaging satellite observations over the year, it produces only a single annual estimate for
each source; we will thus consider that the three types of annual/monthly estimates are all equal to this single estimate.
It is important to note that the annual and monthly estimates are affected by temporal sampling biases when inversion
methods use data filtered by cloud cover. Specifically, the presence of denser cloud cover during winter generally results in
over-representation of emission estimates during summer and hence could lead to an underestimation of annual estimates as
emissions are higher during winter due to increased fossil fuel consumption associated with electricity and heat production.
Although more advanced methods, such as fitting periodic curves to capture seasonal cycles as demonstrated by Kuhlmann
et al. (2021) could potentially enhance the accuracy of estimates, they are not included in this study. However, these
temporal sampling biases are integrated in the results as the annual/monthly estimates are compared to the true
annual/monthly emissions which are computed by considering all the days of the year/months.

**3 Results on emission estimates based on individual images**

The following subsections present a comparative study of the CSF, GP, IME, and LCSF methods for estimating emissions
from single images. In the following, we will refer to these kinds of estimates as *single-image* estimates. It is worth
mentioning that, as the methods use different algorithms for plume detection and emission quantification, which include
different rejection criteria (Sect. 2.1), they produce different sets of estimates.

**3.1 Sensitivity to the emission strengths of the sources**

In the optimal scenario (cloud-free, SMARTCARB winds, $CO_2$ and $NO_2$ data), all methods tend to provide more accurate
estimates for strong sources than for weak sources, and this trend is particularly noticeable for the IME and CSF methods
(Fig. 3). The median values of the absolute relative deviations for weak sources (emissions ranging from 0 to 6.9 $MtCO_2$/yr
in the 1[st] row of Fig. 3) are 207 % (IME method) and 54 % (CSF method), respectively. In contrast, for strong sources
(emissions ranging from 15.6 to 53.2 $MtCO_2$/yr in the 4[th] row of Fig. 3), they are approximately 47 % (IME) and 28 %
(CSF), respectively. The inversion methods are also more prone to produce unrealistic values for weak sources as the
distributions are strongly skewed for this type of sources: the 95[th] percentile accuracy indicator is indeed 1128 %, 584 %,
172 % and 178 % for the IME, CSF, GP and LCSF inversion models respectively (1[st] row in Fig. 3). For strong sources, this
indicator is significantly lower, decreasing to 200 %, 108 %, 90 % and 76 %, respectively (4[th] row in Fig. 3). Atmospheric
signals generated by strong sources are more distinct from the background than those from weak sources and as a result, the
signal-to-noise ratio in the $XCO_2$ and $NO_2$ images is better which helps to reduce uncertainties in the determination of their
emissions. For low-emitting sources, the performance of the inversion methods can be degraded by the limited number of
enhanced pixels that are detected in images with noise; this limitation makes the identification of plume centre-lines by the
CSF, IME and GP methods challenging (Sect. 2.1). This problem could have impacted the GP method, but its current
implementation incorporates prior knowledge filtering out estimates that fall outside the 25 % to 400 % range from the prior.
This filtering process is expected to improve the accuracy of the GP method, especially for weak sources.
Biases in the emission estimates may also depend on the strength of the source, as observed in the IME and CSF methods
which strongly overestimate the emissions of weak sources compared to strong sources. For weak sources, the median of the
deviation distributions for the IME and CSF models (blue bars, 1[st] row of Fig. 3) are +116 % and +50 %, respectively,
compared to +16 % and +11 % for strong sources (blue bars, 4[th] row of Fig. 3). This discrepancy is probably due to the
plume detection algorithm, which, for weak sources, may wrongly attribute enhancements from other sources in the vicinity
of the source of interest and thus artificially increase the amplitude of the detected emissions. Conversely, the LCSF
approach tends to underestimate the emissions of strong sources while slightly overestimating those of weak sources, with
the median of the deviation distribution being –26 % (blue bar, 4[th] row of Fig. 3) and +12 % (blue bar, 1[st] row of Fig. 3)
respectively. The underestimation of source emissions could be attributed to a tendency of the method to overestimate the
amplitudes of the background for non-isolated sources: contrary to the other methods, the LCSF method does not remove the
influence of neighbouring plumes when computing the background around a given source. Another explanation could lie in
the fact that this method uses 100-m winds as effective winds while, especially for strong emitting sources, these winds are
lower than the GNFR-A average winds used by the other methods.

**3.2 Impact of the use of $NO_2$ images for the detection of plumes**

The use of $NO_2$ data to identify and characterise plumes increases the number of estimates for all inversion methods
compared to $CO_2$-only inversions, as shown in Figure 4 (blue vs orange bars). The increase is significant for the IME and GP
methods (~93 % and ~70 %), moderate for the CSF method (~34 %), and slight for the LCSF method (~4 %). The IME, GP,
and CSF methods rely on a plume detection algorithm that is less reliable when using only $CO_2$ observations (Kuhlmann et
al. 2019). Of these three, the CSF method requires fewer pixels to detect and quantify plumes, resulting in a larger proportion
of still quantified plume cases than the IME and GP methods when having $CO_2$ data only. The detection of plumes by the
LCSF method is performed on data slices whose pixels are relatively close to sources and where $XCO_2$ enhancement signals
due to emissions are thus relatively strong; this may explain the only small benefit for this method of using joint $CO_2$ and
$NO_2$ images to better determine the shape of the plumes.
When using $CO_2$ and $NO_2$ data, the maximum number of estimates obtained from each inversion method varies
significantly: the IME method produces the smallest number of estimates, with 1661, while the LCSF method produces the
largest, with 2722. The GP and CSF methods, based on the same algorithm of plume detection as the IME method, produce
up to 1776 and 2012 estimates, respectively. These differences can be attributed to the differences in the number of detected
pixels below which the algorithm rejects plumes and, in the emission quantification algorithms used by the different
methods. In addition, the overall complexity of the IME, CSF and GP methods, which use a relatively large number of
rejection criteria likely explains why these three methods deliver much fewer estimates than the LCSF method. The relative
efficiency and robustness of the plume detection algorithm of the LCSF method is evidenced when using $CO_2$ data only to
determine emissions:  the number and accuracy of estimates is hardly changed compared to the inversions performed with
$CO_2$ and $NO_2$ data; contrarily to the other methods whose algorithms are more sensitive to uncertainties in $XCO_2$ data and
which need $NO_2$ data to accurately fit a plume coordinate system to the data.
The inclusion of $NO_2$ data does not appear to significantly improve the overall performance of the GP and LCSF methods
in terms of accuracy of the $CO_2$ emission estimates (lower panel in Fig. 4). However, for the LCSF method, there is a notable
reduction in the 95[th] percentile of the relative absolute deviations from 175 % without $NO_2$ to 115 % with $NO_2$. For the CSF
method, the use of $NO_2$ data strongly improves its overall performance as the 3[rd] quartile and the median of the absolute
residuals are for example significantly decreased, from ~127 % down to ~74 % and from ~54 % to ~36 %, respectively. As

the CSF method rejects fewer estimates when using $CO_2$ data only than the GP method, its accuracy decreases because with a more permissive filtering, it may include complex cases for which emissions are difficult to estimate. This may also explain why the CSF estimates are less biased, with a significantly lower median relative deviation, in cases where inversions also use $NO_2$ data (upper panel in Fig. 4).

In contrast, the precision of the IME method decreases when using $NO_2$ data, but this fact could be related to a numerical artefact: the IME method performs much better for high-emitting sources than for low-emitting sources (see Sect. 3.1) and the use of $NO_2$ data likely allows constraining small sources more efficiently than with $CO_2$ data only. Therefore, when adding $NO_2$ data, the number of low-emitting sources which are estimated increases more than for the high-emitting sources and then the overall performance degrades. This bias associated to the relative bad estimation of low-emitting sources is confirmed when deviations are used to assess performance instead of relative deviations: the absolute deviations associated to the IME estimates globally decrease with the use of $NO_2$ data with for example the median error decreasing from ~15 to ~11.5 $MtCO_2$/yr.

## 3.3 Impact of the cloud cover

The impact of clouds is studied by comparing inversions with cloud-free images to inversions with cloud-filtered images (Sect. 2.3). When disregarding cloudy pixels in the $XCO_2$ and column-averaged $NO_2$ data, the number of estimates from all the methods is considerably reduced, with a decrease of 94 %, 85 %, 85 % and 88 % for the IME, CSF, GP and LCSF methods respectively (Table 3). The number of estimates that can be provided for the cloud-filtered configuration with SMARTCARB winds is at the maximum equal to 313 (LCSF) and decreases to 96 for the IME method which can provide robust estimates for images free of clouds only as this method requires integrating enhancements over the full extent of plumes. As sources are characterized by different cloud covers, the number of estimates per year and per source ranges from 1 to 12 (IME), from 6 to 28 (CSF), from 8 to 23 (GP) and from 15 to 26 (LCSF).

Furthermore, the filtering of data pixels removing those with a significant cloud cover not only affects the number of estimates but also impacts the performance of the methods, although to a much lesser extent. When comparing results obtained from the same images, cloud-free inversions produce slightly better results than cloud-filtered inversions (Fig A3). This is because, in images partially masked by cloud cover, some pixels containing useful information are likely removed, which can lead to less accurate determination of emissions. Consistently, if the threshold of cloud cover above which $XCO_2$ images are discarded for the analysis is increased from 1 % to 2 % or 5 %, the performance of the methods does not significantly increase, unlike the number of estimates, which can increase, e.g. by 12 % and 29 % respectively when using the LCSF method (Fig. A4).

## 3.4 Impact of uncertainty in the wind

As mentioned above, in order to assess the impact of potential uncertainties in the wind, a series of inversions is carried out with a different wind product than the one used to generate the synthetic $XCO_2$ and $NO_2$ data. For this purpose, the

SMARTCARB winds are replaced by ERA5 winds and the differences between these two wind products are characterised at
the sites of this study by random and systematic components (Sect. 2.3 and Fig. A3). Notably, ERA5 winds show
systematically lower values.
For all inversion methods, the global accuracies of the estimates, evaluated in terms of relative absolute deviations, are
only slightly reduced when using ERA5 winds instead of SMARTCARB winds (lower panel in Fig. 4, green *vs* red bars).
There are a few possible explanations for this: the temporal or spatial uncertainties in wind components are only a minor
source of uncertainty compared to other factors impacting the determination of the estimates by the different inversion
methods such as, for example, uncertainties in the $XCO_2$ and $NO_2$ columns densities (Sect. 2.2) or over-simplified
assumptions in plume detection or quantification algorithms. Kuhlmann et al. (2020, 2021) showed, for instance, that the
determination of the $CO_2$ background field could introduce significant uncertainties in the estimates. Furthermore, as
indicated by Reuter et al. (2019), one of the important benefits of satellite imagery is that uncertainties related to
meteorological variables likely average out when emission estimates are sampled along significant areas of plumes.
However, the fact that ERA5 wind values are systematically lower than those of SMARTCARB winds has an impact on
the median values of the relative deviations, i.e. on the biases in the estimates. While the accuracies in terms of relative
absolute deviations are slightly affected by using either wind product (bottom panel in Fig. 4, green *vs* red bars), biases can
be significantly increased, as in the cases of the GP and LCSF methods whose estimates are on average underestimated if
inversions use ERA5 winds instead of SMARTCARB winds. The lower amplitudes of the ERA5 winds explains also that the
results for the IME and CSF methods improve, especially for the 95[th] percentiles of the absolute deviation distributions
which respectively decrease from around 504 % and 411 % to 370 % and 286 % respectively. The systematic overestimation
of the estimates evidenced above for the CSF and the IME methods is therefore mitigated when using ERA5 winds (top
panel in Fig. 4).
As mentioned previously (Sect. 2.3), the benchmarking scenario for which inversions are performed with ERA5 winds
and data filtered for cloud cover, is the closest to real conditions of monitoring emissions from data images delivered by
satellites. For this scenario with $CO_2$ and $NO_2$ data, the GP and LCSF methods show the best performances in terms of
global accuracies with respectively IQRs of 25–62 % and 17–55 % for the distributions of the absolute relative deviations
(red boxes in Fig. 4). It is interesting to note that the overall accuracies of these methods are similar for this realistic scenario
and the ideal scenario where inversions are performed with cloud-free data and SMARTCARB winds. Contrarily, the
number of estimates strongly decreases when inversions are performed with cloud-filtered data such as, for example, from
2722 to 318 estimates for the LCSF method (see Table 3).

## 4 Results on annual and monthly averages of the emissions

### 4.1 Annual estimates

To evaluate how well an inversion method performs on an annual basis, we include all image estimates generated by the method, regardless of their uncertainty. We calculate annual estimates for a given source using three methods, as described in Sect. 2.4: 1) by taking the average of all available image estimates for the source over the entire year, 2) by taking the weighted average of these image estimates based on their uncertainty, and 3) by taking the median value of these image estimates. Because the Div method only provides one estimate per year, its annual estimates are the same, irrespective of the calculation method used. In order to compare for a given source the three estimated annual values to the true emission, we define this latter as the arithmetic mean of the true emissions values for the source over all 365 days of the year.

As noted earlier (Sect. 2.1.5), the Div method computes the annual emission estimate for a given source by averaging the divergence map from all available overpasses in 2015. However, the other methods select overpasses for which they succeed to detect plumes, likely increasing the reliability of their estimates. These selections generally correspond to conditions — in terms of wind, of background variability or of emission strength — that should be favorable to all methods, including the Div method. The lack of selection and thus the use of unfavourable overpasses when applying the Div method may therefore hamper the comparison between the annual estimates of the Div method and that from the other methods.

When annual estimates are calculated as arithmetic means or medians of individual image estimates, the GP and LCSF methods generally outperform the other methods. Indeed, for cloud-free inversions with $CO_2$ and $NO_2$ data, the median deviations for the annual arithmetic means (solid lines, 2nd column of Fig. 5) are 8 % (GP), 14 % (LCSF), 73 % (IME), 35 % (CSF), and 64 % (Div), and the median deviations for the annual medians (dotted lines, 2nd column of Fig. 5) are 14 % (GP), 21 % (LCSF), 54 % (IME), 13 % (CSF), and 64 % (Div). However, if annual estimates are calculated as the means of image estimates weighted by their uncertainty, the relative performance of the methods changes. In this case, the median deviations for annual weighted means (dashed lines, $2^{nd}$ column of Fig. 5) are 28 % (GP), 48 % (LCSF), 46 % (IME), and 12 % (CSF). Thus, using weighted means to calculate annual estimates significantly improves, especially for low-emitting sources, the performance of the IME and CSF methods while having a negative impact on the GP and LCSF methods. This finding indicates the reliability of the uncertainties in the estimates produced by the IME and CSF methods compared to the other methods and, if we use weighted means to compute annual estimates, the accuracies of the IME and CSF methods increase significantly.

Figure 6 displays the inversion results for the annual estimates in a different but complementary way compared to Fig. 5: the estimated annual emissions are represented with respect to the true ones which in particular allows illustrating whether annual estimates are over- or under-estimated for a certain type of source and by a given inversion method. In order to consider the best performance for each method according to what has been shown above, annual estimates represented in the figure, and used for the analysis of the results made below, are arithmetic means of single-image estimates for the LCSF and the GP methods, while they are weighted means for the IME and CSF methods. Furthermore, Fig. 6 illustrates more clearly

than Fig. 5 the fact that, when weighted averages are used as annual estimates, the latter methods produce annual estimates whose precision is comparable for weak *and* strong sources while the global precision of estimates derived from single images by these methods is significantly lower for weak sources (Fig. 3); averaging single-image estimates weighted by their uncertainty thus strongly increases the performance of the IME and CSF methods at the annual scale for low-emitting sources. However, even though the amplitudes of the relative deviations are similar between strong and weak sources, they have opposite signs: annual estimates for strong sources are generally underestimated while annual estimates for weak sources are generally overestimated.

Contrary to the results for the estimates retrieved from single images (Fig. 4), the CSF, GP and LCSF approaches show similar performance, with a slight advantage for the GP method, when estimating annual emissions if we consider the ensemble of the benchmarking scenarios. For example, in the case of inversions from cloud-filtered $CO_2$ and $NO_2$ data and, with SMARTCARB/ERA5 winds, the relative RMSEs are 18/27 % (CSF), 20/20 % (GP) and 17/31 % (LCSF). The analysis of Fig. 3 shows that the LCSF method produces single-image estimates that are slightly more accurate but more biassed than that of the GP method. Thus, the compensation of errors when averaging single-image estimates over a year may be less effective for the LCSF method than for the GP method leading to similar global accuracies for both methods. For instance, the LCSF method has a greater tendency to underestimate high emissions (4[th] row of Fig. 3) which likely explain why, contrarily to the GP method, it systematically underestimates the emissions of the strong emitting power plant located in Jänschwalde, regardless of the inversion scenario (Fig. 6). With respect to its results for single-image estimates, the CSF method has significantly better results at the annual scale when annual estimates are computed as weighted averages of single-image estimates.

Even when annual estimates are computed for the IME method as weighted averages of the single-image estimates, this method still show smaller accuracies compared to the CSF, GP and LCSF methods: the median values of the deviations for the annual estimates are for example 39 % (IME), 20 % (CSF), 11 % (GP) and 21 % (LCSF) when considering the best scores for the inversions performed with ERA5 winds and cloud-filtered data (4[th] column of Fig. 5). The relative performance of the IME method is even worse when analysing the performance in terms of RMSE because, despite a weighting of estimates according to their quality or uncertainty in the annual averages, this method produces for some sources annual estimates that strongly deviate from the actual values, as in the cases of Boxberg or Schwarze Pumpe power plants (Fig. 6). Moreover, the deviations of the Div method compared to that of the CSF, GP and LCSF methods are higher for most of sources except for strong sources (true annual emissions > 15 $MtCO_2$/yr) when inversions are performed using cloud-filtered data and ERA5 winds (4[th] column of Fig. 5).

It is noteworthy that annual estimates for most inversion methods are comparable between inversions using data with or without clouds (comparison between the 2[nd] and 3[rd] columns, Fig. 5), and surprisingly the deviations of the IME and Div approaches are even smaller for inversions with cloud-filtered data. Despite significant differences in the number of image estimates between those two (i.e., cloud-filtered and cloud-free) inversion configurations, annual estimates are *on average* slightly affected when cloud cover is considered in the data, at least for the year and sources examined in this study.

However, even though the relatively small number of image estimates in the inversion configuration with clouds does not hinder most methods from determining annual emissions of most sources, discrepancies can be high for some sources when estimates do not sample correctly the entire year and thus introduce an important temporal bias. For example, the GP method mostly estimates emissions during summer for the Jänschwalde power plant when it uses the cloud-filtered inversion setup, explaining the strong underestimation of the annual emission of this source compared to the cloud-free case (top-left vs bottom-left panel of Fig. 6); this explains additionally why the RMSE increases significantly for the GP method (from 13 % to 20 % when inversions use SMARTCARB winds) when the cloud cover limits the number of single-image estimates. The IME method is also impacted by this temporal bias when the number of estimates is too small to properly capture the seasonal cycle of the emissions, as in the case of the Boxberg power plant. Moreover, whatever the benchmarking scenario, most inversion methods produce annual estimates for all the sources studied in this work, with the notable exception of the Div approach, which estimates annual emissions for only 10 out of 16 sources. This limitation, also present for cloud-free data configurations, is related to the fact that some sources don't produce strong enough divergence peaks from which annual estimates can be made by this method.

As for the results concerning single-image estimates, the use of ERA5 winds instead of SMARTCARB winds has on average a very low impact on annual estimates delivered by the IME, CSF, GP and LCSF methods. For emissions estimated from cloud-free $CO_2$ and $NO_2$ data, the median deviations when inversions use SMARTCARB winds are indeed 46 % (IME), 12 % (CSF), 8 % (GP) and 14 % (LCSF), and when inversions use ERA5 winds, they are equal to 46 % (IME), 12 % (CSF), 9 % (GP) and 12 % (LCSF) as shown in the comparison between the $2^{nd}$ and $4^{th}$ columns of Fig. 5. On the other hand, the overall accuracy of the Div method improves when inversions use ERA5 winds rather than SMARTCARB winds to estimate emissions. In this case, annual estimates are less prone to overestimation due to the generally lower amplitude of ERA5 winds compared to SMARTCARB winds (Fig. A2). This also explains a stronger underestimation of the emissions of strong sources by the LCSF method, resulting in a decrease in the accuracy of the annual estimates for this kind of sources when this method uses ERA5 instead of SMARTCARB winds (left-bottom vs right-bottom panel of Fig. 6).

The overall precision of the annual estimates computed by the IME, CSF, GP and LCSF methods are, for all the benchmarking scenarios, significantly higher than the overall precision of their single-image estimates. For example, when inversions are performed with ERA5 winds and cloud-filtered data, which is the benchmarking scenario with the poorest results, the median deviations of the annual estimates are 39 %, 20 %, 11 % and 21 % whereas the median deviations of the single-image estimates are 73 %, 35 %, 46 % and 37 % for the IME, CSF, GP and LCSF methods. Despite the biases that can hamper the image estimates, the compensation for errors when averaging across a year allow to generate annual estimates that are more precise and this positive effect is amplified when error-weighted averages are used, as in the case of the IME and CSF methods.

## 4.2 Monthly estimates and seasonal cycle

Monthly estimates can be computed using the same three methods as the annual estimates but, according to the results analysed in the former section, we choose to estimate monthly emissions with the method leading to the best performance at the annual scale: monthly estimates are thus calculated as the arithmetic means for the GP and LCSF methods and, as weighted means for the CSF and IME methods. Then, considering the distributions of image estimates month by month allows us to study how well inversion approaches capture the seasonal cycle of the true emissions. The analysis of Fig. 7 shows however that none of them are able to do this when the cloudy pixels are masked: the seasonal cycle of the actual monthly emissions, i.e. maximal/minimal emissions for winter/summer months, is not reproduced by the inversion methods whose estimates are characterised by an erratic monthly evolution leading to inconsistent seasonal cycles. Even though a method correctly estimates annual emissions, some of its monthly estimates can be in important disagreement with the *true* monthly emissions as it is the case for the CSF method on the Heyden source or for the LCSF method on the Dolna Odra source (Fig. 7). Moreover, the methods generally fail to produce estimates for the winter months of the year due to the temporal sparsity of data when the impact of the cloud cover is taken into account.

If the number of estimates is higher, i.e. when clouds are not considered in the data, seasonal cycles derived from monthly estimates are in better agreement with that of the observations for most of inversion methods: the amplitude of the seasonal cycle of the data can be well reproduced as it is the case for the Jänschwalde and Dolna Odra sources for example (Fig. A5). But, the averaged values of the seasonal cycles of the monthly estimates, i.e. the annual estimates, can still be in strong disagreement with that of the data even though the number of estimates is higher; this fact supports the presence of systematic biases in the estimates that was evidenced for most of the methods in the analysis of the results for single-image image estimates (Sect. 3.1).

## 5 Discussion

### 5.1 Accuracy *vs* number of estimates

For a given benchmarking scenario, the analysis conducted in Section 3 has evaluated the performance of the different methods in inferring estimates from individual images by considering all the estimates provided by each method for this scenario. In other terms, the analysis did not integrate any diagnostic regarding the quality of the estimates from these methods. However, we demonstrated in Sect. 4.1 that computing annual means of estimates weighted by their uncertainties can significantly improve the accuracy of the annual estimates when uncertainties are effectively characterised as in the case of the IME and CSF methods. Therefore, a study of the performance of inversion methods for estimating single-image estimates from synthetic $XCO_2$ images should as well integrate a characterization of the quality of its estimates. More precisely, different performance indicators or error estimates can be derived from the application of the inversion methods and such indicators can be used to identify and select the most reliable estimates. Nevertheless, there are no objective criteria

to impose a threshold on the quality of the estimates; higher quality thresholds come with smaller sets of estimates, and
optimal values depend on the inversion method. Indeed, not only do the different inversion methods calculate the
uncertainties in the estimates in different ways but also the computed uncertainties only reflect part of the total/actual
uncertainties, focusing on subsets of sources of uncertainties which differ across the different methods.
For a given inversion method, we attempt an effective quality indicator (QI) which would allow selecting estimates in a
manner that the global accuracy of the method increases when the QI increases, and which would provide indications on the
actual/total errors. We assume that the uncertainties in the estimates derived by the methods provide the best basis we can
get from the algorithms described in Sect. 2.1 for the derivation of such an indicator. In principle, since dealing with sources
of quantitatively different amplitudes (see Sect. 2.3) we should derive the QI in terms of *relative* uncertainties. And, if we
define the QI as a threshold selecting the estimates whose relative uncertainties are below it, we should select the most
reliable estimates regardless of the strength of the source they are associated with. However, this would be true if the
methods perform independently with respect to the amplitudes of the emissions and this is not the case for most methods as
illustrated in Sect 3.1. The CSF and IME methods for example strongly overestimate low-emitting sources compared to
high-emitting sources which implies that the relative uncertainties of weak sources are underestimated by these methods
(Fig. 3). Therefore, if the threshold value of relative uncertainty was decreased, we would tend to select more bad than good
estimates and the overall performance would decrease. Therefore, for these methods, we prefer to select estimates with
respect to their uncertainties, and not to their *relative* uncertainties, which will mitigate the impact of the bias in the
estimation of low-emitting sources.
In any case, determining whether a QI should be based on absolute or relative uncertainties depends on whether the
overall performance of the method improves when estimates with decreasing absolute or relative uncertainties are chosen.
Preliminary tests (not shown here) have established that the overall accuracy of the IME and CSF methods increases when
the *absolute* uncertainty below which estimates are selected is decreased. For the GP and LCSF methods, this behaviour is
obtained when *relative* uncertainties are used to discriminate estimates. Consistently, for all methods, the increase of
performance is then associated with a reduction in the number of estimates and, in order to get a significant number of high-
quality estimates, the value of uncertainty corresponding to the maximal accuracy of the method is arbitrarily set to the 10[th]
percentile of the distribution of the absolute/relative uncertainties. Then, by varying its QI between this value and the
maximal uncertainty of its estimates, each method can be thus associated to a range of accuracies with their respective
number of estimates for a specific benchmarking scenario (e.g. cloud-filtered or cloud-free). In other words, inversion results
can be represented by curves of accuracy *vs* number of estimates, which gives for each inversion method a complete
overview of its performance in terms of accuracy and number of estimates.
To assess the inherent performance of the methods without considering the impact of the cloud cover or of the
uncertainty in the winds, inversion results are analysed for the inversion configuration using $XCO_2$ and $NO_2$ cloud-free data
and SMARTCARB winds, *i.e.* the same winds used to generate the synthetic $XCO_2$ and $NO_2$ observations. Figure 8
illustrates that the overall accuracies of the CSF and IME methods are highly dependent on the selection of their estimates,

and are therefore strongly correlated with their number of estimates. For instance, the IME and CSF methods exhibit large increases in the 3[rd] quartiles of their deviation distribution when the QIs of their estimates decrease: from 81 % to 231 % (IME) and from 43 % to 75 % (CSF) respectively. For these methods, the selection of estimates based on their quality indicators appears to be effective, as the 3[rd] quartiles and 95[th] percentiles, which indicate the proportion of poor estimates, significantly decrease with increasing quality index, *i.e.* with decreasing number of estimates. Therefore, the IME and CSF methods are very likely to produce reliable uncertainty estimates in the individual emission estimates and the definition and derivation of their QI reflect the level of accuracy of their estimates.

The LCSF and GP methods display a slight correlation between most of their accuracy indicators and the number of estimates. For instance, the 3[rd] quartiles of the distributions of relative absolute deviations remain relatively stable, varying only from 46 % to 56 % and from 51 % to 59 % for the LCSF and GP methods respectively, over their entire range of number of estimates. For these methods, the tradeoff between precision and number of estimates is not a critical issue and retrieving an important number of estimates does not imply a significant deterioration in accuracy. On the other hand, this also indicates that the current quality indicators for the GP and LCSF methods do not reflect the total/actual uncertainties in their estimates.

As the methods present different sensitivities of the accuracy to the number of estimates, the relative performances of the methods in terms of accuracy change according to the number of estimates. In other terms, as is the case for the LCSF and CSF methods in Fig. 8, one method may outperform another method depending on the number of estimates we consider. Indeed, below 1000 estimates, the CSF method is characterised by a better precision than the LCSF method for all the statistical indicators and in particular for the 95[th] percentile of the deviation distribution. The best performance of the CSF methods in terms of precision is then reached for ~400 estimates where the median of the deviations is ~25 % compared to ~29 % for the LCSF method. But, if the number of estimates increases beyond 1000, the LCSF method starts outperforming the CSF method with respect to the 95[th] percentile and when estimates are not filtered by their QI (right ends of the curves of Fig. 8), it totally outperforms the CSF method not only in terms of precision but also in terms of number of estimates: if all estimates are considered, the LCSF/CSF method generates 2722/2028 estimates whose deviations from the truth are characterised by an IQR of 17 %–56 %/17 %–75 %. Furthermore, the LCSF method discards outliers much more efficiently than the CSF method insofar as the 95[th] percentile of the deviation distribution is much lower for the former (118 %) than for the latter method (341 %).

Selecting one method over another involves making a trade-off between precision and the number of estimates obtained. Taking the example from Fig. 8, if the primary objective of an application is to obtain as many estimates as possible, the LCSF method would be the preferred choice, as it can provide 2722 estimates with an IQR of the deviations ranging from 17 % to 56 %. On the contrary, if the main priority is to obtain estimates with the highest precision, the CSF method would be more suitable, providing approximately 400 estimates with an IQR of the deviations ranging from 11 % to 45 %. The trade-off between accuracy and number of estimates in the choice of method is even more accentuated in the case where inversions are made with ERA5, as the use of this wind product increases the accuracy of the CSF method through bias compensation

(Sect. 3.4): in this case, using the CSF method, a maximum precision can be obtained, with an IQR equal to 11 %–42 %, for
650 estimates. If, on the other hand, the LCSF method is used, a maximum number of estimates, 2670, can be obtained with
an IQR of 18 %–55 % (Fig. A6).
The difficulty in achieving the best possible precision for a given method lies in determining an appropriate QI for their
estimates. Here, we adopted a relatively simple approach by defining high-quality estimates as those with relative or absolute
errors below the $10^{th}$ percentile of the distribution relative to all the uncertainties of the estimates. However, as seen in the
curves of Fig. 8, highest precision may not be achieved at this value but at a higher one as in the examples of the IME and
CSF method. This is because misleading estimates, such as those resulting from the overlap of plumes from two sources, can
be characterised by very small uncertainties but at the same time by important deviations from the truth, and their impact on
the results becomes significant when the number of estimates gets relatively small. More generally, the QIs defined in this
study reflect the actual uncertainties in the estimates more or less well and the definition of a more reliable QI that ensures
increased accuracy with higher values of the indexes and deliver the maximum achievable precisions for all of the methods
is beyond the scope of this study, as it likely requires extensive studies in order to provide a common and an accurate
characterization of the total uncertainties in the estimates for all the inversion methods. Finally, we will note that all the
qualitative insights stated above about the relationships between accuracy and number of estimates are also valid when
considering inversions using cloud-filtered data and ERA5 winds (Fig. A7).

## 5.3 Single methods vs ensemble approaches

In this study, we create ensemble approaches by averaging the single-image estimates – for the same source and from the
same individual image – produced by different inversion methods. The aim is to obtain more robust and reliable predictions
if individual biases and errors associated with each approach compensate each other. We want thus to analyse whether an
ensemble method, although more expensive from a computational point of view, would perform quantitatively better than a
single method among CSF, GP and LCSF; these methods clearly outperforming the IME method in terms of accuracy and
number of estimates.
Four sets of ensemble approaches are considered: the first one integrates the CSF, GP and LCSF inversion methods, and
the remaining three ensemble approaches integrate pairs of methods (CSF & GP, CSF & LCSF and GP & LCSF). Moreover,
in order to assess the impact of the QIs of the different inversion methods on the performance of the ensemble methods,
results are analysed by considering 1) all the estimates and 2) only the best estimates produced by each method. As results
are assessed for the inversions using ERA5 winds and cloud-filtered data which provide a relatively small number of
estimates, we consider the best estimates as the estimates whose relative/absolute errors are below the $25^{th}$ percentile of their
respective error distribution.
The ensemble approaches do not provide clear improvements in terms of estimate accuracy over the individual methods
from which they are derived (Fig. 9), with the exception of the important number of outliers produced by the CSF method
when estimates are not filtered: the $95^{th}$ percentile of the deviation distribution is equal to 286 % for the CSF method only,

while it decreases to 160 % for the ensemble approach gathering the CSF, GP and LCSF methods. On the other hand, the skewness of the CSF distribution of deviations lead to an increase of the 95[th] percentile of the deviations of the ensemble approaches compared to the 95[th] percentiles of the LCSF and GP methods. Otherwise, the IQR of the deviations are similar for all the ensemble and individual approaches and roughly ranges from 15 % to 65 % when estimates are not selected based on their uncertainty and from 15 % to 60 % when the best estimates are selected. Therefore, errors and biases in the estimates produced by a given method are generally not compensated by the estimates of other inversion methods which suggest that in general, for the same images and sources, the estimates produced by other inversion methods may also present larger errors or similar biases.

The great benefit of using ensemble approaches lies in the significant increase in the number of estimates, which is a crucial issue in the real world when the amount of satellite data is strongly limited by the cloud cover. The ensemble approach gathering the CSF, GP and LCSF methods can supply a maximum of 412 estimates over the year analysed in this study, representing a 30 % increase compared to the LCSF method which is the individual method that supplies the most estimates (318). This result indicates that the CSF, GP and LCSF methods can provide estimates from different images, i.e. if one method does not provide an estimate from a given image, another method from the ensemble may, conversely, provide one (Fig. A8). This allows the ensemble method to produce a maximum number of estimates (412) that is close to the number of usable satellite images (~500). When only best estimates are considered, the ensemble approach generates more than twice as many values compared to the LCSF method (195 *vs* 80) whereas the other ensemble approaches (CSF & GP, CSF & LCSF and GP & LCSF) only provide about 140 estimates.

While combining the estimates generated by the CSF, GP and LCSF methods seems to be the optimal choice for an ensemble approach providing the largest number of predictions, the computational cost of using these methods together may not outweigh the benefits in terms of number of estimates compared to using a single method. For example, in the most realistic scenario of inversions conducted with cloud-filtered data and ERA5 winds, the computational time required for the CSF-GP-LCSF ensemble method is more than three times that of the LCSF method alone (see Sect. 2.1) whereas the overall precision of the LCSF method is better and the increase in the number of estimates is only 30 % when using the ensemble approach. Therefore, if the performance of computer systems remains an important factor to take into account, one would prefer to use the LCSF method, which is the fastest method of this study, instead of using an ensemble approach.

In order to investigate the benefit of using ensemble approaches for the estimation of annual emissions, we use the same three individual methods that produce much better results than the IME and Div methods (see Sect. 4.1), but we consider different definitions of the annual estimates depending on the inversion method: annual estimates are arithmetic means of image estimates for the LCSF and the GP methods whereas they are weighted means for the CSF method. This choice corresponds to the best performance at the annual scale that has been found in this study for each method (Sect. 4.1.) Besides, no selection of the estimates was performed to compute the annual estimates although the quality of the estimates is integrated within the annual estimates of the CSF method which are averages weighted by the errors in the estimates. Among the ensemble methods considered here, only the approach gathering the CSF and GP methods yields better results than the

best individual method composing it for most of benchmarking scenarios (Fig. A9). For example, when inversions are
performed with cloud-filtered data and SMARTCARB winds, the CSF, GP and their ensemble approach are characterised by
relative RMSE equal to 18 %, 20 % and 16 %, respectively. The benefit of using ensemble methods for estimating annual
estimates is thus questionable, especially considering that the gain in accuracy, if any, is very small compared to the
individual methods which, depending on the inversion scenario, produce the more accurate annual estimates. This is due to
the fact that the inversion methods generate annual estimates that are generally biased in the same way: emissions of strong
sources are generally underestimated while emissions of weak sources are generally overestimated (see median values in
Fig. 6).

## 6 Conclusions

In this paper, we tested and benchmarked several lightweight data-driven inversion methods for estimating local (city and
power plant) emissions from $XCO_2$ and $NO_2$ satellite images. The five methods that have been studied are the Integrated
Mass Enhancement (IME), the Cross-Sectional Flux (CSF), the Gaussian Plume (GP), the Light Cross-Sectional Flux
(LSCF) and the Divergence (Div); this last method generating only annual estimates. In a domain centred over the city of
Berlin, which extends about 750 km in the east-west and 650 km in the south-north direction, inversions were performed
with almost one year of synthetic SMARTCARB $XCO_2$ and tropospheric column $NO_2$ satellite observations with similar
characteristics as the upcoming CO2M mission. The ability of the inversion methods to estimate emissions has been assessed
by comparing the deviations of estimates from the corresponding "true" values used in the simulations, for 16 sources
including the city of Berlin and 15 power plants. To get a complete overview of performance, several benchmarking
scenarios were considered in order to analyse the benefit of using auxiliary $NO_2$ data or the impacts of the cloud cover in the
data or of uncertainties in the wind data.
In terms of quantifying emissions from single satellite images, the implementations of the CSF, GP and LCSF methods
used in this study outperform that of the IME method. Furthermore, we have demonstrated that the performance in terms of
accuracy and number of estimates varies, to a greater or a lesser extent depending on the method, with the selection of the
estimates based on their relative or absolute uncertainty. The overall accuracies of the IME and CSF methods are
significantly enhanced when a strict screening for high quality estimates is applied but at the cost of an important decrease in
the number of estimates. The GP and LCSF methods, on the other hand, perform more robustly showing only a variation in
their global precisions with increasing quality screening. This behaviour points out the need for these methods of a better
characterization of the uncertainties in the estimates. When estimates are filtered, the CSF method yields the best results in
terms of accuracy while, when estimates are not filtered, the LCSF method provides the highest number of estimations with
a slight decrease in accuracy. Overall, the CSF, GP and LCSF methods show similar accuracies for all the benchmarking
scenarios and when the less reliable estimates of the CSF method are removed: most of IQRs of the absolute deviations
range from 15 % to 60 % with an average median around 35 %. Moreover, for the most realistic benchmarking scenario, i.e.
for the inversions using cloud-filtered $NO_2$ & $CO_2$ data and ERA5 winds, the IME, CSF, GP and LCSF methods generate on
average 6 (IME), 18 (CSF), 17 (GP) and 20 (LCSF) estimates per source and per year with great differences between sources
(See Sect. 3.3), which is equivalent to a maximum number of estimates equal to 96 (IME), 295 (CSF), 274 (GP) and 318
(LCSF) for all 16 sources. These figures are significantly lower than the number of usable images (~500) that can provide a
hypothetical constellation of 3 satellites as analysed here; this suggests that methodological improvements could increase the
number of estimates.
The accuracy of the CSF and IME methods was found to depend on the strength of the sources with important errors
when determining low emissions; the GP and LCSF methods, in contrast, show similar performances across different ranges
of emissions. Moreover, the advantage of using co-located $NO_2$ signal for plume detection and quantification appeared to be
clear for the CSF, IME and GP methods, for which the number of single-image estimates significantly increased, while it
was rather weak for the LCSF method. When a cloud cover mask was taken into account in the data, the number of estimates
significantly decreased for all the inversion methods with an average reduction of 85 %; the global precision however hardly
decreased and even improved for the IME method. For all the inversion methods, the sensitivities of the results to wind
uncertainties were surprisingly found to be insignificant when replacing the SMARTCARB winds (used in the simulation)
by ERA5 reanalysis winds. Finally, if we do not take computational cost into account, the interest in using ensemble
approaches instead of a single method lies mainly in an increased number of single-image estimates as the availability of
estimates from the different methods complements each other.
Part of the effectiveness of the implementations of the cross-sectional flux method may come from the generation of
multiple estimates of cross-sectional fluxes along plumes and the subsequent averaging in order to get an unique emission
estimate for a given source and satellite overpass. Probably, errors in the satellite data or in the simplifying assumptions of
the cross-sectional approaches partly cancel out when averaging. The CSF implementation uses a complex algorithm of
plume detection which makes it possible to use the total detectable plume, probably leading to more accurate estimates than
for the LCSF implementation, which only uses observations near the source. However, the plume detection and the
computation of the curved centreline can fail for weak sources (i.e. short plumes) at the cost of having a large number of
outliers. On the contrary, the LCSF implementation uses a simpler but more robust algorithm that uses the wind vector to
estimate the location of the plume, which likely explains why this method generates more estimates, and without the need of
$NO_2$ data, compared to the CSF implementation. However, efforts should be made to correct the systematic underestimation
of strong emissions by the LCSF implementation. A way forward can be merging the CSF and LCSF method into a single
algorithm that takes the advantages of both approaches.
When compared to other methods, the relative ability of the GP method in estimating emissions probably relies on the
use of a Gaussian function whose optimization determines the emissions while taking into account the entire structure of the
plumes, and calculating effective winds that are consistent with that of the plumes. However, this optimization and thus the
performance of the GP method highly depend on the first-guessed values to be assigned to its parameters (not shown). And,
in this study, the first-guessed values of the emissions are the summer average emissions for each source; this could be a
strong constraint on the estimated values and could lead to an overestimation of the GP performance in this benchmarking
study. Finally, the GP method is computationally expensive due to the heavy plume detection algorithm and to the multi-
parameter optimization required for the Gaussian fitting of the plumes (Table 1).

878        The IME method also integrates information retrieved from the entire structure of the plumes but, contrarily to the GP
method, it does not use this information when computing effective winds. Therefore, these winds may be inconsistent with
the characteristic lengths of plumes used by the IME method to estimate $CO_2$ emissions (Sect. 2.1.4) and this could explain
the relatively poor performance of the IME method in this study. Varon et al. (2018) probably found that the IME method
was adapted to estimate $CH_4$ emissions from high-resolution plumes because they inferred a relationship between the
effective winds and the characteristic lengths through LES simulations. Another drawback of the IME method is that it is
very sensitive to missing data as it needs an entire coverage of the plume area by data to efficiently integrate the total mass
enhancement. Other single-image methods (GP, CSF and LCSF) are less sensitive to missing data as they fit functions to the
data and can handle data gaps; this explains why these methods provide a much larger number of estimates when the impact
of cloud cover on the data is considered (see Sect. 3.3).

888        In this study, we chose not to analyze the potential of the divergence method for estimating instant emissions from single
satellite overpasses because of the lack of studies on such an application of this method. As highlighted in the introduction
section, our aim is to compare proven approaches for the local scale estimation of strong sources (such as the application of
the divergence method to time-averages of satellite images). Moreover, the strong spatial variability of the divergence fields
derived from single images suggest that only averaged fields could be processed properly with the version of the divergence
approach which is used here for annual estimates and which relies on the peak-fitting of temporally averaged divergence
fields. However, we have conducted some preliminary analysis on a version of the divergence method which instead
integrates the divergence signal spatially (over disks centered on the sources). The results, documented in appendix A,
demonstrate that with a range of integration radii close to that of the spatial resolution of image, this approach can yield
estimates that would be comparable in terms of accuracy and quantity to that of the best inversion methods of our benchmark
evaluation for single-image based estimates. A better understanding of the behavior of this approach as a function of the
integration radius, and an assessment of the estimation errors are needed to conduct a proper comparison to the other
methods. This deserves further investigations. However, these preliminary results raise optimistic perspectives regarding the
potential of using the divergence method for estimating instant emissions from single-overpass images.

902        For estimating annual emissions, the CSF, GP and LCSF methods outperform the Div and IME methods when annual
estimates are computed as error-weighted means of single-image estimates for the CSF method and as arithmetic means of
these estimates for the GP and LCSF methods. Across the different benchmarking scenarios, the GP method shows better
precisions in its annual estimates because its single-image estimates have similar absolute deviations from the truth but are
less affected by biases compared to the CSF and LCSF methods (see Fig. 3). However, despite biases, errors in the single-
image estimates provided by the CSF, GP and LCSF methods likely compensate when averaging and these methods also
generate annual estimates with a better precision than for their single-image estimates. In the most realistic benchmarking

scenario – where inversions use cloud-filtered $XCO_2$ & $NO_2$ data and ERA5 winds and where performances are the lowest compared to other scenarios – the relative RMSE for the annual emissions of the 16 sources is 20 % (GP), 27 % (CSF), 31 % (LCSF), 55 % (IME) and 79 % (Div). The relatively weak performance of the Div method could be explained by the fact that this method was originally developed for the estimation of $NO_x$ emissions and the fields of this chemical species are generally characterised by stronger divergence peaks than for $CO_2$ fields. Its performance may also be hindered by the fact that our implementation of this method does not select the overpasses from which the annual divergence maps are derived (see Sect. 4.1). Further investigation is needed to determine whether the filtering of overpasses which could be favourable to the method could strongly increase the accuracy of its annual estimates. The performances of ensemble approaches gathering several inversion methods in terms of annual estimations is not better, and in some cases even worse, than the individual methods. Finally, none of the methods were able to correctly reproduce the monthly seasonal cycle of the emissions when data underwent a cloud-filtering, i.e. when data were not available for some months, which points out the need for an extensive temporal coverage of the observations when aiming to capture the monthly variability in emissions.

In addition to the technical improvements that could be made on the algorithms of the methods, further developments could extend this study such as the integration of new data streams for estimating $CO_2$ emissions such as satellite data of other co-emitted gases than $NO_2$, e.g. CO data provided by the TROPOMI instrument. A companion paper (Hakkarainen et al., 2024) analyses the ability of the inversion methods in determining $NO_x$ emissions, from synthetic and TROPOMI $NO_2$ satellite data for the Matimba and Medupi power plants in South-Africa. The $NO_2$ synthetic data are extracted from the high-resolution MicroHH Large Eddy Simulations (LES) (Van Heerwaarden et al., 2017) and used in particular to study the nitrogen dioxide to nitrogen oxide scaling factors that are required for satellite-based estimations of $NO_x$ emissions. Moreover, the capacity of the inversion methods to estimate city emissions has been analysed in this study on the single example of the city of Berlin and, as most of the methods have provided correct estimates for its emissions, it would be interesting to expand this study to other cities and other local sources. Finally, this benchmarking study has not integrated the new and promising type of inversion methods that are the methods derived from deep learning techniques (e.g. Lary et al., 2016). After a potentially complex training phase, deep-learning methods could quickly process large amounts of data and provide estimations with similar or better accuracy than the methods studied here (Dumont le Brazidec et al., 2023). They could also complement these methods by allowing a fine differentiation of the plumes compared to the background with advanced image segmentation techniques.

The aim of this study is to contribute to the development of the $CO_2$ Monitoring and Verification Support system that will use the upcoming CO2M satellite data. And, although this benchmarking study has been performed with synthetic observations, the methods studied here can be easily adapted to the analysis of real satellite observations and to deal with sources of unknown location as demonstrated in Hakkarainen et al. (2024).

## Appendix A: Potential of the divergence approach to estimate local $CO_2$ emissions from single-overpass satellite images of $XCO_2$ and $NO_2$

In this study, the performance of the divergence approach to estimate local $CO_2$ emissions from $XCO_2$ and $NO_2$ synthetic satellite images is assessed with a standard version of this approach (e.g., Beirle et al., 2021; Hakkarainen et al., 2022), which provides temporally averaged estimates. Results concerning the divergence approach are thus analyzed in the main part of this paper in terms of annual means. However, following the suggestions of a reviewer (S. Beirle), we also tested the potential of this method to estimate instant emissions using single-overpass images. For this purpose, we have used two versions of the divergence approach that have been modified for single image geometry as in Beirle et al. (2023).

For both versions, the computation of the divergence fields is performed by only considering the "advective" term ($10^6 * M_{air} * U * \nabla(VCD)$) of the full expression of the horizontal flux divergence ($\nabla(10^6 M_{air} * U * VCD)$) where $M_{air}$ is the dry air mass, $U$ is the wind vector and $VCD$ is the vertical column density in parts per million. Such reformulation of the divergence method that does not compute the divergence of the wind term was also used by Beirle et al. (2023) for $NO_2$. The advantage of this reformulation for $CO_2$ is that the background (e.g., a constant offset of 400 ppm) is implicitly removed.

These versions of the divergence approach differ from each other in their way of computing emissions from the divergence maps associated with single-overpass images: the first version integrates the divergence fields on disks centered on the sources (Figure A10). And, to mitigate the impact of the uncertainties in the observations, the emission estimate for a given satellite overpass and source can be computed as the average of the estimates when integrating the divergence signal on disks of different radii. This version of the divergence approach will be referred to hereinafter as the *integral* divergence method. The second version proceeds in a similar way to the one used in the main part of the article and fits a 2-D Gaussian function to the divergence maps in order to retrieve source emissions (e.g. Beirle et al. 2020). The modified peak fitting model is similar to the original but with a reduced number of estimated parameters. Namely, the parameters related to the background and to the location correction are removed from the model parameters. This version of the divergence approach will be referred to hereinafter as the *peak-fitting* divergence method.

For both versions, potential peaks are detected by using $NO_2$ fields which are integrated over disks of 6 km radius centered on the sources. If the integral of the divergence map on the disk is larger than the integral on the area outside the disk, then the enhancement, related to a given source and for a given satellite overpass, is considered strong enough and the emission estimation can be carried out. Many sources in the SMARTCARB dataset are weak and enhancements may be barely visible which causes challenges for both versions.

To evaluate the potential of these two versions of the divergence approach, we use the SMARTCARB dataset described in section 2.2. which provides about 3000 images to determine the emissions of the 16 local sources that are considered in this study (if we take into account the cloud cover, only 500 images remain usable). Furthermore, we consider two

benchmark scenarios (see table 2 and section 2.3) where inversions are performed using $CO_2$ and $NO_2$ data with
SMARTCARB winds. In one case, we use cloud-free data, while in the other, cloud-filtered data.
The analysis of the deviations from the truth of the instant estimates shows that the integral divergence approach is
strongly sensitive to the radius of the integration disks (Fig. A11). No clear trend appears except that errors increase sharply
for a radius greater than 10 km, with a significant presence of outliers. Below this value, the absolute relative deviations
(bottom panel of Fig. A11) can increase or decrease depending on the value of the radius Furthermore, the integral
divergence approach can underestimate or overestimate emissions depending if the radius is lower or greater than ~4 km. A
possible explanation for this behavior could be that the impacts of the two main sources of errors in the divergence method
— namely, the uncertainties in the observations and the influence of additional but unwanted sources on the background of
the divergence fields — evolve in opposite directions as the integration radius increases. The impact of the uncertainties is
mitigated when the area of the integration disk increases because errors have more probability to cancel out. Conversely, the
impact of neighboring sources on the background of the divergence field intensifies as the integration radius increases,
because the likelihood of capturing features in the divergence maps that are not directly related to the emissions of the
targeted sources grows. This impact consistently introduces a positive bias in the estimates (as we capture more sources) and
is likely more important than the one related to the uncertainties as performance overall degrades when the integration radius
increases.
The peak-fitting divergence method is characterized by a poor performance compared to the integral divergence method
for the ensemble of integration radii that we have considered here (Fig. A11). The estimation of small emitting sources may
be more difficult for the peak-fitting version as the fit of the 2-D Gaussian function to the data associated to these sources
often fails and does not provide optimal and reliable parameter combinations, yielding poor and often overestimated
emission estimations. Therefore, even though the peak-fitting divergence method is generally more efficient at the annual
scale, these results suggest that it is not the case when estimating instant emissions from single overpass images.
The configuration of the integral divergence method which averages estimates across the integration radii of 2, 3 and 4
km shows the best performance amongst the configurations that we have tested. Probably, the impacts of the data
uncertainties and the background are well balanced for this range of radii and the fact of averaging estimates across three
different radii further reduces the influence of the data uncertainties on the results. When compared to other inversion
methods analyzed in this study, the performance of this configuration of the integral divergence method is similar to that of
the best inversion methods (Fig. A12). For the benchmarking scenario considering cloud-free data, its relative absolute
deviations are for example characterized by a median value of ~38 % and Interquartile Range (IQR) of [~19 % − ~64 %]
which are comparable to deviations associated to the Light Cross-Sectional Flux (LCSF) method which have a median value
of ~32  % and an IQR of [~15  % − ~56  %]. Notably, the integral divergence method generates fewer estimates (2174)
compared to the LCSF method (2722), but more than the Gaussian Plume (GP) method (1776).
These preliminary results regarding the potential of the integral divergence method for estimating local $CO_2$ emissions
from single-overpass images of $XCO_2$ and $NO_2$ appear promising, especially since this method allows for the detection of

plumes from unknown sources (Beirle et al., 2021). However, further investigation is required to properly assess factors such as the integration radius based on data resolution, and to generalize this method to various types of satellite data. Additionally, a thorough quantitative error assessment is essential to evaluate the accuracy of the estimates, enabling the classification and selection of estimates, which would enhance the method's overall performance.

*Code and data availability.* The code repository of the python package *ddeq* is available on Gitlab.com: https://gitlab.com/empa503/remote-sensing/ddeq. The SMARTCARB dataset is available on Zenodo: https://doi.org/10.5281/zenodo.4048227.

*Author contributions.* DS made the diagnostics and led the analysis for the intercomparison of the results from the different inversion methods. All co-authors contributed to the decisions for the configuration, diagnostics and analysis of the intercomparison. DS wrote the manuscript with inputs from all co-authors. DS, GB and FC carried out the analysis specific to the LCSF method. JH, II, HL, JN and LA carried out the analysis specific to the Div method. GK developed the original ddeq library that has been used as a basis for the application of the different methods. GK provided the SMARTCARB dataset used to test the different methods. GK carried out the analysis specific to the IME method. EK carried out the analysis specific to the CSF and GP inversion methods. The project was coordinated by JT, DB and GB.

*Competing Interests.* Some authors are members of the editorial board of Atmospheric Measurement Techniques. The authors have no other competing interests to declare.

*Acknowledgements.* Most of the work performed in this paper was done in the framework of EU H2020 project CoCO2 (grant No. 958927). The FMI team would like to thank the Research Council of Finland project 353082. All authors would like to thank the ICOS Carbon Portal for providing access to their JupyerLab servers, which were used for code development and data sharing. Finally, the authors would like to thank the two reviewers for their insightful comments, and especially S. Beirle for his suggestions on the application of the divergence approach for estimating instant emissions.

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

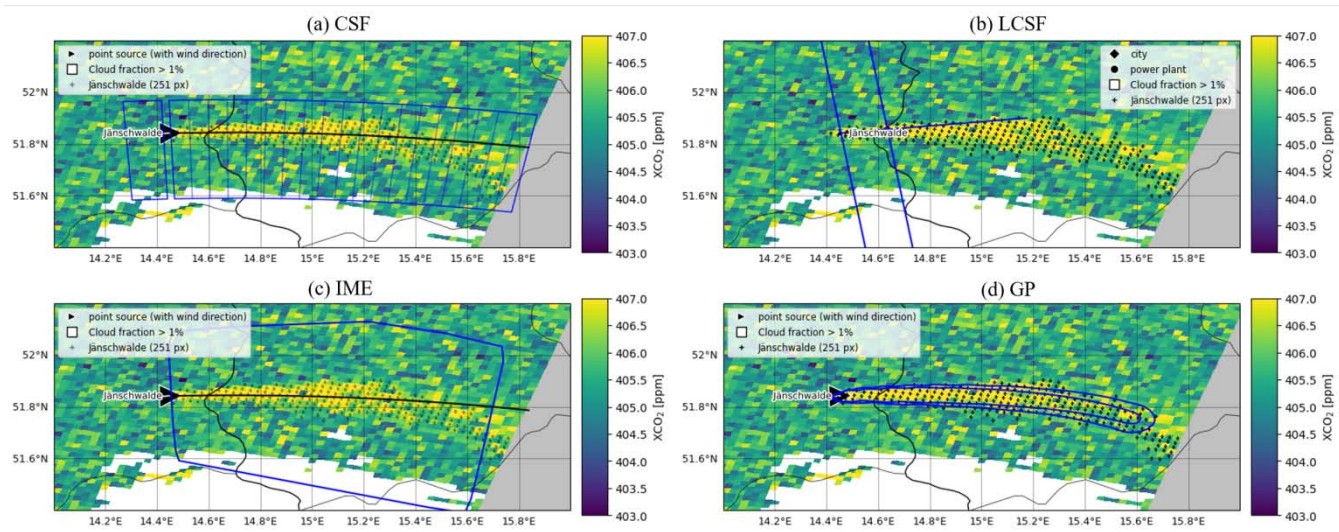

**Figure 1. Illustration of different inversion methods for a plume produced by the Jänschwalde power plant on April 23[rd], 2015. For all figures, pixels with dots are the selected enhancements representing the plume a) CSF method: the blue boxes depict the areas where the Gaussian fits of the plume cross-sections are made and the black line the centre-line of the plume. b) LCSF method: the blue lines represent the domain where the Gaussian fits of the plume cross-sections are made and the black line the along-wind direction at the source. c) IME method: the blue curve represents the domain on which mass enhancements are integrated. d) GP method: Blue curves depict contour lines of the 2-dimensional Gaussian curve that fits the plume.**

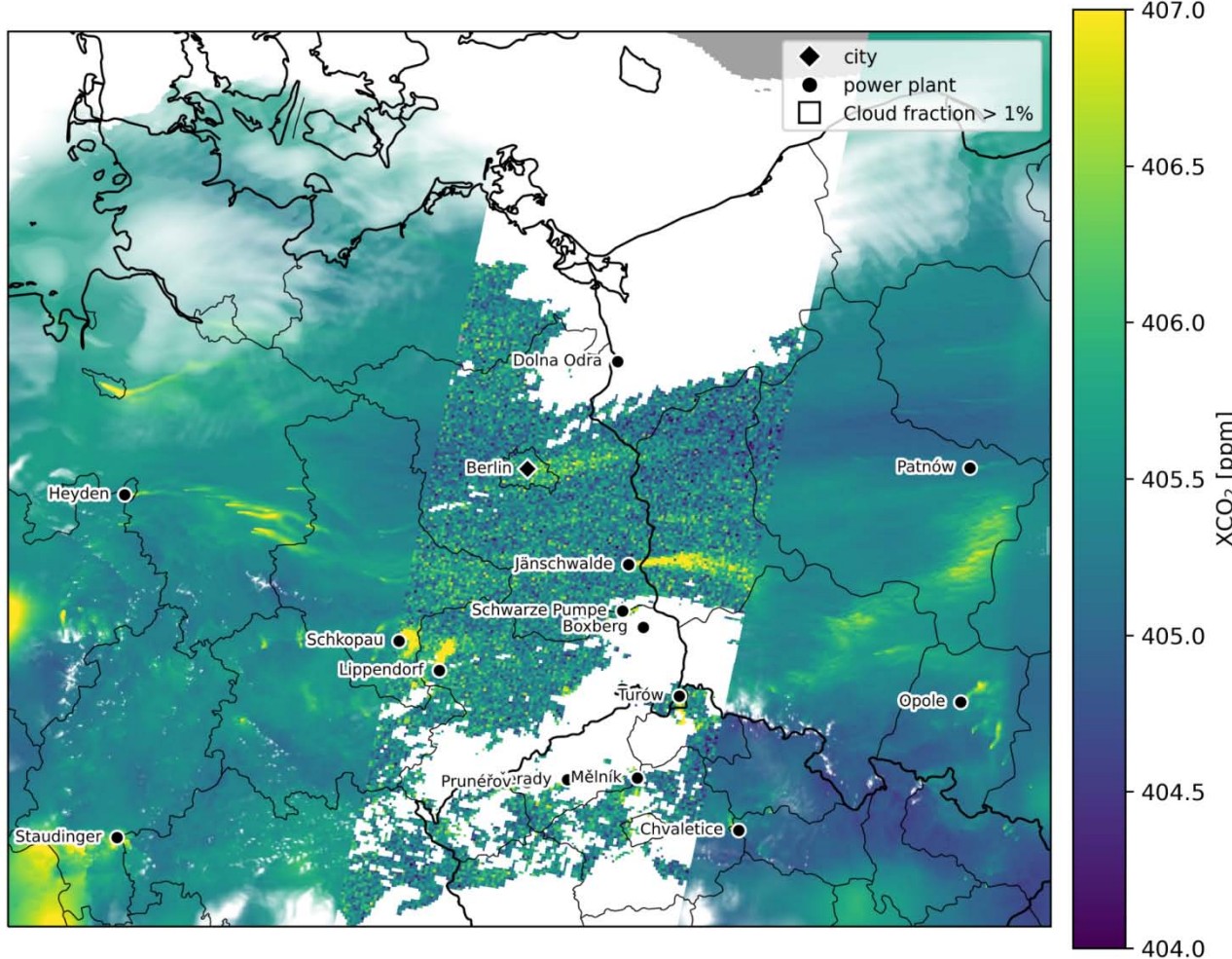


Figure 2. Simulations of $XCO_2$ on 23 April 2015 over the SMARTCARB domain. Synthetic $XCO_2$ observations over a 250 km wide
swath are represented in the centre of the figure for a low noise scenario. Missing $XCO_2$ observations due to a cloud fraction larger
than 1 % are shown in white. The 16 emission sources considered in this study are highlighted along with their names



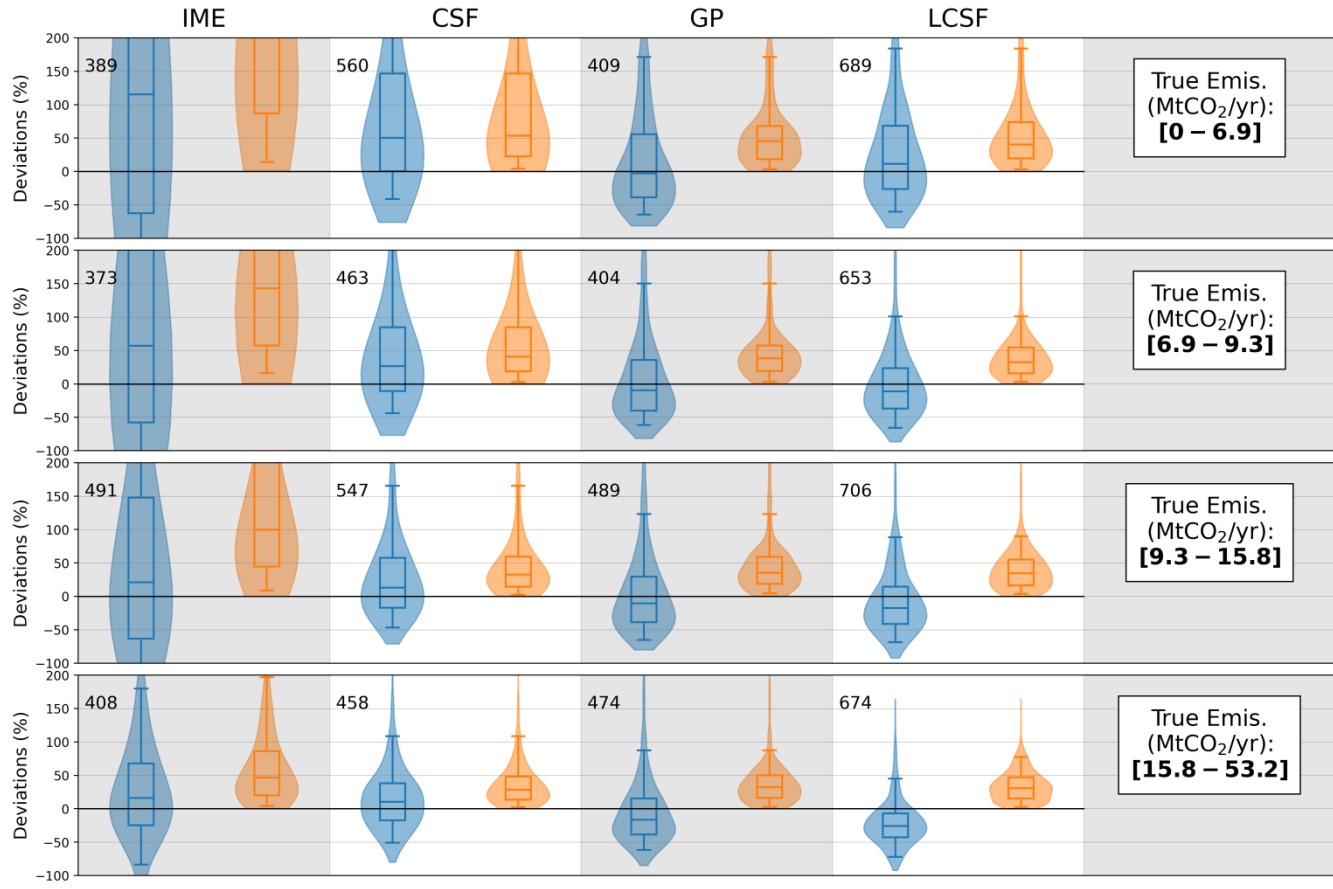

Figure 3. Performance when estimating $CO_2$ emissions from individual images of the different single-image inversion methods (columns) across different ranges of true emissions (rows) using SMARTCARB winds and cloud-free $CO_2$ and $NO_2$ data. The distributions of relative deviations (in blue) and relative absolute deviations (in orange) are illustrated using violin plots. The inter-quartiles are represented by the boxes, while the whiskers indicate the 5th and 95th percentiles, and medians are the lines inside the boxes. The numbers alongside boxes show the numbers of estimates corresponding to true emissions ranges and inversion methods.

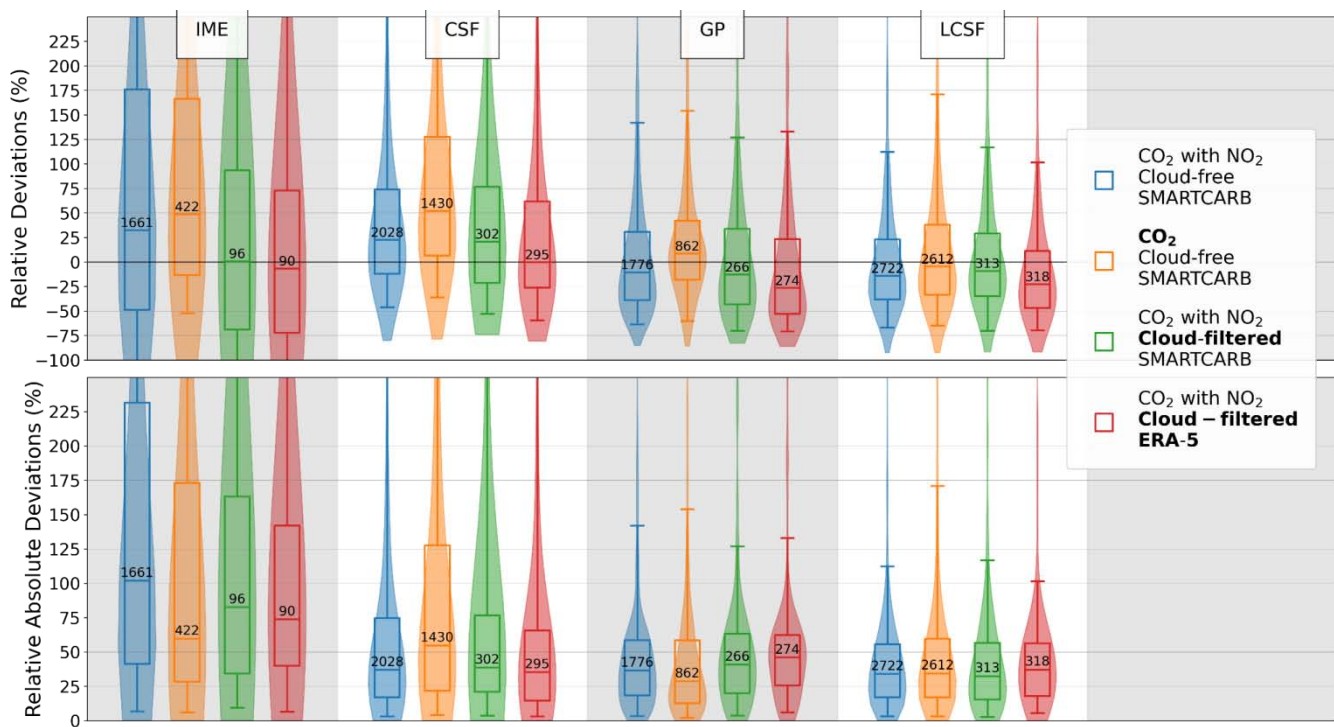


**Figure 4. Performances of the inversion methods when estimating emissions from single images for different benchmarking**
**scenarios: cloud-free $CO_2$ and $NO_2$ data with SMARTCARB winds (in blue), cloud-free $CO_2$ data only with SMARTCARB winds**
**(in orange), cloud-filtered $CO_2$ and $NO_2$ data with SMARTCARB winds (in green), cloud-filtered $CO_2$ and $NO_2$ data with ERA5**
**winds (in red). Bold texts in the legend indicate the elements of benchmarking scenarios that differ from those in the ideal**
**benchmarking scenario. Distributions of the relative deviations (top panel) and relative absolute deviations (bottom panel) are**
**illustrated using violin plots. Boxes are the inter-quartiles of the distributions, the whiskers are the 5th and 95th percentiles, and the**
**lines within boxes are the medians. Numbers in the inter-quartile boxes are the number of estimates for each benchmarking**
**scenario and inversion method.**

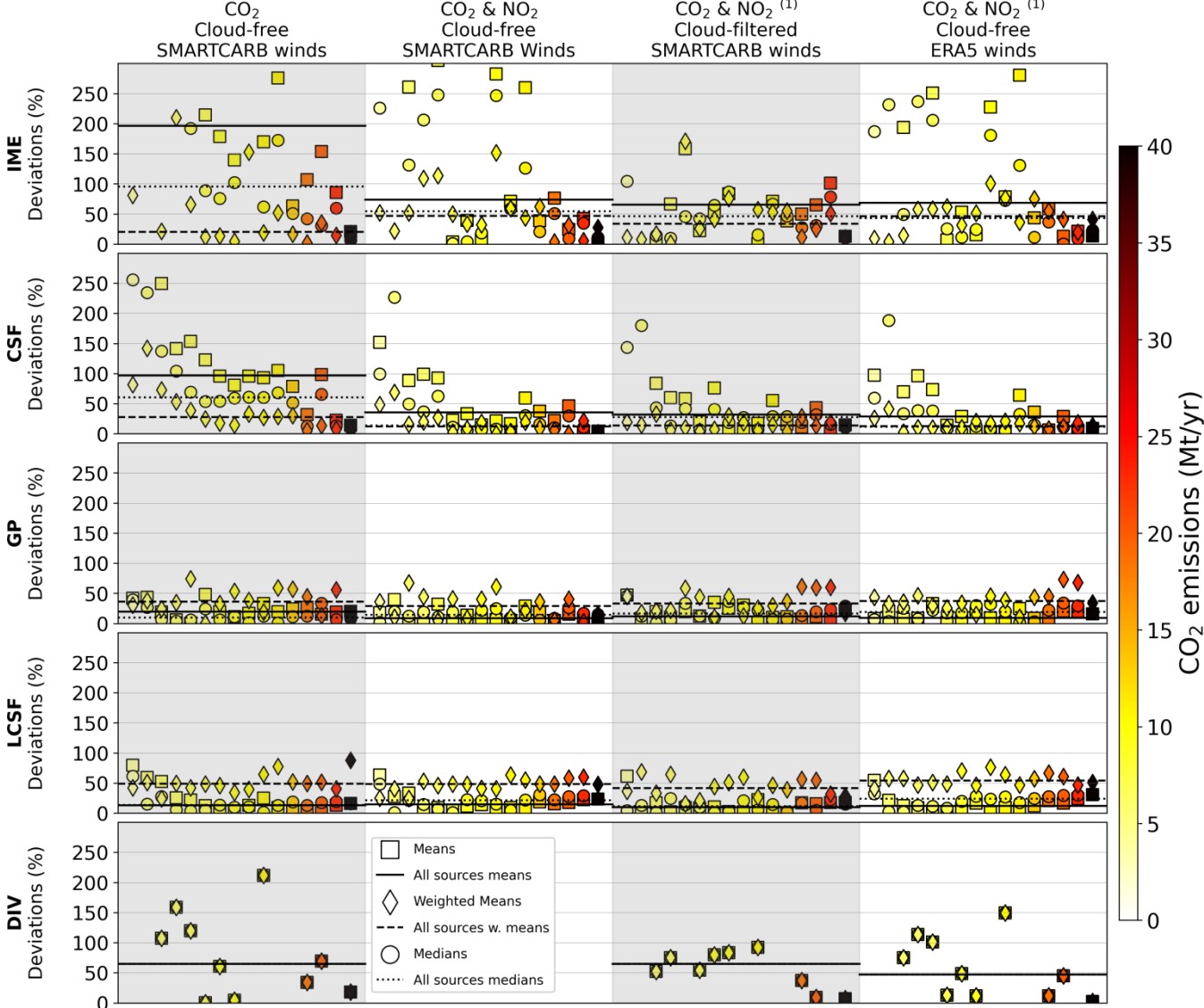


Figure 5. Performance of the inversion methods for annual estimates of $CO_2$ emissions. The markers represent for a given source the relative absolute deviations from the true annual emissions of the arithmetic means (squares), the weighted means (diamonds) and the medians (circles) of the estimates over a year. The lines represent the median values of the annual estimates over the entire set of sources. The inversions are performed using $CO_2$ cloud-free data and SMARTCARB winds (1st column), using $CO_2$ and $NO_2$ cloud-free data and with SMARTCARB winds (2nd column), using $CO_2$ and $NO_2$ cloud-filtered data and SMARTCARB winds (3rd column), and using $CO_2$ and $NO_2$ cloud-free data and with ERA5 winds (4th column). (1) For the Divergence methods, the inversions of the 3rd and 4th columns are performed using $CO_2$ data only. Markers color indicates the true $CO_2$ annual emissions of the corresponding source.

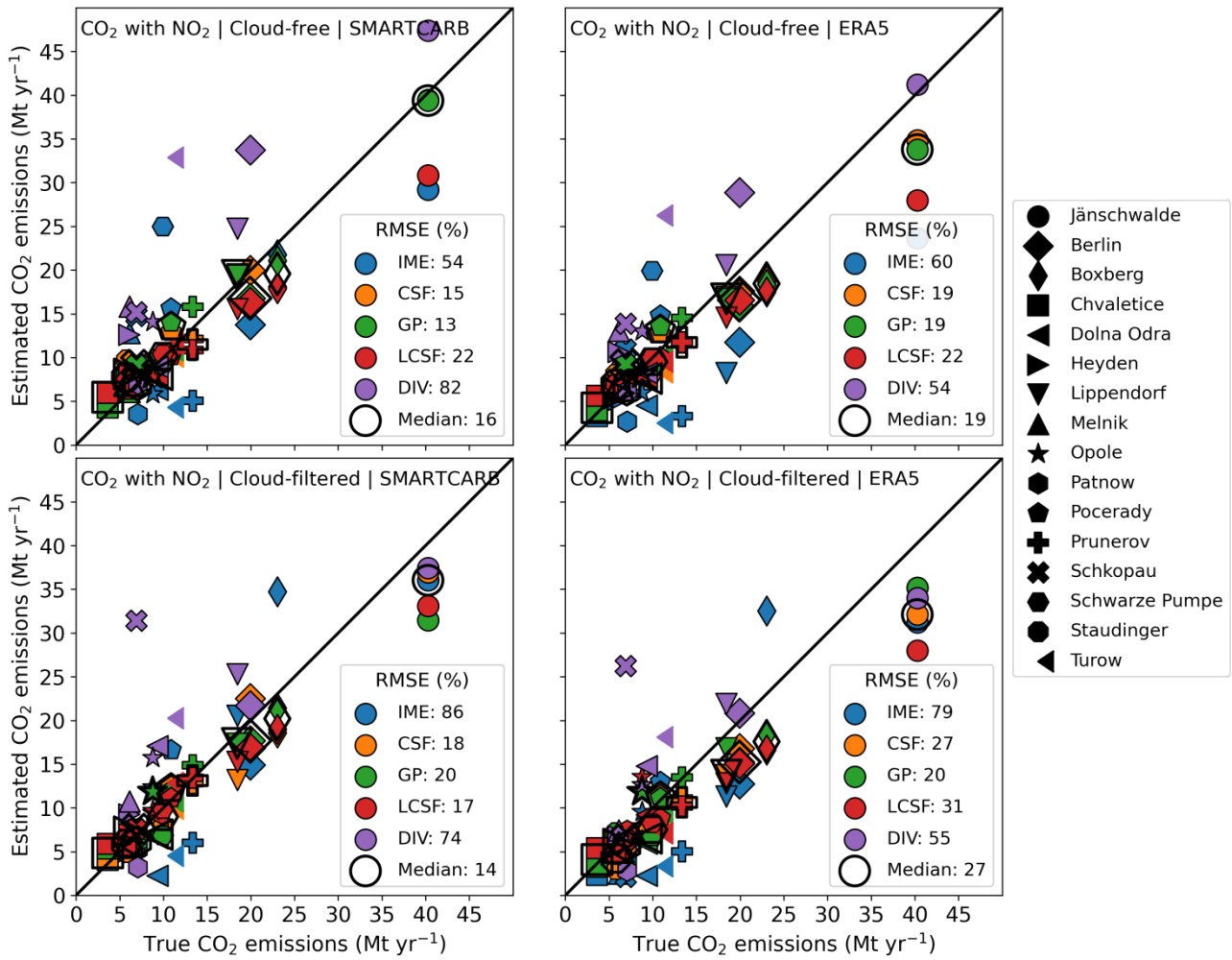

Figure 6. Estimated vs true annual emissions for 4 inversion scenarios (titles of the panels). For the IME and CSF methods, annual estimates are weighted means of the single-image estimates while they are arithmetic means for the GP, LCSF and Divs methods. Each marker represents a given emission source and each color a given inversion method. The unfilled markers represent the median values of all the estimates for each source. The divergence inversion method uses $CO_2$ data for all the inversion scenarios. The plain line represents the 1:1 line. The bottom-right legends display for each inversion method the relative RMSE which is the RMSE between estimated and true annual emissions divided by the median of true annual $CO_2$ emissions of all sources (~9.6 Mt $yr^{-1}$).

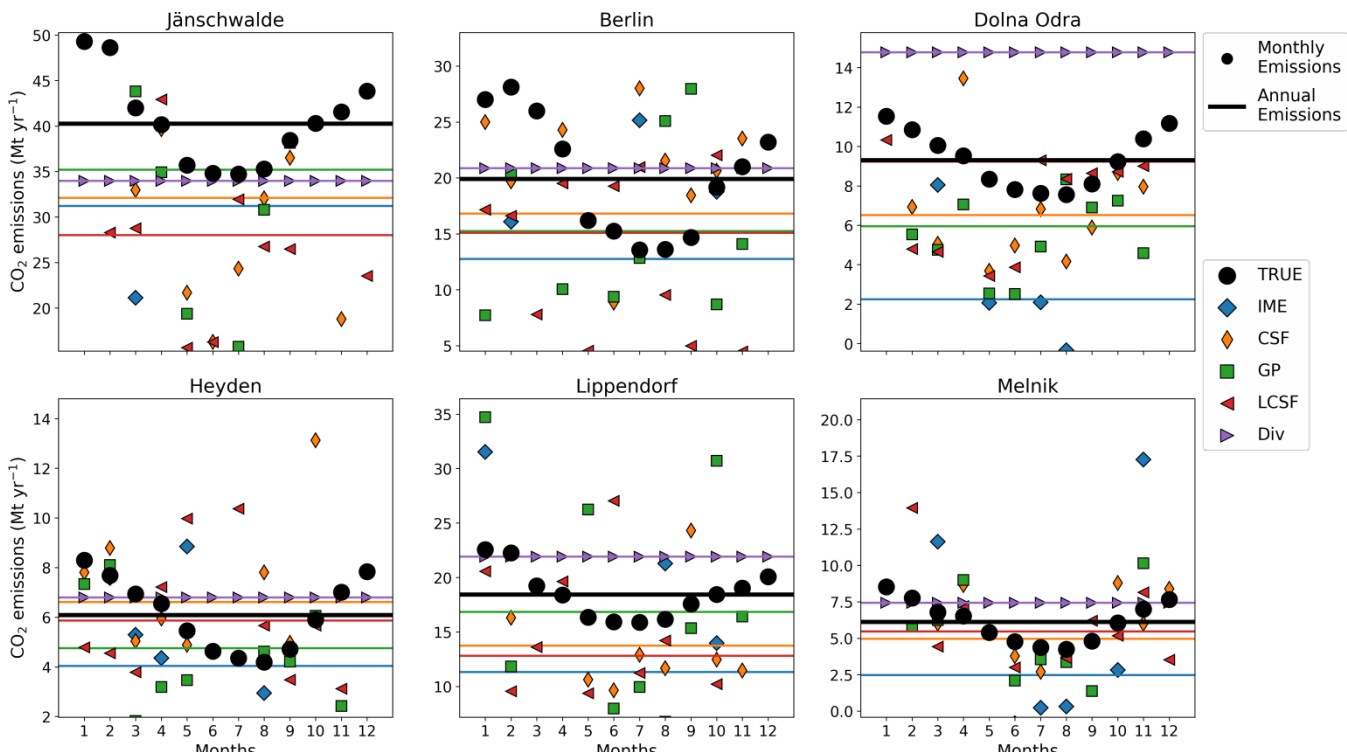

**Figure 7. Annual and monthly estimates of the true and estimated emissions for different sources and for different inversion methods. Each panel is associated with a given source. Plain lines and markers represent annual averages and monthly averages respectively. Colors and markers are associated with different inversion methods (true emissions are represented by black circles). Annual and monthly estimates for the IME and CSF methods are weighted means of image estimates. Annual and monthly estimates for the GP and LCSF are means of image estimates while for the divergence method, we use the annual estimate also for monthly estimates. All inversion methods use $CO_2$ and $NO_2$ cloud-filtered data ($CO_2$ data only for the Div method) with ERA5 winds.**

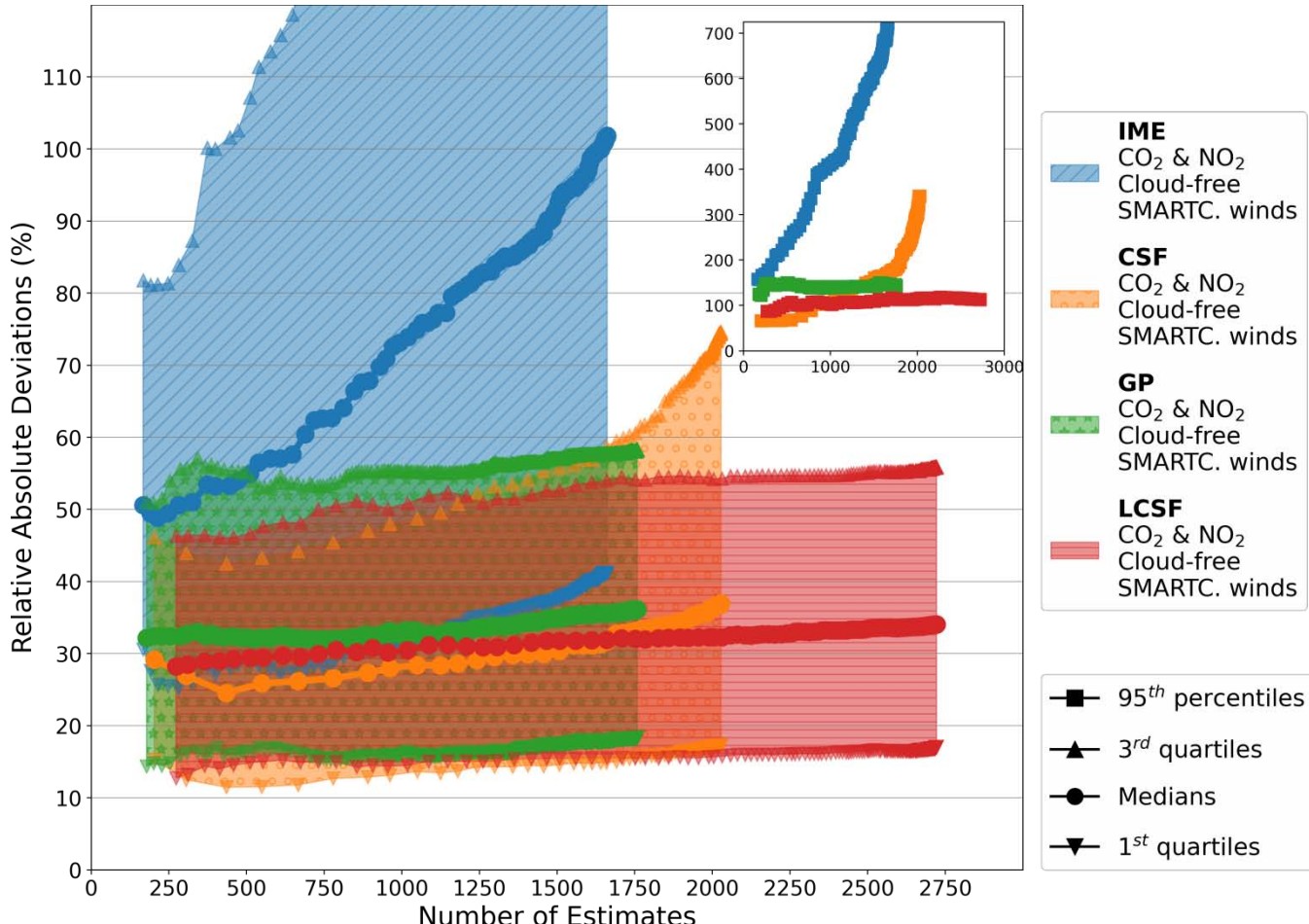

1275

**Figure 8. Accuracy of inversions *vs* number of single-image estimates. The inversion methods shown here use CO$_2$ and NO$_2$ cloud-free data and SMARTCARB winds. The filled areas represent the inter-quartiles of the distributions of the relative absolute deviations depending on the number of estimates. The 95$^{th}$ percentiles of the distributions are represented in the inset. Points belonging to a same curve are associated to different QIs and from left to right along curves, points are associated with a decreasing QI; the points at the left and right ends of the curves are associated with the maximal and minimal QIs respectively.**

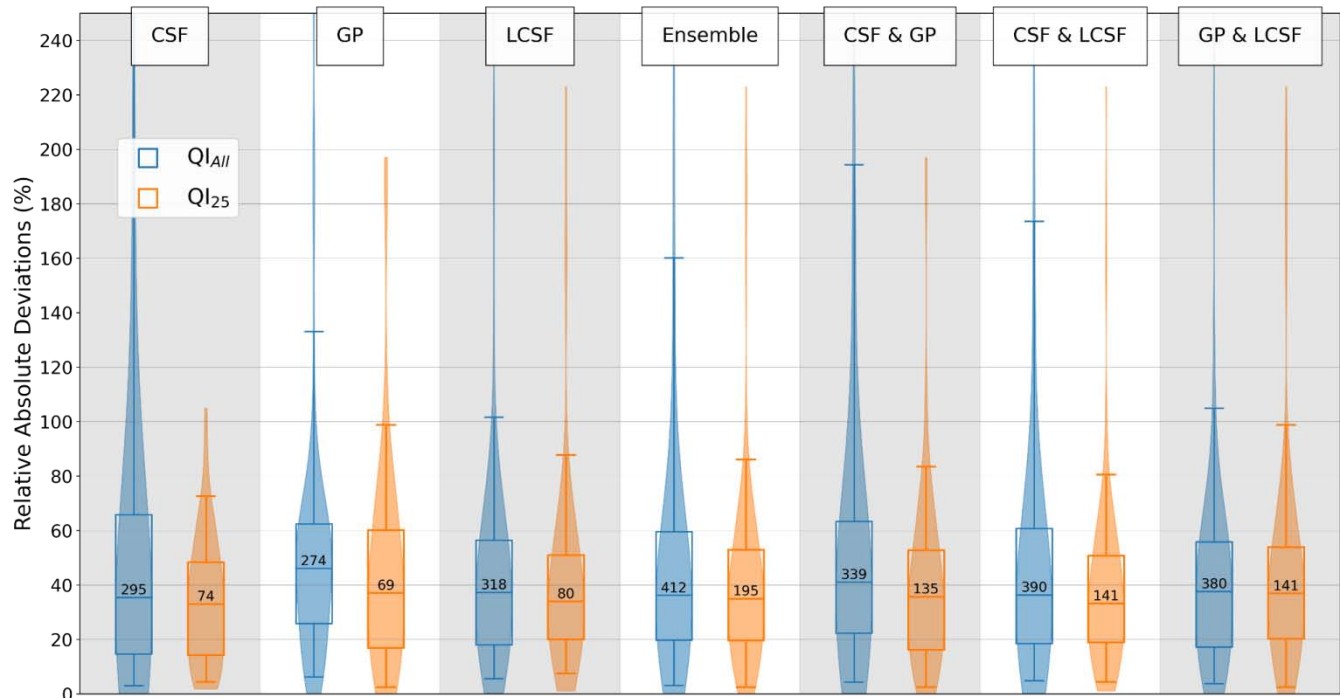

Figure 9: Performance of the inversion methods and ensemble approaches for estimating the emissions with cloud-filtered $CO_2$ & $NO_2$ data and with ERA5 winds. The distributions of the relative absolute deviations for all the inversion results (in blue) and for the best estimates (in orange) provided by each method (see text) are illustrated using violin plots. Boxes represent the inter-quartiles of the distributions, the whiskers the 5th and 95th percentiles, and the lines within boxes the medians. Numbers in the inter-quartile boxes are the number of estimates for each benchmarking scenario and inversion method.

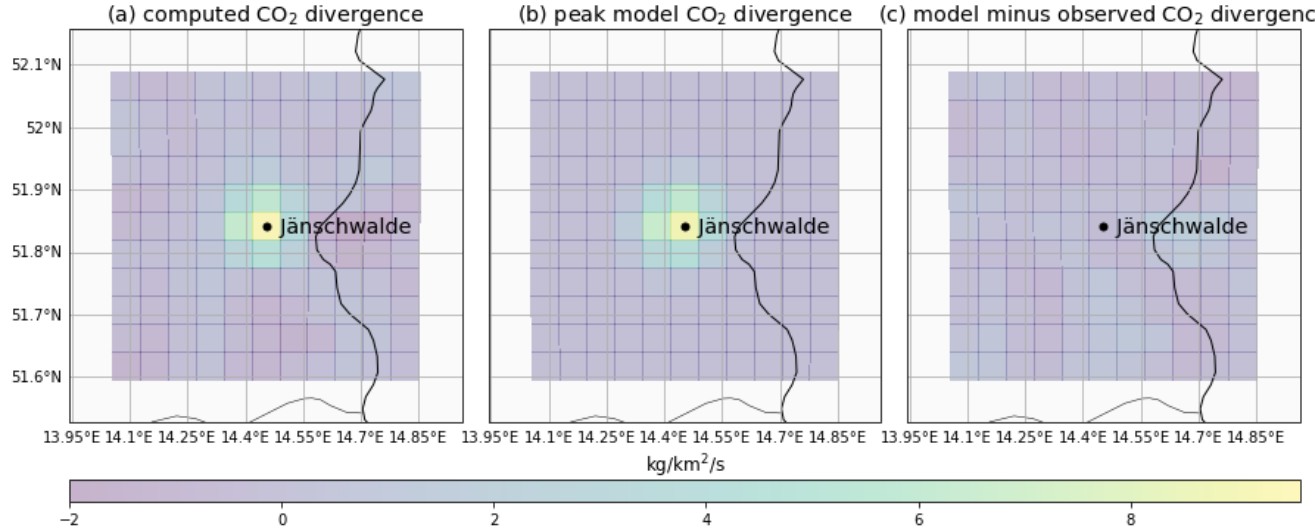

Figure A1: Illustration of the divergence method for the Jänschwalde power station in 2015 based on the synthetic SMARTCARB dataset (see text). The figures represent the annual fields of the computed $CO_2$ divergence (a), the modelled $CO_2$ divergence (b) and the difference of both quantities (c). Sink terms are considered negligible for $CO_2$, divergence fields are considered equal to the emission fields for $CO_2$.

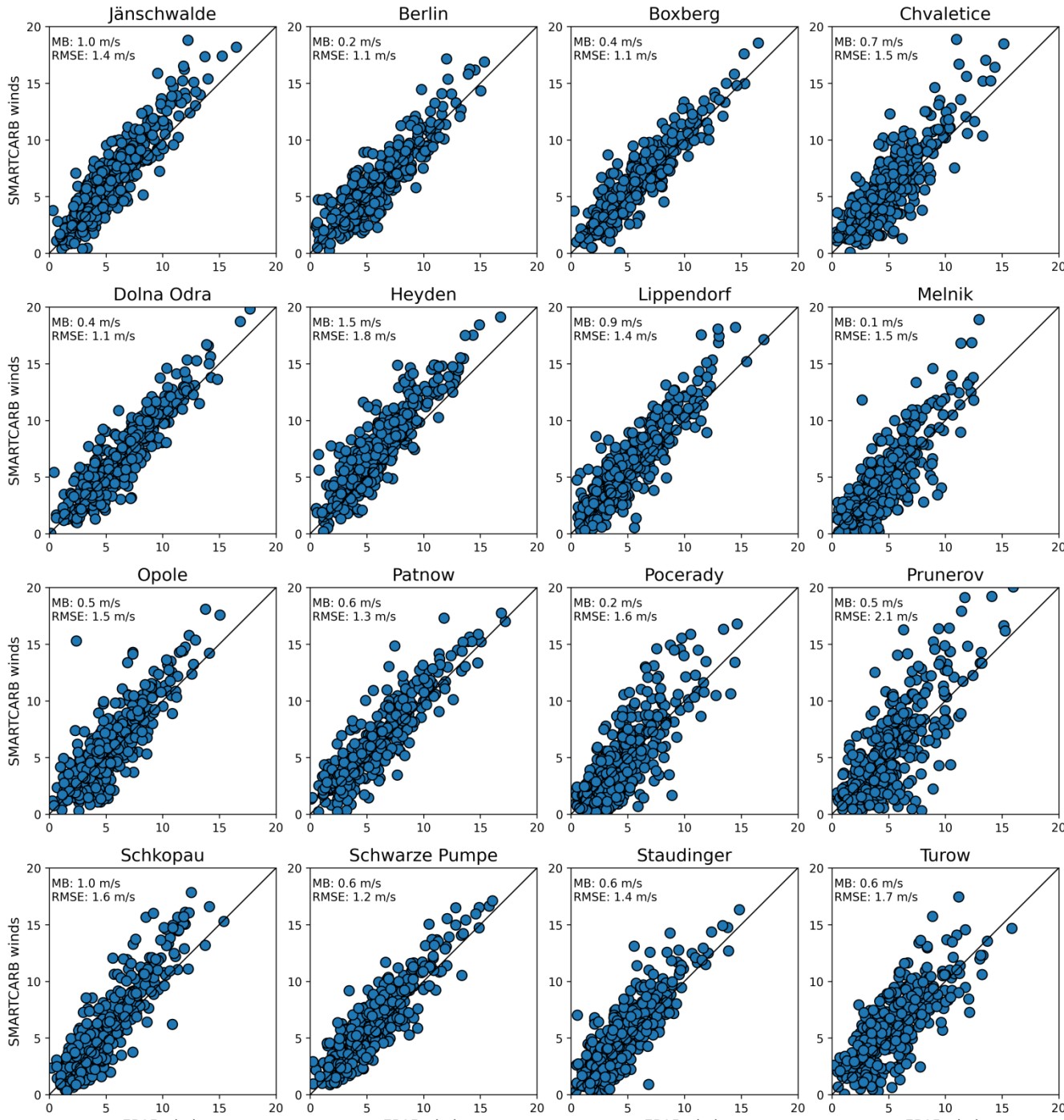

Figure A2: Norms of the ERA5 winds vs norms of the SMARTCARB winds at the sources considered in this study and for all the days of 2015. Black lines represent the 1:1 agreement line. Mean biases of the SMARTCARB norms minus the ERA5 norms and RMSEs are noted at the top left of the figures.

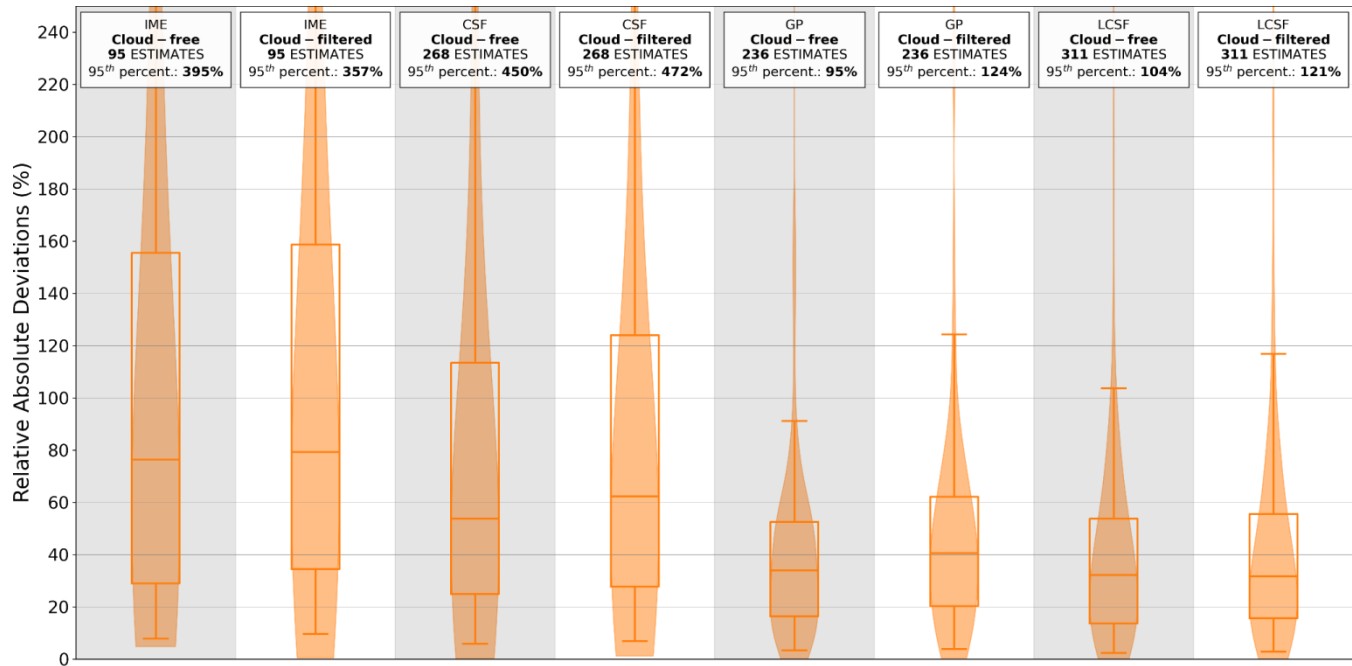


**Figure A3: Performance of the inversion methods when using data with or without clouds for the emissions estimated from the**
**same images. The inversion methods use $CO_2$ and $NO_2$ data and SMARTCARB winds. The boxes represent the inter-quartiles of**
**the distributions of the absolute relative deviations, the whiskers the $5^{th}$ and $95^{th}$ percentiles, and the lines within boxes the**
**medians.**

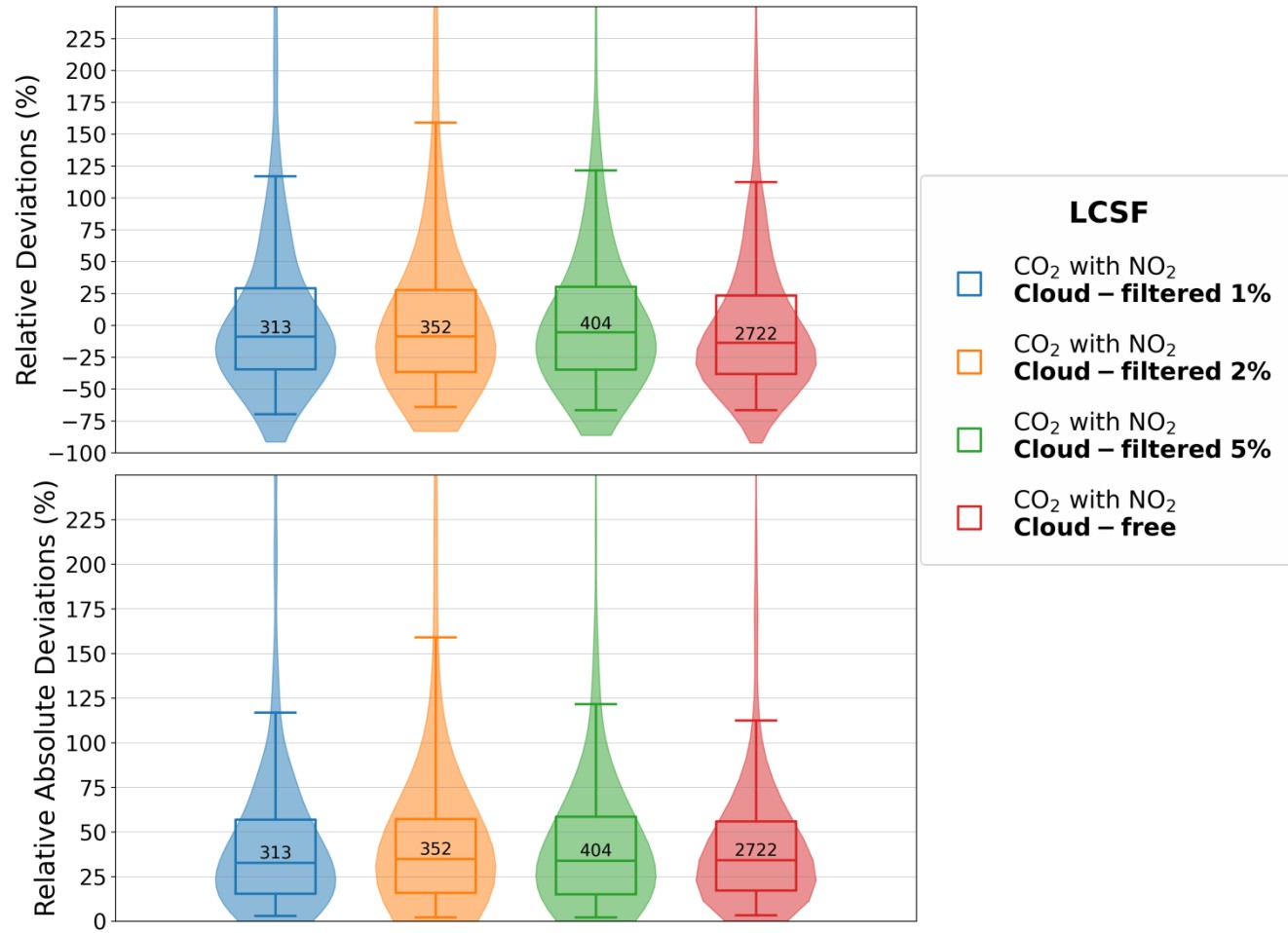

Figure A4: Performance of the LCSF method when estimating emissions from single images of $CO_2$ and $NO_2$ without considering clouds (in red) and for different cloudiness thresholds: 1 % (in blue), 2 % (in orange) and 5 % (in green). Distributions of the relative deviations (top panel) and relative absolute deviations (bottom panel) are illustrated using violin plots. Boxes are the inter-quartiles of the distributions, the whiskers are the 5[th] and 95[th] percentiles, and the lines within boxes are the medians. Numbers in the inter-quartile boxes are the number of estimates for each benchmarking scenario.

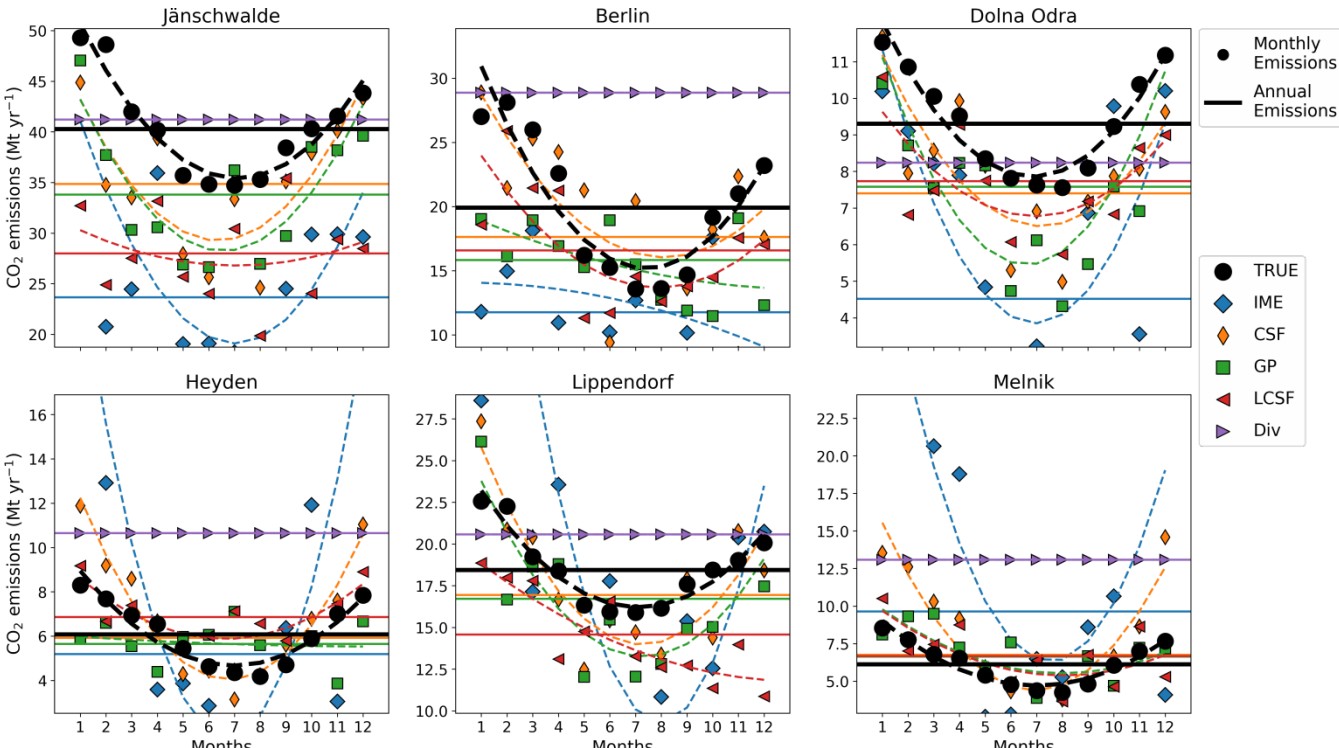

Figure A5: Annual and monthly estimates of the true and estimated emissions for different sources and for different inversion methods. Each panel is associated with a given source. Plain lines and markers represent annual averages and monthly averages respectively. Dashed lines represent the fits by a 2$^{nd}$ order polynomial of the monthly estimates. Colours are associated with different inversion methods (true emissions are in black). Annual and monthly estimates for the IME and CSF methods are weighted means of image estimates. Annual and monthly estimates for the GP and LCSF are means of image estimates while for the divergence method, we use the annual estimate also for monthly estimates. All inversion methods use $CO_2$ and $NO_2$ cloud-free data ($CO_2$ data only for the Divs methods) with ERA5 winds.

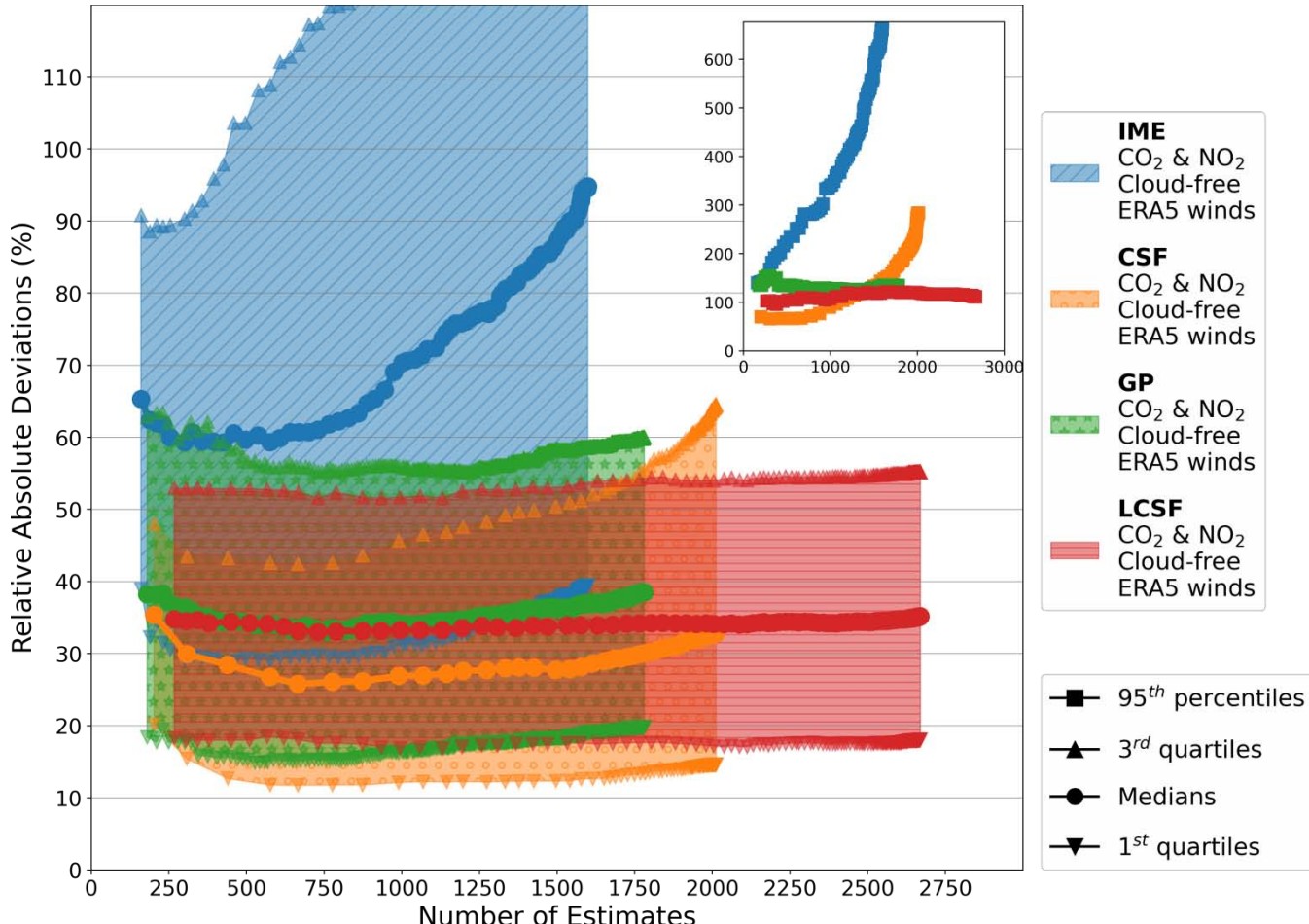

1319

**Figure A6. Accuracy of inversions *vs* number of instant estimates. The inversion methods shown here use $CO_2$ and $NO_2$ cloud-free**
**data and ERA5 winds. The filled areas represent the inter-quartiles of the distributions of the relative absolute deviations**
**depending on the number of estimates. The 95th percentiles of the distributions are represented in the inset. Points belonging to a**
**same curve are associated to different QIs and from left to right along curves, points are associated with a decreasing QI; the**
**points at the left and right ends of the curves are associated with the maximal and minimal QIs respectively.**

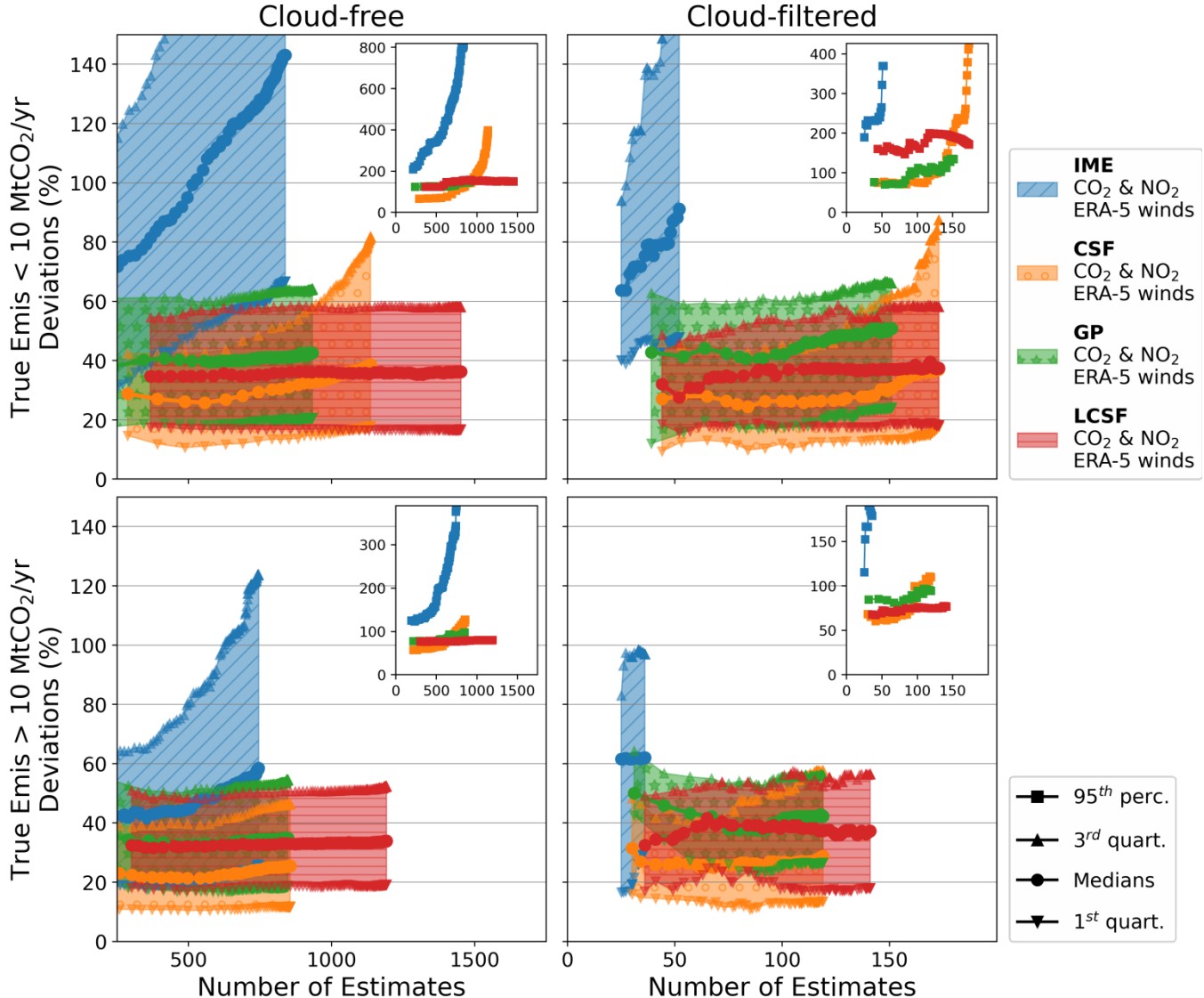

Figure A7: Accuracy of inversions vs number of instant estimates. The inversion methods shown here use $CO_2$ and $NO_2$ data, ERA5 winds and for cloud-free (1st column) and cloud-filtered data (2nd column). Results are shown for the cases where true $CO_2$ emissions of sources are below (1st row) and above (2nd row) 10 Mt yr$^{-1}$. The filled areas represent the inter-quartiles of the distributions of the relative absolute deviations depending on the number of estimates. The 95th percentiles of the distributions are represented in the insets. Each point belonging to a same curve is associated with a different QI and from left to right along a same curve; points are associated with a decreasing QI.

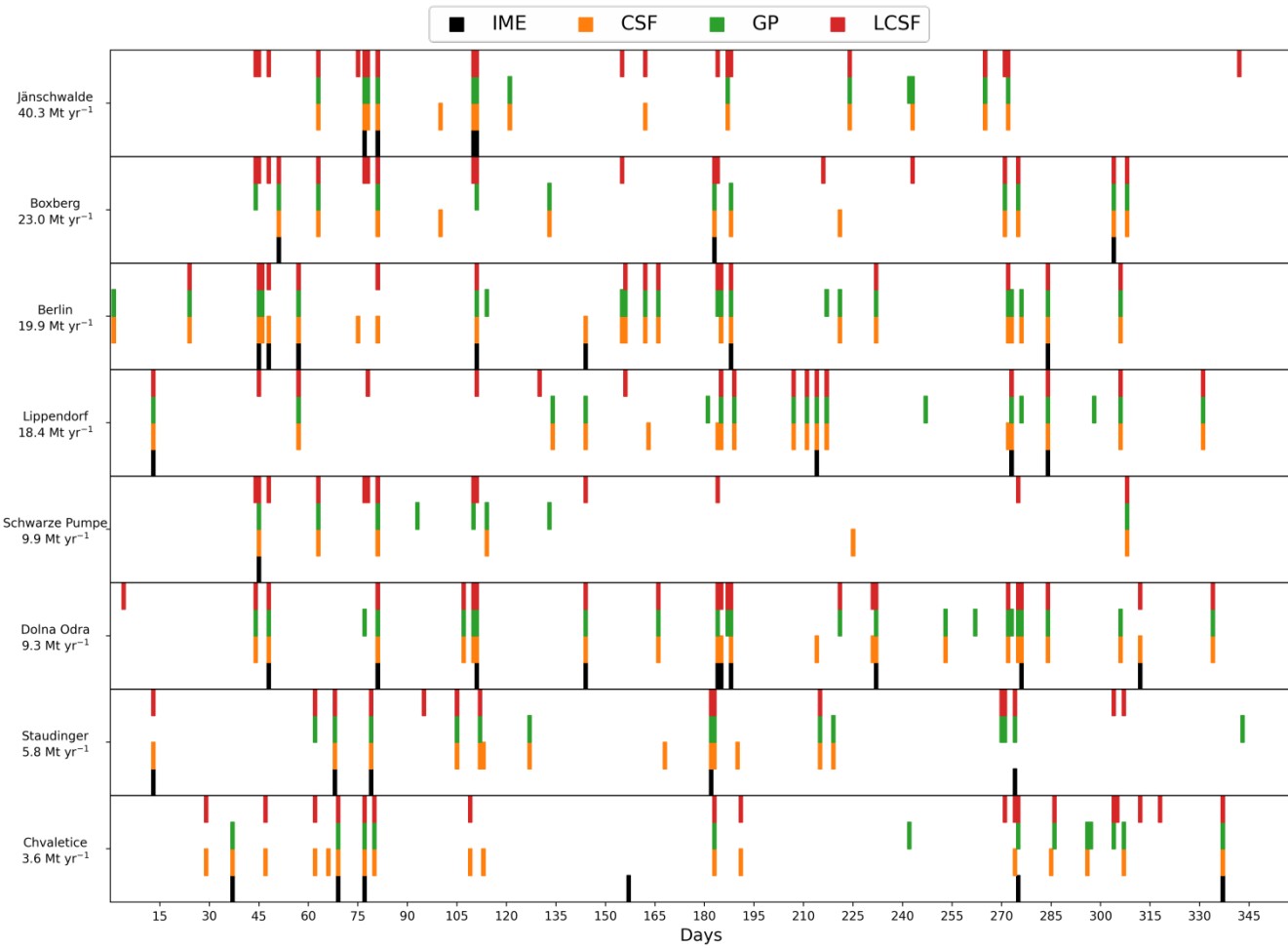

Figure A8: Days of 2015 (x-axis) for which the IME, CSF, GP and LCSF methods produce estimates for the $CO_2$ emissions of eight sources (y-axis). For a given day, the availability of an estimate from a given inversion method is illustrated by a color bar (for color explanation, see legend of the figure). Inversions use $CO_2$ and $NO_2$ cloud-filtered data and ERA5 winds.

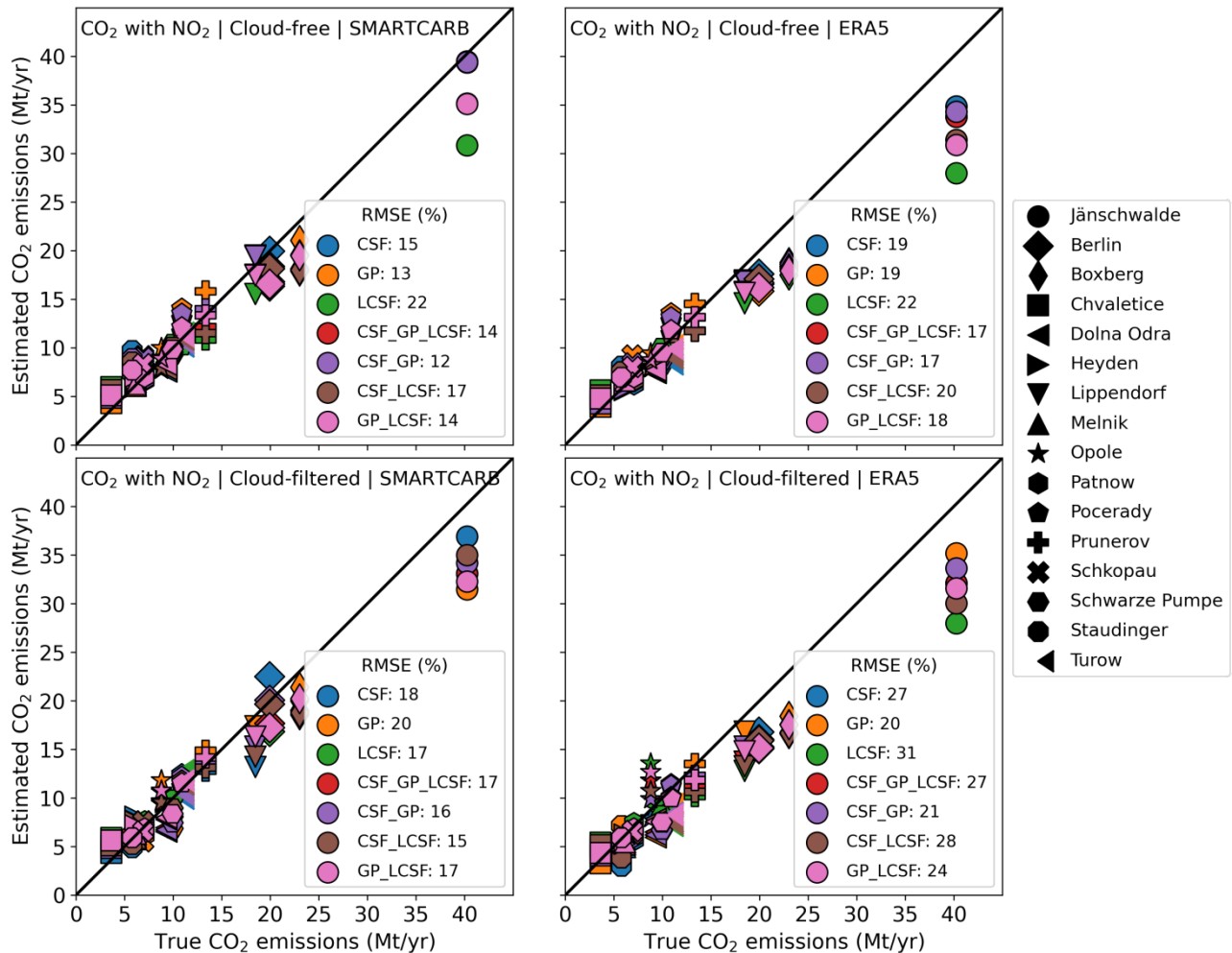

Figure A9: Estimated vs true annual emissions for 4 inversion scenarios (titles of the panels). Results are displayed for the CSF, GP, LCSF and ensemble methods that gather 2 or 3 of these individual methods. For the CSF method, annual estimates are weighted means of the instant estimates while they are arithmetic means for the GP and LCSF methods. Each marker represents a given emission source and each color a given inversion method. The divergence inversion method uses $CO_2$ data only for all the inversion scenarios. The plain line represents the 1:1 line. The bottom-right legends display for each inversion method the relative RMSE which is the RMSE between estimated and true annual emissions divided by the median of true annual emissions of all sources (~9.6 $MtCO_2$/yr).

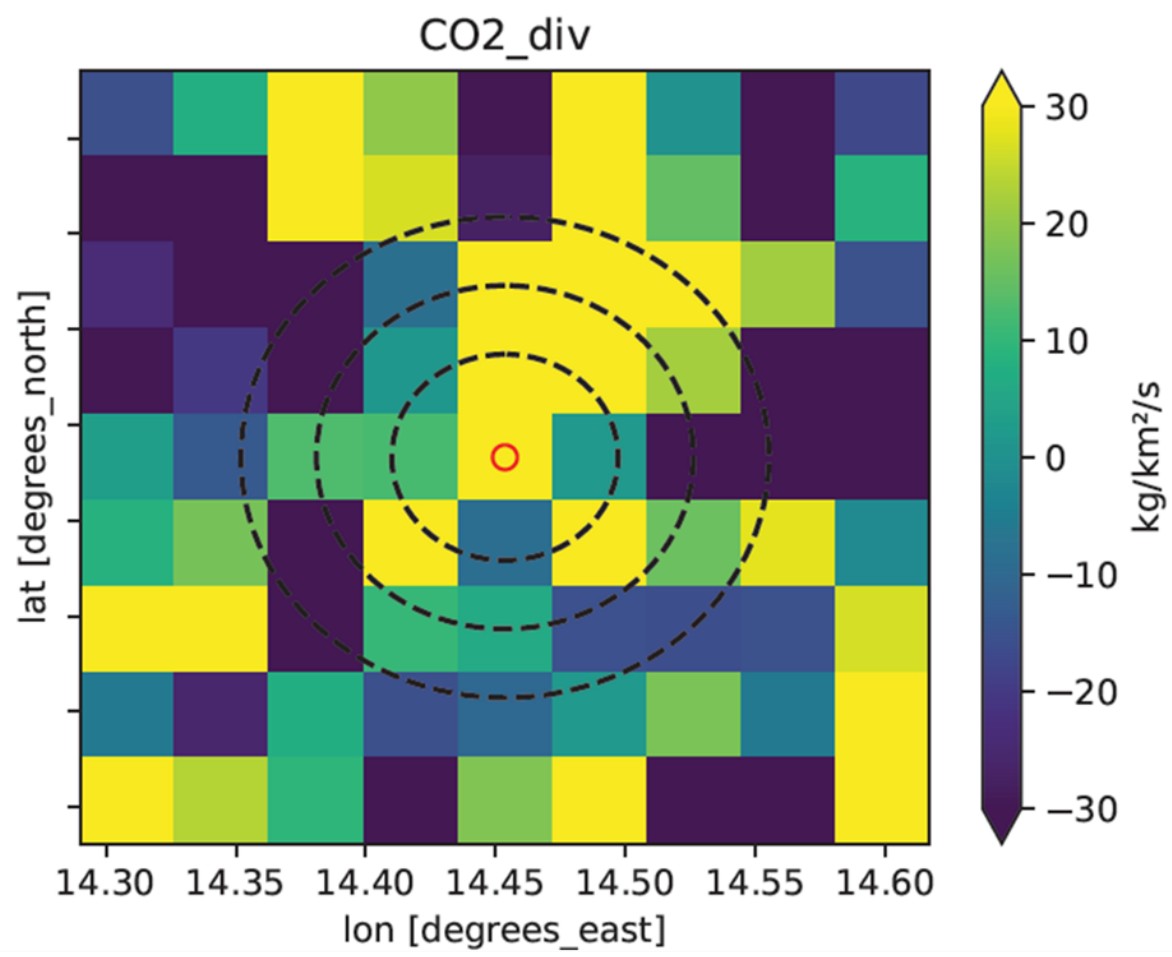

1348

**Figure A10: Divergence map estimated around the Jänschwalde power station on January 2015 the 12th. Dotted circles show different radii (3 km, 5 km and 7 km) which define integration disks that could be used by the integral divergence method.**

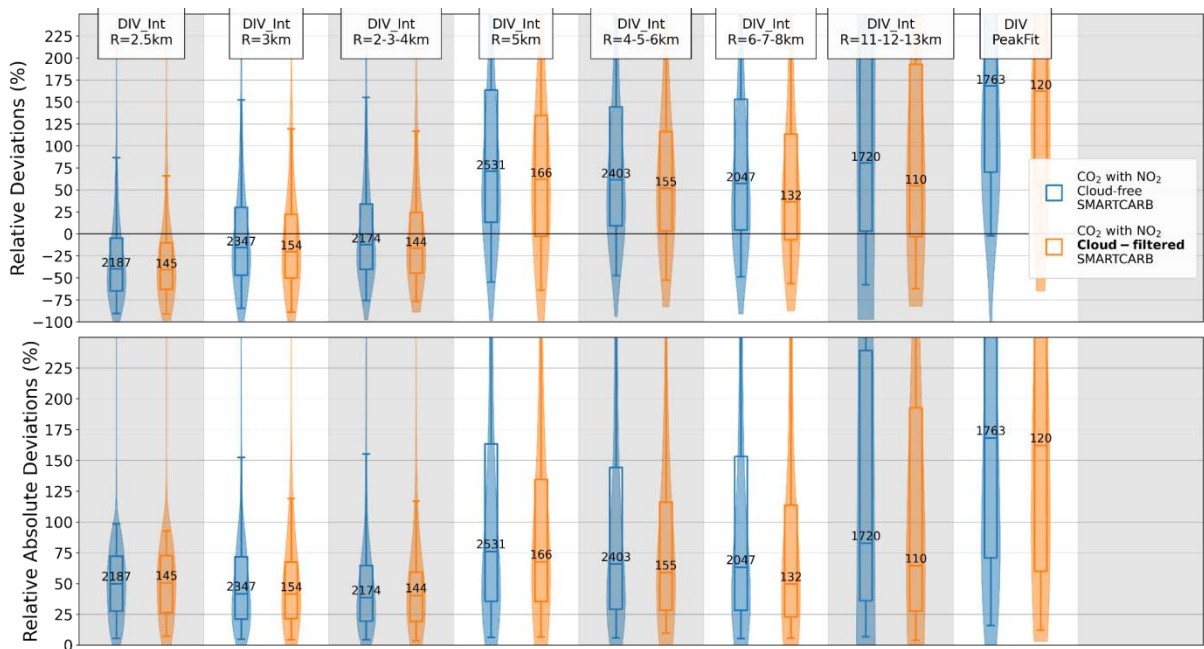

1351

**Figure A11: Performances of the different versions of the divergence inversion method when estimating emissions from one year of single images for different benchmarking scenarios: cloud-free $CO_2$ and $NO_2$ data with SMARTCARB winds (in blue) and cloud-filtered $CO_2$ and $NO_2$ data with SMARTCARB winds (in orange). Distributions of the relative deviations (top panel) and relative absolute deviations (bottom panel) are illustrated using violin plots. Boxes are the inter-quartiles of the distributions, the whiskers are the 5[th] and 95[th] percentiles, and the lines within boxes are the medians. Numbers in the inter-quartile boxes are the number of estimates for each benchmarking scenario and inversion method. Methods DIV_int_R=xkm and DIV_PeakFit are the integral (for an integration radius of x km) and peak-fitting versions of the divergence approach respectively. For a given overpass and source, the emission estimate of the method DIV_int_R=x-y-zkm is the average of the estimates when integrating over circles of x, y and z km radius around the source.**

1361

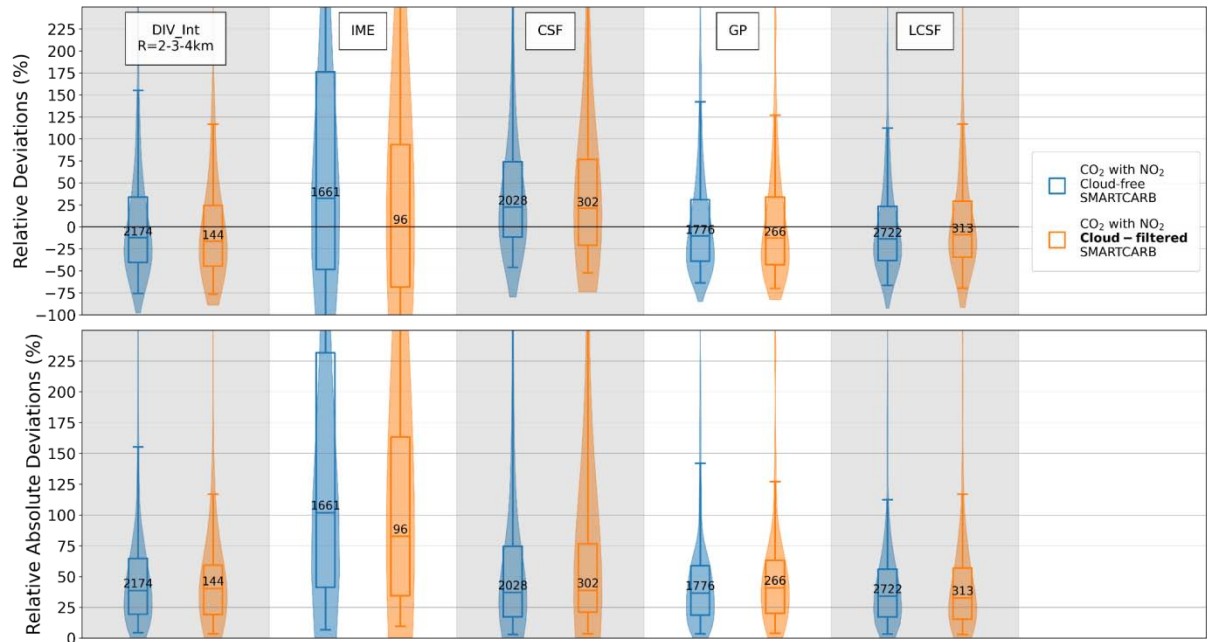

**Figure A12: Performances of the inversion methods when estimating emissions from one year of single images for different benchmarking scenarios: cloud-free CO$_2$ and NO$_2$ data with SMARTCARB winds (in blue) and cloud-filtered CO$_2$ and NO$_2$ data with SMARTCARB winds (in orange). Distributions of the relative deviations (top panel) and relative absolute deviations (bottom panel) are illustrated using violin plots. Boxes are the inter-quartiles of the distributions, the whiskers are the 5$^{th}$ and 95$^{th}$ percentiles, and the lines within boxes are the medians. Numbers in the inter-quartile boxes are the number of estimates for each benchmarking scenario and inversion method. Methods DIV_int_R=2-3-4km and DIV_PeakFit are the integral and peak-fitting versions of the divergence approach respectively. For a given overpass and source, the emission estimate of the method DIV_int_R=2-3-4km is the average of the estimates when integrating over circles of 2,3 and 4 km radius around the source.**

| Method | Time frame | Computational cost (1) |
| --- | --- | --- |
| Integrated Mass Enhancement (IME) | Single-Image estimates | Medium: ~20 min |
| Cross-Sectional Flux (CSF) | Single-Image estimates | Medium: ~25 min |

| | | |
|---|---|---|
| Gaussian Plume (GP) | Single-Image estimates | High: ~110 min |
| Light Cross-Sectional Flux (LCSF) | Single-Image estimates | Low: ~10 min |
| Divergence (Div) | Averaged estimates from ensemble of images | Medium: ~23 min |

**Table 1: Summary of characteristics of the benchmarked methods. (1) Computation time was estimated by inverting one month of $CO_2$ and $NO_2$ cloud-free SMARTCARB data on the same server using the ddeq package (Kuhlmann et al., 2023)**

| Benchmark Scenario | Wind dataset | Cloud fraction thresholds | Joint use of $NO_2$ and $CO_2$ |
|---|---|---|---|
| Scenario 1 | SMARTCARB | 100 % (no clouds) | Yes |
| Scenario 2 | SMARTCARB | 1 % for $CO_2$, 30 % for $NO_2$ | No |
| Scenario 3 | SMARTCARB | 100 % (no clouds) | No |
| Scenario 4 | SMARTCARB | 1 % for $CO_2$, 30 % for $NO_2$ | Yes |
| Scenario 5 | ERA5 | 100 % (no clouds) | Yes |
| Scenario 6 | ERA5 | 1 % for $CO_2$, 30 % for $NO_2$ | No |
| Scenario 7 | ERA5 | 100 % (no clouds) | No |
| Scenario 8 | ERA5 | 1 % for $CO_2$, 30 % for $NO_2$ | Yes |

**Table 2: List of the different benchmarking scenarios: from the most optimistic (scenario 1) which considers inversions with cloud-free data and SMARTCARB winds to the most realistic (Scenario 8) with cloud-filtered data and with ERA5 winds. Note that a cloud fraction threshold of x % corresponds to the rejection of data pixels if their cloud cover exceeds x %, so that a cloud fraction of 100 % yields full images without a loss of data pixels.**

| Inversion method | Cloud-free data | Cloud-filtered data |
|---|---|---|
| IME | 1661 | 96 |
| CSF | 2028 | 302 |
| GP | 1776 | 266 |
| LCSF | 2722 | 313 |

1383    **Table 3. Number of estimates for each inversion method when data with or without clouds are used. Inversions are**
1384    **performed with CO$_2$ and NO$_2$ data and, with SMARTCARB winds.**

1385