# Peer review of "Benchmarking data-driven inversion methods for the estimation of local CO2 emissions from XCO2 and NO2 synthetic satellite images"

_Atmospheric Measurement Techniques, 2023_

## Author Comment (AC1)

**Responses to Referee#1 comments:**

*General*                                                                         *remarks*

*The paper compares 5 data-driven methods to estimate local $CO_2$ emissions from satellite images as an extension of earlier studies (e.g. Kuhlmann et al. 2021). The images including clouds were generated using a regional atmospheric transport model and known emissions from power plants and a city with output sampled as for the satellite in construction. In general the paper is well written and contains a lot of details.*

We thank Referee#1 for this positive feedback, for his insightful comments and for thoroughly reading the manuscript, which allowed errors to be corrected.

*Concerning the cloud effects the language is sometimes not clear or too lengthy. It should be clearly said that in the 'cloud-free' scenario (section 2.3) modeled clouds are ignored and all pixels used for analysis, right? It might be interesting to include a remark what will happen for the accuracy of the methods if a cloudiness threshold of 2 or 5% is selected which might be more typical for real images.*

We agree with the referee/s remark and rephrased the sentence describing the cloud-free scenario and add a new paragraph in the text to describe the impact of the cloudiness threshold on the results:

**Old sentence:**
*The most optimistic or ideal scenario considers that inversions are performed with $CO_2$ and $NO_2$ cloud-free data using directly the winds from the COSMO-GHG simulations (SMARTCARB winds). It is the ideal case because 1) with the inclusion of $NO_2$ data, the data constraints on the estimates are stronger than when using $CO_2$ data only; 2) the absence of clouds maximizes the number and quality of the estimates, and 3) the winds are perfectly consistent with the data as they were used to simulate the $XCO_2$ and $NO_2$ fields.*

**New sentence:**
*The most optimistic or ideal scenario corresponds to the application of inversions to $CO_2$ and $NO_2$ images without the removal of pixels associated to cloud-cover (ignoring the clouds modelled with the COSMO-GHG model; we label such inversions "cloud-free" hereafter), and with a perfect knowledge of the wind field (i.e. using directly the winds from the COSMO-GHG model, denoted SMARTCARB winds). It is the ideal case because 1) the joint analysis of $NO_2$ and $CO_2$ images strengthen the estimates compared to the analysis of $CO_2$ images only; 2) ignoring the potential loss of data due to cloud cover in the $CO_2$ and $NO_2$ images yield full images, whose analysis is more robust than that of partial images, and thus provides a higher number and precision of estimates.*

*Concerning the impact on the performance of the methods of increasing the cloudiness threshold to 2% or 5%, we rewrote the last paragraph of section 3.3. (Impact of the cloud cover) and we added the figure A3 to illustrate this impact on the LSCF method*

**New paragraph:**

*Furthermore, the filtering of data removing those with a significant cloud cover not only affects the number of estimates but also impacts the performance of the methods, although to a much lesser extent. When comparing results obtained from the same images, cloud-free inversions produce slightly better results than cloud-filtered inversions (Fig A2). This is because, in images partially masked by cloud cover, some pixels containing useful information are likely removed, which can lead to less accurate determination of emissions. Consistently, if the threshold of cloud cover above which $XCO_2$ pixels of the image are discarded for the analysis is increased to 2% or 5%, the performance of the methods does not increase significantly, unlike the number of estimates, which can increase, e.g. by 12% and 29% respectively when using the LCSF method (Fig. A3).*

**New figure A3:**

[Figure]

**Figure A3: Performance of the LCSF method when estimating emissions from single images of CO2 and NO2 without considering clouds (in red) and for different cloudiness thresholds: 1% (in blue), 2% (in orange) and 5% (in green). Distributions of the relative deviations (top panel) and relative absolute deviations (bottom panel) are illustrated using violin plots. Boxes are the inter-quartiles of the distributions, the whiskers are the 5th and 95th percentiles, and the lines**

**within boxes are the medians. Numbers in the inter-quartile boxes are the number of estimates for each benchmarking scenario.**

A companion paper in preparation (Kuhlmann et al., 2023) is not a proper reference. Include only a short remark in parentheses on that in the text or refer to the code repository.

The paper has been recently published. We updated the reference accordingly:

Kuhlmann, G., Koene, E. F. M., Meier, S., Santaren, D., Broquet, G., Chevallier, F., Hakkarainen, J., Nurmela, J., Amorós, L., Tamminen, J., and Brunner, D.: The *ddeq* Python library for point source quantification from remote sensing images (Version 1.0). Geoscientific Model Development, 17(12), 4773-4789, https://doi.org/10.5194/gmd-17-4773-2024, 2024.

In the section "code and data availability", we give also the address of the code repository (`https://gitlab.com/empa503/remote-sensing/ddeq`)

Some acronyms are defined several times in abstract and text, including section headers. Usually it should be defined only at the first occurrence.

We assume that this comment mainly refers to the definition of acronyms both in the abstract and in the introduction. However, we think that the main text cannot rely on definitions made in the abstract, e.g. see the point 4. in https://www.lithoguru.com/scientist/litho_papers/JM3%20editorial%202012%20q4_Acronyms.pdf.

Therefore, we prefer to keep these definitions in both the abstract and introduction in this new version of the manuscript, and let the editor rectify this if needed.

*Specific remarks*

Line 13 and 47: Conflicting definitions of an acronym.

We keep the definition of CO2M as "Copernicus CO2 monitoring mission" (Line 13). We then remove the definition at line 47 and only keep the acronym.

Line 16: Insert 'mole fractions' or 'volume mixing ratio'.

As suggested, we insert in the text a proper definition of XCO2: "To support the development of the operational processing of satellite *column-averaged $CO_2$ dry air mole fraction* ($XCO_2$) and $NO_2$ imagery.

Line 17: Is that 'tropospheric column NO2' like later in the text? Please be consistent.

Yes, for sake of simplicity, tropospheric column NO2 is referred to later in the text as NO2.

This simplification is now mentioned in the introduction:

*Each satellite will carry an imaging spectrometer providing images of $XCO_2$ and of $NO_2$ tropospheric column densities **(referred to as $NO_2$ hereinafter)** along a 250 km wide swath with a resolution of 2 km × 2 km (Sierk et al., 2019).*

Line 29ff: This long sentence is difficult to understand and should be split and improved for clarity.

We rephrase this sentence as:

*The GP and the LCSF methods generate the most accurate estimates from individual images. The deviations between the emission estimates and the true emissions from these two methods have similar Interquartile Ranges (IQR): between ~20% and ~60% depending on the scenarios.*

Line 55 and 108: Define acronyms.

TANGO acronym is defined as "Twin ANthropogenic Greenhouse Gas Observers" and ESA acronym as "European Space Agency".

Line 199: This is in strong contrast to the swath of CO2M (line 14). Add a remark please.

As suggested we add a remark at the end of the sentence:

*"It is derived from the method originally developed by Zheng et al. (2020). This original method was designed to estimate the $CO_2$ emissions of cities and industrial areas in China that produce atmospheric plumes clearly detectable in the relatively narrow transects of OCO-2 observations. These observations are characterized by a resolution of few $km^2$ but by a swath about 10 km wide, which is almost 25 times narrower than the ~250 km wide swath of the CO2M instruments."*

Line 220: Because of typical stack heights?

The answer is a bit complex. In case of power plants, the effective injection height of the plumes (accounting for the stack heights and plume rise) would tend to be higher than 100m. However, for the different sources in the cities, it ranges from the surface to such heights, and on average, it would tend to be lower than 100m (Brunner et al., 2024,

https://acp.copernicus.org/articles/19/4541/2019/). We would thus keep this choice of the 100m height as a pragmatic choice based on preliminary sensitivity tests (actually, it raises results that are very close to those when using 200m or 300m height). We have now added to the text the following indication

*"This result may be reflecting a trade-off between the need to account for emission injection heights higher than 100m when considering isolated power plants, and lower than 100m when considering the mix of sources within cities, whose emissions are not dominated by large power plants (Brunner et al., 2024). The automatic process of sources limits the ability to derive a case by case selection of the height for the wind extraction, but a finer option for future analysis might be to discriminate this selection as a function of the type of target (considering, at least, isolated power plants vs. urban areas)."*

Line 255ff, 275 and 280: Better use equation style with separate lines, don't repeat parts of an equation.

The text was reformatted as suggested.

Line 257: This differs from the recommendation for LCSF.

Yes, indeed. The practical derivation of the effective winds can be different between the specific versions of the different methods. Varon et al. derived an estimate of the effective wind based on 10 m wind speeds probably because they generalized cases where the analysis are supported by local meteorological stations providing near surface measurements for such a variable. As explained above, the choice made for LCSF arose from the assumption that the analysis would rely only on meteorological reanalysis, so that the effective wind should be extracted at a higher altitude, and from a pragmatic analysis of preliminary sensitivity tests. Finally, here, the effective wind is derived using the same practical computation in the version of the IME and CSF methods, i.e. using vertically GNFR-A weighted averages. This is mentioned a few lines after in the text. To emphasize the fact that the choice of an effective wind based on 10 m wind speeds is specific to the version of the IME method in Varon et al., we added "*For example*" at the beginning of the sentence.

Line 397: Also more combustion of fossil fuels for heating (cities).

As suggested, we rephrased the end of the sentence:
*…emissions are higher during winter due to increased fossil fuel consumption associated with electricity and heat production.*

Line 608: Refer to Fig.4.

Added as suggested.

No, we meant the estimates filtered based on their uncertainties. We clarify by changing: "*when estimates are not filtered*" by "*when estimates are not selected based on their uncertainty*".

Table 2: 'Cloud fraction' is confusing here. In meteorology a cloud fraction of 100% means overcast sky and not clear sky or cloud free (scenario 1, 5, 0%?). If you like to say with a 'threshold of 100%' that model simulated clouds are ignored as in Kuhlmann et al. (2021) or line 324 please say that in caption or footnote.

As suggested, we added to the caption the sentence:

*Note that a cloud fraction threshold of x% corresponds to the rejection of data pixels if the cloud cover exceeds x%, so that a cloud fraction of 100% yields full images without a loss of data pixels.*

*Technical corrections*

Corrected as suggested: covers -> cover

Corrected as suggested.

Corrected as suggested.

References: Use consistent style for the year of publication.

Indeed. We put the year of publication at the end of all the references.

Line 974ff: Is there a preprint (URL) available? Status?

This paper has been recently published, we updated the reference accordingly:

Hakkarainen, J., Kuhlmann, G., Koene, E., Santaren, D., Meier, S., Krol, M.C., van Stratum, B.J.H, Ialongo, I., Chevallier, F., Tamminen, J., Brunner, D., Broquet, G. Analyzing nitrogen dioxide to nitrogen oxide scaling factors for data-driven  satellite-based emission estimation

methods: a case study of Matimba/Medupi power stations in South Africa, Atmospheric Pollution Research, Volume 15, Issue 7, 2024, 102171, ISSN 1309-1042, https://doi.org/10.1016/j.apr.2024.102171, 2024.

Skip lines 991 to 994 (twice!).

Corrected as suggested

Line 1018f: In the doi of this important paper '689838' is missing at the end.

Corrected as suggested

Line 1020ff: This is not a reference. Status?

The paper has been recently published, we updated the reference accordingly:

Kuhlmann, G., Koene, E. F. M., Meier, S., Santaren, D., Broquet, G., Chevallier, F., Hakkarainen, J., Nurmela, J., Amorós, L., Tamminen, J., and Brunner, D.: The ddeq Python library for point source quantification from remote sensing images (Version 1.0). Geoscientific Model Development, 17(12), 4773-4789, https://doi.org/10.5194/gmd-17-4773-2024, 2024.

Figure 3: Shouldn't it be [ ] in the legends?

Corrected as suggested

Figure 4, 9: Mention the meaning of the numbers in the inter-quartile boxes in the caption (number of estimates?).

Yes. We have updated the captions accordingly: …***Numbers in the inter-quartile boxes are the number of estimates for each benchmarking scenario and inversion method.***

Figure 5, 6, 7, A4, A7: Remove '.' in units at label, better write '$CO_2$ emissions (Mt yr$_{-1}$)'.

Corrected as suggested. Captions were also corrected.

Figure 6, 7, A3, A4, A7, A8: Jänschwalde! Correct spelling!

Corrected as suggested

Figure 8, A5, A6: Legend and caption inconsistent concerning percentiles.

Captions corrected: 90th percentiles -> 95th percentiles

Table 1: What is 'mn'? Should it be 'min' for minute?

Corrected as suggested.

---

## Author Comment (AC2)

**Responses to Referee#2 (Steffen Beirle) comments and appendix A:**

The study "Benchmarking data-driven inversion methods for the estimation of local CO2 emissions from XCO2 and NO2 satellite images" investigates the performance of various approaches for the quantification of CO2 emissions from synthetic satellite images. As the quantification of CO2 emissions from upcoming satellites is a very important task with strong political and economical impact, this study provides an important scientific contribution. The study is generally well written, and matches the cope of AMT. However, I see one of the main results of this study, i.e. the rather poor performance of the divergence method, related to the way this method was implemented (temporal mean only), which is quite different from for treatment of the other methods (single images). I thus recommend publication after dealing with the comments below.

Thank you for your positive comments. We agree that focusing on a version of the divergence method which provides temporal mean fluxes only in the comparison to the other methods, which also provide single image estimates, can be questioned. However, the overarching goal of our paper is to benchmark data-driven methods in their current standard versions, i.e. in the most common way they are used. And, the different versions of the divergence method that have been studied in recent years provide temporal-averaged (mostly annual) emissions (Beirle et al., 2019, 2021; Hakkarainen et al., 2022, Sun, 2022). To our knowledge, there is a lack of study on the use of the divergence method to estimate instant emissions from single satellite images, and thus of knowledge about how to adapt this method for such an application. This explains why we had to conduct a series of sensitivity experiments to properly address the following comments, as detailed below.

Actually, a second reason for which we did not investigate the application of the divergence method to estimate emissions from single overpasses in the first version of our manuscript was that the first insights from the analysis of divergence maps suggested that they were unusable for this purpose: these maps extracted from single images can indeed show important variabilities as shown by the example of the strong emitting power plant of Janschwalde for January 12 (Figs R1, R2). This and other examples suggested that the computation of the emissions from single divergence maps using direct integration lacks reliability when applied to this type of data. Our parallel analysis on TROPOMI SO2 data (not documented in this study) strengthened this assumption, suggesting that the problem was not specific to the application to the SMARTCARB dataset, but likely a methodological limitation of the divergence approach based on the spatial integration of the divergence signal (called hereinafter *integral* divergence approach): small changes in the integration radius could lead to large changes in the estimated emissions suggesting that this approach was not very robust, as shown in the example of the figure R1.

This is why, in the first version of the manuscript, we chose the peak fitting approach to extract the emissions from the divergence map instead of integrating the divergence signal (Beirle et al., 2019, 2023). This approach works better with noisy

data thanks to the natural smoothing effect of the bivariate Gaussian function and provides relatively accurate annual estimates as shown in our study. However, its application to individual overpasses raises challenges. Proper and nice peaks are rarely visible in CO2 divergence maps extracted from individual images. Attempting to fit the data with the peak model often yields poor results both visually and in terms of emission estimates; the divergence map around a source hardly fits with a "peak"-like shape. This is the case for the example in Figures R1 and R2 where the determination of the parameters of the Gaussian bivariate function does not converge. The peak-fitting approach may also attempt to fit false plumes if the divergence from a given source is not strong enough or because there are numerical artifacts in the divergence map close to the source.

Our explanation for the lack of robustness of the divergence approaches when applied to single CO2 images that was suggested by this analysis lied on:

(1) the fact that we do not compute the divergence method perfectly in line with the theory. For example, as can be seen in Figure 7 of Koene et al. (2024, https://doi.org/10.1029/2023JD039904), the divergence flux map for a plume is *never* merely a simple enhancement at the source and "zero" elsewhere. There are too many violations of assumptions, e.g., that the effective wind field is perfect. Integration of the divergence flux map thus starts capturing features that are not related to source emissions at all.

(2) the presence of other (CO2) sources and sinks, meaning that a different radius selects additional but unwanted sources/sinks.

We illustrate graphically the benefit of averaging the divergence maps, using, again, results for the Janschwälde power plant. The annual averaged divergence map in the area of this power plant is given in Figure R3. In this example, the enhancement over the background is much closer to a 2D Gaussian shape and the estimate by the peak-fitting divergence approach was found to be ~44.9 MtCO2/yr. This estimate of the annual emissions is indeed close to the true value (~41.5 MtCO2/yr). The integrated divergence method also generated good results: 36.6, 46.3 and 46.3 MtCO2/yr for an integration radius of 3, 5, and 7 km respectively.

[Figure]

**Figure R1:** Divergence map estimated around the Janschwalde power station on January 2015 the 12th. Dotted circles show the different integration radii (3 km, 5 km and 7 km) used by the divergence method when emissions are derived from the integration of the divergence maps. Estimates are about 60.3, 111.2 and 120.2 MtCO2/yr when using integration radii 3, 5, and 7 km respectively. The true value being 52.2 MtCO2/yr, this suggests that the integrated divergence approach generates uncertain results which strongly depends on the integration radius when inverting single images. The peak-fitting divergence approach was not able to produce any result because it was unable to fit a Gaussian bivariate function to the divergence map.

[Figure]

**Figure R2:** XCO2 map (ppm) around the Janschwalde power station on January 2015 the 12th.

[Figure]

**Figure R3:** Map of the annual average divergence around the Jänschwalde power station for 2015

However, following this review, we have implemented the suggestions made by the referee and retrieved emission estimates from single images using several versions of the divergence approach. These versions compute the divergence maps by considering only the advective term of the divergence expression (see section 3.5 in Beirle et al., 2023) which removes the impact of the background. The results when considering one year of single overpass images showed that the performance of the integral divergence approach is much better than the peak-fitting divergence approach. This led us to conduct an extensive set of computations, varying the integration radius when applying this approach. This finally revealed that, with integration radii close to the spatial resolution of the data, the performance of the integral divergence approach can get comparable to that of the other methods in our study (Figure R4 and R5).

[Figure]

**Figure R4:** Performances of the different versions of the divergence inversion method when estimating emissions from one year of single images for different benchmarking scenarios: cloud-free $CO_2$ and $NO_2$ data with SMARTCARB winds (in blue) and cloud-filtered $CO_2$ and $NO_2$ data with SMARTCARB winds (in orange). Distributions of the relative absolute deviations are illustrated using violin plots. Boxes are the inter-quartiles of the distributions, the whiskers are the 5[th] and 95[th] percentiles, and the lines within boxes are the medians. Numbers in the inter-quartile boxes are the number of estimates for each benchmarking scenario and inversion method. Methods DIV_int_R=xkm and DIV_PeakFit are the integral (for an integration radius of x km) and peak-fitting versions of the divergence approach respectively. For a given overpass and source, the emission estimate of the method DIV_int_R=x-y-zkm is the average of the estimates when integrating over circles of x, y and z km radius around the source.

[Figure]

**Figure R5:** Performances of the inversion methods when estimating emissions from one year of

single images for different benchmarking scenarios: cloud-free $CO_2$ and $NO_2$ data with SMARTCARB winds (in blue) and cloud-filtered $CO_2$ and $NO_2$ data with SMARTCARB winds (in orange). Distributions of the relative deviations (top panel) and relative absolute deviations (bottom panel) are illustrated using violin plots. Boxes are the inter-quartiles of the distributions, the whiskers are the 5[th] and 95[th] percentiles, and the lines within boxes are the medians. Numbers in the inter-quartile boxes are the number of estimates for each benchmarking scenario and inversion method. Methods DIV_int_R=2-3-4km and DIV_PeakFit are the integral and peak-fitting versions of the divergence approach respectively. For a given overpass and source, the emission estimate of the method DIV_int_R=2-3-4km is the average of the estimates when integrating over circles of 2,3 and 4 km radius around the source.

We conclude from this series of reasoning and analysis that:

- We agree with the reviewer that the integral divergence method has a good potential to estimate emissions from single overpasses, comparable to that of the other methods tested in this study. This topic raises very interesting questions
- This deserves some discussion in our manuscript, and a documentation of the new results described here in answer to the review

but

- Fully including the integral divergence method into the comparison of emission estimates based on single images in the main text of the manuscript would require a better analysis of the potential of this approach, and thus some further investigations, which are out of the scope of this study. We managed to get performances comparable to that of the other methods based on a large but non exhaustive set of tests of sensitivity to the integration radius (Fig. R4), while our application of the other methods could rely on configurations from past experiments. Such investigations would include: testing whether the skill of the approach could be further improved by refining the integration radius, identifying why the best results are currently obtained for a radius close to the spatial resolution of the images etc... In addition to deriving meaningful indices of the uncertainties in the corresponding emission estimates.

This adds to our choice to benchmark data-driven inversion methods in standard configurations documented by previous studies.

Therefore, we have decided not to include the results of the divergence method for single-image estimates in the main part of the article, but to discuss them in the conclusion section 6 and to document them in the new Appendix A (see end of this document) as preliminary results. The paragraph inserted in the conclusion section is as follows:

*In this study, we chose not to analyze the potential of the divergence method for estimating instant emissions from single satellite overpasses because of the lack of*

*studies on such an application of this method. As highlighted in the introduction section, our aim is to compare proven approaches for the local scale estimate of strong sources (such as the application of the divergence method to time-averages of satellite images). Moreover, the strong spatial variability of the divergence fields derived from single images suggest that only averaged fields could be processed properly with the version of the divergence approach which is used here for annual estimates and which relies on the peak-fitting of temporally averaged divergence fields. However, we have conducted some preliminary analysis on a version of the divergence method which instead integrates the divergence signal spatially (over disks centered on the sources). The results, documented in appendix A, demonstrate that with a range of integration radii close to that of the spatial resolution of image, this approach can yield estimates that would be comparable in terms of accuracy and quantity to that of the best inversion methods of our benchmark evaluation for single-image based estimates. A better understanding of the behavior of this approach as a function of the integration radius, and an assessment of the estimation errors are needed to conduct a proper comparison to the other methods. This deserves further investigations. However, these preliminary results raise optimistic perspectives regarding the potential of using the divergence method for estimating instant emissions from single-overpass images*

In addition to Appendix A and to this new paragraph of the conclusion section, we added some sentences in the section describing the divergence method (section 2.1.5.) in order to mention our choice of applying the divergence method on temporal averaged maps:

Old paragraph:

*Divergence maps are computed from flux fields by using a finite difference approximation and in order to clearly detect point sources, the method needs to average the divergence fields over a long period. Here, divergence maps are averaged over one year.*

New paragraph:

*Divergence maps are computed from the mass flux field using a finite difference approximation. The divergence map is then averaged over a long period to enhance the emission signal, while reducing the impact of noise and the spatio-temporal variations of the $CO_2$ background. Here, divergence maps are averaged over one year. In theory, the divergence method can also be used to estimate emissions from single-overpass images such as the cross-sectional flux method (as the two methods are in theory similar, see Koene et al. 2024). However, we choose in this study to focus on the standard application of this method (e.g., Beirle et al. 2019, 2021, 2023; Hakkarainen et al., 2022, Sun et al., 2022), which provides temporally averaged estimates. Appendix A provides a brief overview of the performance when estimating emissions from individual images with different versions of the divergence approach.*

Finally, we mention in the introduction section that the divergence method is, in its standard version, used for inverting temporal-averaged emissions.

Old sentence:

*Contrarily to the other methods of this study, the Div method produces annual estimates from average fields extracted from multiple images.*

New sentence:

*Contrarily to the other methods of this study, the Div method is generally used to generate annual estimates from average fields extracted from multiple images.*

**Implementation of the divergence method**

The authors find the divergence method to show poorest performance for the quantification of CO2 emissions. However, I suspect that this result is partly due to the way the retrieval was done. In particular, the divergence method was treated quite differently, and - if I understood correctly - uses a different data selection then the other methods: The quantification of emissions from methods (1) to (4) require the identification of a plume. I.e. these methods are only applied to favourable conditions which are close to steady state - which is more or less assumed in all approaches. In contrast, the divergence method was applied to a temporal mean flux which probably contains unfavourable conditions as well - please clarify.

We agree with the referee and we have added in the conclusion the sentence: *However, its performance could be improved by selecting and averaging images that are characterized by favorable conditions such as strong signals or wind speeds important enough to guarantee the predominance of advective processes in the atmospheric transport.*

In any case, I don't see why the divergence method is not applied to single images as well as all other methods. For a plume as the one shown in Fig. 1, the divergence of the flux should directly yield the corresponding emissions. The motivation to use a long-term mean in Beirle et al. (2019, 2021, 2023) was that we wanted to *identify* and *localize* point sources first. These tasks are considered as solved in this study - the locations of the considered point sources are given a-priori. Thus the divergence might easily be calculated for the single image data as well, and emissions can be derived by simply integrating the divergence signal within e.g. 15 or 30 km radius around the point source.

We assessed qualitatively the effect of estimating emissions by integrating the divergence signal, using integration radii varying from 3 to 30 kms, for an example similar to the one shown in Fig. 1 of the paper. We found that emissions estimates show an important difference with respect to the true emissions and that they are very sensitive to the chosen integration radius. Probably, as the radius increases,

more and more noisy pixels are included in the integration, yielding varying results depending on the surrounding area. For this reason, estimates corresponding to integration radii greater than 10 km are far from the truth (Fig. R5).

As example, Figure R1 shows the divergence map around the Jänschwalde power plant for January the 12th. Even for this example which is characterized by a "nice" plume in amplitude and shape (Figure R1), the integrated divergence method gives estimates (60.3, 111.2 and 120.2 MtCO2/yr for an integration radius of 3,5 and 7 km respectively) that are very sensitive to the integration radius and in average far from the truth (97 vs 52.2 MtCO2/yr)

The analysis of the example above would suggest that estimating emissions with the integral divergence method from single images is unreliable. However, as discussed above, when analyzing inversion results for a whole year of single images (Figure R5), the performance of the divergence method is much better for a version of the divergence method which averages the estimates derived from the integration of divergence maps for radii of 2,3 and 4 km.

I don't see the need for an a-priori background correction, as the derivative does this automatically.

The divergence method as used in the paper is the "full" divergence method, i.e., $\nabla \cdot (VCD*U)$, while the reviewer undoubtedly expects that the divergence is computed in simplified form as $U.\nabla(VCD)$. While in the latter expression it is true that a (constant) background is removed by the derivative operation, this does not hold for the "full" divergence method. Then, rather, it really does pay to subtract the background, i.e., to compute $\nabla.( [VCD-BG] * U)$. A motivation for such an approach has been given in Koene et al. (2024): the effective wind field *U* should only describe the *enhancement* of the CO2 plume – not the total-column density-weighted wind between the surface and the top-of-the-atmosphere. As *U* is typically taken on a fixed model level (e.g., 200 m), not removing the background would mean there is a considerable mis-match between *VCD* (total column) and *U* (wind at one altitude), invalidating the assumptions of the divergence method. By removing the background, we can more safely assume that VCD-BG (the plume enhancement) and U (wind at one altitude) form a consistent data pair. Hakkarainen et al. (2022) showed as well that a background (and noise) removal should be applied to make the divergence method robust for estimating emissions from CO2 data.

However, for the integral divergence methods that have been studied in appendix A, a-priori background correction was indeed found to have a weak impact on the results as we computed the divergence of fluxes by only considering the advective term $U.\nabla(VCD)$ (see section 3.5 in Beirle et al., 2023).

Furthermore, the noise of CO2 is probably not critical neither - an outlier pixel causes a high positive and a corresponding negative derivative next to each other, which just cancel out in the spatial integral. Only outliers at the edge of the integration radius might be problematic; this effect can easily be quantified by varying the integration radius.

We agree with the reviewer and mention this in Appendix A. The impact of the edge outliers is indeed mitigated when averaging estimates from different integration radius. Figure R4 illustrates, for example, that the performance of the integral divergence method is improved when averaging estimates across integration radii of 2, 3, and 4 km, compared to using a single radius of 3 km for estimation.

Thus I would like to ask the authors to add a further simple divergence-based emission estimate by just calculating and integrating the divergence on the original (unsmoothed) CO2 data for those days where a plume could be detected, and update all figures accordingly. I consider this to be a fairer comparison to the other methods, and actually expect the divergence method to be competitive to the other methods.

As said above, we have decided not to include single-image estimates of the divergence method in the main method-result sections because we wanted to benchmark the different data-driven inversion methods in their standard version which, for the divergence method, process and generate temporal averaged quantities and because, the recently obtained results, while promising, are preliminary and need to be refined. We have however included some preliminary results in Appendix A concerning the potential of the divergence approach in estimating emissions from single images.

Title: The authors should add "synthetic" before "satellite" in order to avoid misunderstandings.

Corrected as suggested

Line 91: I would propose to have a real enumeration here with new lines for each item.

Corrected as suggested

Line 143: "however the ability of the different approaches to detect unknown point sources has not been studied here"
It might be mentioned that the divergence method is particularly suited for this task.

Following the suggestion of the reviewer, we added the sentence:

*Of mention is that the divergence, cross-sectional flux and machine-learning methods are particularly well-suited for automatic detection of plumes from unknown sources (Zheng et al., 2020; Beirle et al., 2021; Schuit et al., 2023)*

Line 284: Yes - if the focus is the detection/localization of point sources, temporal averaging is needed. But in this study, the locations are known and assumed as given for all other methods, so they should be considered as given for the divergence approach as well.

As mentioned earlier, temporal averaging helps to identify the enhancements related to CO2 emissions from a specific source in the divergence maps. In most cases, the noise in the observations, spatial variations in the background divergence field, and plumes from other sources make it difficult to distinguish the plume from a particular source in divergence maps derived from single images.

We precise this in the text by rephrasing the sentence:

Old sentence: *Divergence maps are computed from flux fields by using a finite difference approximation and in order to clearly detect point sources, the method needs to average the divergence fields over a long period*

New sentence: *Divergence maps are computed from the mass flux field using a finite difference approximation. Divergence maps are then averaged over a long period to enhance the emission signal, while reducing the impact of noise and the spatio-temporal variations of the $CO_2$ background*

Table 1: I find the statement that the "Potential for joint use of NO2 to detect plumes" is not given for the divergence method highly misleading.
The divergence does not need to detect a plume - it is based on changes of the flux. But for this, it is of course very helpful to have information on the respective divergence for NO2!

We agree with the reviewer and remove this column from the table as in addition to being misleading, it does not really provide much information.

Generally, the divergence method is the only method capable of localizing a point source without a-priori knowledge, i.e. the NO2 measurements analyzed with the divergence method could build the base for all different methods for CO2 emission estimates by providing the location of point sources. For the concrete focus of this study, the NO2 divergence might be used as indicator (filter) for favorable (steady state) conditions - if the divergence does not yield reasonable results for NO2, also the CO2 results are probably questionable and should be skipped.

The approach suggested by the reviewer is indeed a good idea, but beyond the scope of this paper. However, we have added a phrase in the conclusion to this effect:

*"However, its performance could be improved by selecting and averaging images that are characterized by favorable conditions such as strong signals or wind speeds important enough to guarantee the predominance of advective processes in the atmospheric transport"*

We have included below the appendix A that we want to include in the new version of the manuscript:

[revised manuscript text omitted]

---

## Author Response (AR2)

**Responses to Report #1**

General

Almost all of my concerns and remarks were taken into account in the text properly, including new figure and corresponding text. The paper provides now a useful overview of methods for inversion of local CO2 emissions from satellite images.

We thank the reviewer once again for the positive feedback and the careful examination of the manuscript

Technical corrections

Line 149: Check language ('Worth mentioning ...').

As suggested by the reviewer, we corrected and rephrased several sentences of the text:

Line 149: "Of mention is that…" => "It is worth mentioning that …"

Line 164: "Of particular note the fact …" => "It is worth emphasizing the fact…"

Line 426: "Note that…" => "It is worth mentioning that …"

Line 809: "biassed" => "biased"

Line 1000: "Note that the" => "Notably, the …"

Line 1296: "Note that as sink terms …" => "Sink terms…"

Line 998: Disentangle characters!

As mentioned, we disentangle in the text all the numbers followed by the character "%"

Line 1072: Which journal?

The reference is indeed outdated. We correct to:

Dumont Le Brazidec, J., Vanderbecken, P., Farchi, A., Broquet, G., Kuhlmann, G., & Bocquet, M.: Deep learning applied to $CO_2$ power plant emissions quantification using simulated satellite images. *Geoscientific Model Development*, *17*(5), 1995-2014. 2024.

There are still some '.' left in the units of Fig. 5 (right axis), Fig. A2 (16 X in legends) and Fig. A9 (axes and caption + typo in right legend). Please correct.

Corresponding figures were corrected as suggested by the reviewer

Fig. 1 is dizzy, improve quality.

As suggested, we improve the quality

**Response to Report #2**

*I highly appreciate that the authors did investigate the performance of the divergence method on single day basis in great detail, and I am excited to see that the results are quite promising. And I agree with the authors that "further investigation is required" here.*

We thank S. Beirle for his positive and constructive feedbacks, and for having pointed out the interest of applying the divergence method on single day basis, which led us to strengthen our study.

*There is only one aspect I would like to point out:  As written in my review, "The authors find the divergence method to show poorest performance for the quantification of CO2 emissions. However, I suspect that this result is partly due to the way the retrieval was done. In particular, the divergence method was treated quite differently, and - if I understood correctly - uses a different data selection than the other methods: The quantification of emissions from methods (1) to (4) require the identification of a plume. I.e. these methods are only applied to favourable conditions which are close to steady state - which is more or less assumed in all approaches. In contrast, the divergence method was applied to a temporal mean flux which probably contains unfavourable conditions as well - please clarify.*

We agree with the reviewer that via different selection processes, the other methods identify favorable cases for which they provide emission estimates (typically when the plume detection step is successful). These favorable cases generally correspond to scenes where the plumes can be distinguished from the background variations, and primarily depend on factors such as the wind conditions, the background field, or the amplitudes of the emissions. They do not necessarily correspond to steady-state conditions. However, they probably correspond to conditions which could also favour the application of the divergence method.

*In response, the authors extended the conclusions. However, I think that this is an important aspect in order to understand the poor performance of the divergence method that also needs to be discussed in the main part. Thus I would propose to - state that the application of the divergence to annual means implies that the input data sample is different that for the other methods, which were only applied when a plume could be identified (e.g. in line 106 or line 298, might also be a footnote) - remind the reader about this difference in data sample when presenting the poor performance of the divergence method: for the other methods, steady state is more or less  fulfilled (otherwise there would be no clear plume), but for the annual mean, also non-steady-state cases are averaged in. I think it is important to point out that the poor results of the divergence method are probably not caused by the method itself, but by the selection of input data.*

Following the reviewer suggestion, we emphasized in the new version of the manuscript that our implementation of the divergence method computes the annual means without any selection of the overpasses, while the other methods have internal checks which leads to a selection of favorable overpasses when estimating the annual means. This point is mentioned at the beginning of the section analyzing the results in terms of annual estimates (Section 4.1.):

*As noted earlier (Sect. 2.1.5), the Div method computes the annual emission estimate for a given source by averaging the divergence map from all available overpasses in 2015. However, the other methods select overpasses for which they succeed to detect plumes,*

*likely increasing the reliability of their estimates. These selections generally corresponds to conditions — in terms of wind, of background variability or of emission strength — that should be favorable to all methods, including the Div method. The lack of selection and thus the use of unfavourable overpasses when applying the Div method may therefore hamper the comparison between the annual estimates of the Div method and that from the other methods.*

We also rewrite the description of the divergence results in the conclusion section:

*The relatively weak performance of the Div method could be explained by the fact that this method was originally developed for the estimation of $NO_x$ emissions and the fields of this chemical species are generally characterized by stronger divergence peaks than for $CO_2$ fields. Its performance may also be hindered by the fact that our implementation of this method does not select the overpasses from which the annual divergence maps are derived (see Sect. 4.1). Further investigation is needed to determine whether the filtering of overpasses which could be favorable to the method could strongly increase the accuracy of its annual estimates.*